# Task Me Anything

**Jieyu Zhang[1], Weikai Huang[1],\* Zixian Ma[1],\* Oscar Michel[2], Dong He[1],**
**Tanmay Gupta[2], Wei-Chiu Ma[2], Ali Farhadi[1,2], Aniruddha Kembhavi[2], Ranjay Krishna[1,2]**
[1]University of Washington, [2]Allen Institute for Artificial Intelligence
https://github.com/JieyuZ2/TaskMeAnything

## Abstract

Benchmarks for large multimodal language models (MLMs) now serve to simultaneously assess the general capabilities of models instead of evaluating for a specific capability. As a result, when a developer wants to identify which models to use for their application, they are overwhelmed by the number of benchmarks and remain uncertain about which benchmark's results are most reflective of their specific use case. This paper introduces TASK-ME-ANYTHING, a benchmark generation engine which produces a benchmark tailored to a user's needs. TASK-ME-ANYTHING maintains an extendable taxonomy of visual assets and can programmatically generate a vast number of task instances. Additionally, it algorithmically addresses user queries regarding MLM performance efficiently within a computational budget. It contains 113K images, 10K videos, 2K 3D object assets, over 365 object categories, 655 attributes, and 335 relationships. It can generate 750M image/video question-answering pairs, which focus on evaluating MLM perceptual capabilities. TASK-ME-ANYTHING reveals critical insights: open-source MLMs excel in object and attribute recognition but lack spatial and temporal understanding; each model exhibits unique strengths and weaknesses; larger models generally perform better, though exceptions exist; and GPT4O demonstrates challenges in recognizing rotating/moving objects and distinguishing colors.

## 1 Introduction

Benchmarks in computer vision have traditionally served to evaluate progress towards important research problems. They shepherd the research community's attention towards a specific capability by providing reproducible evaluation protocols to identify the best solution. For example, the NYUv2 benchmark has served to identify the best model for depth estimation for the last decade [82]. In a surprising twist, the role of recent benchmarks has shifted with the advent of general-purpose large multimodal language models (MLMs) [73, 74]. This shift has similarly led to the curation of general-purpose benchmarks that assess the diversity of capabilities and not any one single capability [60, 97, 52, 51, 24, 53, 21, 77, 62]. As a result, they are now less informative to the communities they are meant to serve—researchers, developers, and users.

When a developer wants to identify which models to use for their application, they remain uncertain about which benchmark results are most aligned with their specific use case. Consider a scenario where an application developer needs a model that can most accurately identify object shapes. They may find there are existing datasets such as SHAPES [4] and CLEVR [43] that contain shape-related task instances, yet the involved objects are simple geometric primitives instead of objects in the real world. Similarly, consider a team of researchers at a big technology corporation hoping to identify the limitations of their proprietary MLM. Although MLMs are released with evaluations on benchmarks like MMBench, MMMU, BLINK and SeedBench [60, 97, 52, 51, 24], their performance across these holistic benchmarks do not pinpoint which fine-grained capabilities are lacking.

---

\* The authors contribute equally to this work.

38th Conference on Neural Information Processing Systems (NeurIPS 2024) Track on Datasets and Benchmarks.

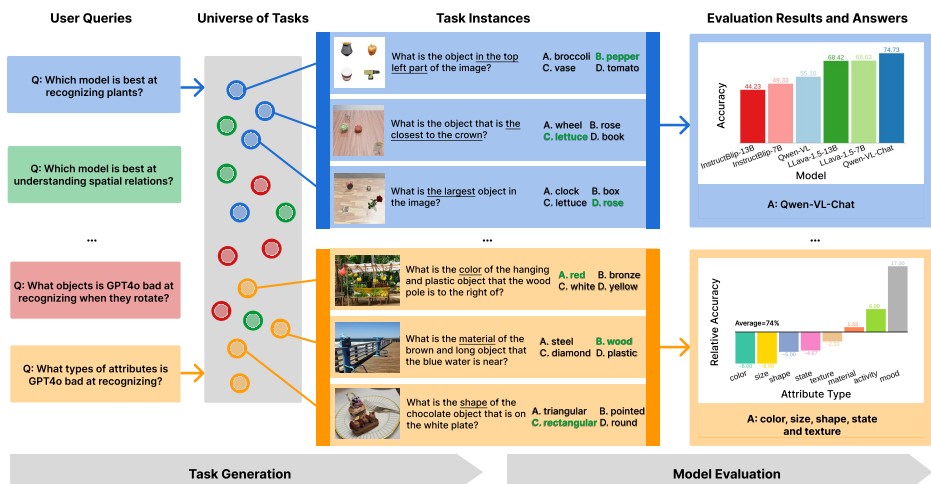

Figure 1: We present examples of user queries, corresponding task instances generated by TASK-ME-ANYTHING as well as the evaluation results on them that answer the queries.

There is a need for a principled benchmark generation process that answers task-specific user queries: "(**Q1**) Which model is the best at recognizing the shape of objects?" or "(**Q2**) what are the model's weaknesses that we can further improve on?". To actualize such a process, there are several challenges. First, we need to define an extendable taxonomy to represent the space of inputs and outputs. For example, to answer Q1, the taxonomy must include objects and their shapes. This taxonomy should be easily extendable so that future queries can evaluate new concepts. Second, the process must be able to curate a sufficient number of input-output evaluation pairs given a user query. To answer Q1, it must be able to generate thousands of images containing objects with their known shapes. Third, evaluating MLMs is computationally expensive, so the evaluation process should estimate an MLM's performance given a computation budget.

We present TASK-ME-ANYTHING, a benchmark generation engine that curates a custom benchmark given a user query (Figure 1). First, TASK-ME-ANYTHING maintains a extendable taxonomy with corresponding visual assets (*e.g.* images with scene graphs [47], 3D object assets [19], videos with spatio-temporal annotations [40], rendering softwares [15], *etc.*.). It is implemented as an extendable library where new concepts and their corresponding assets and annotations can be easily added. Second, TASK-ME-ANYTHING contains programmatic task generators which sub-select from the taxonomy to curate a large number of input-output pairs. Image/videos are either from existing datasets or programmatically generated with specific configurations. With our current taxonomy, TASK-ME-ANYTHING can generate over 750 million tasks. In comparison, existing benchmarks for MLMs have fewer task instances: MME (2,194), MMBench (3,217), BLINK (3,807), MMMU (11,550), SeedBench (19,242). Programmatic task generation is not new—CLEVR [43] and GQA [39] were also programmatically generated. While their contribution is the final generated benchmark, our contribution is the benchmark generation process itself. Third, TASK-ME-ANYTHING allows users to specify a computation budget. It contains algorithms to approximate the results of user queries via predicting the model performance across a large number of input-output pairs without actually invoking the MLM on each task instance.

The current version of TASK-ME-ANYTHING's library contains $122,866$ scene graphs [39, 29] associated with $113,018$ real images and $9,848$ real videos, $1,996$ 3D object assets [20, 19] with manual annotations, can curate 28 types of tasks (counting "how many . . .?", color questions "what color . . .?", etc.), 365 object categories, 335 relationships, 655 attributes, and 14 spatial positions. With this, we extensively evaluate 13 open-source MLMs over 1M task instances and 18 open-source/proprietary MLMs over $8,400$ task instances, both generated by TASK-ME-ANYTHING. We then address the following questions: (1) "What perceptual capabilities do open-sourced MLMs still lack?"; (2) "Do all models lack the same perceptual capabilities?"; (3) "Do larger (or proprietary) models always exhibit superior perceptual capabilities than smaller (or open-source) ones?"; (4) "What specific capabilities does GPT4O, the recently introduced proprietary MLM, still lack?".

Our analyses produce the following takeaways: (1) open-sourced MLMs exhibit strong object and attribute recognition abilities but struggle at counting, spatial and temporal understanding. (2) while most models perform similarly across different capabilities, individual models showcase different strengths and weaknesses (*e.g.*, QWEN-VL-CHAT is good at spatial relation understanding whereas INSTRUCTBLIP-7B is exceptionally good at understanding emotional relations). (3) Larger MLMs do tend to perform better than smaller ones with a few exceptions (*e.g.*, INSTRUCTBLIP-7B outperforms INSTRUCTBLIP-13B on relation understanding). (4) The best open-source MLM is on par with if not better than the best proprietary model across skills, with a nontrivial performance margin up to 7 and 8% on spatial and 3D attribute understanding. (5) We found that recognizing rotating/moving "furniture", "food", and "plants" is more challenging for GPT4O than for other object categories like animals and vehicles, likely because these objects are typically static in the real world, and GPT4O struggles more with distinguishing colors than other attributes.

## 2   TASK-ME-ANYTHING

Consider a user who wants to know "Which open-sourced MLM is best at recognizing objects even if the object is rotating?". TASK-ME-ANYTHING provides an interface for the user to pose such questions and provides them with an answer (Figure 2). It contains a taxonomy to symbolically represent visual content. A query identifies the relevant portion of the Taxonomy required to answer the query. It also contains task generators that create input-output pairs that test for a specific capability. The Taxonomy subset is used to select the appropriate task generator. We adopt the common input-output format used in existing benchmarks, *i.e.*, all the task instances in TASK-ME-ANYTHING contain an image/video, a question, and multiple options with one ground truth answer. MLMs will be evaluated on these generated task instances and the results will be returned back to the user. Finally, it also supports queries that ask for, not just the best performing model, but also task instances ("Find top-10 task instances that GPT4O performs the worst") or taxonomy concepts ("Find the objects that GPT4O's performance is higher than a threshold"), as well as on-budget results approximation methods for such fine-grained queries. unlike most existing procedural data systems, we design TASK-ME-ANYTHING so that the generation space of tasks can be expanded by adding new source data and/or task generator code. More details in Appendix B, C, and D.

### 2.1   Taxonomy

We adopt a spatio-temporal scene graph as a representation of concepts represented in an image or video [47, 40]. In a scene graph, objects and their corresponding attributes are nodes and relationships between objects are edges. Scene graphs have already been utilized in programmatic generation of VQA task instances in datasets like GQA [39] and AGQA [29, 25]. For example, the object nodes of the scene graph can be used to create counting tasks, relationships edges can encode relative locations and generate spatial understanding tasks, and attributes can ask about color, material, physical states like rotation, etc. The scene graph representation is generic: it can be extended to incorporate concepts like lightning conditions and ask questions about the light source, illumination, and shadows [7]. In fact, we extend traditional scene graphs with 3D object assets from Objaverse [20, 19], enabling us to ask questions about any objects with available 3D models and their spatial positions, *etc.*.

### 2.2   Task generators

A task generator is a Python program that can generate VQA task instances given a subset of the taxonomy. It generates questions using templates of the type: "How many <target object> are there in the image?", where the <target object> can be filled with objects in the scene graph such as "telephone". Also, it programmatically produces the ground truth answer based on the scene graph. It synthesizes incorrect yet plausible options for each question [98]. For the visual input associated with every question, we use the images [39] and videos [29] annotated with scene graphs. However, scene graph data is expensive and therefore, limited. To facilitate diverse user queries, we programmatically generate images/videos from scene graph representations [10, 4]. Since image/video generation models can introduce potential errors into our evaluation pipeline, we leave the use of generative models to future work. Instead, we programmatically generate image/video layouts and render them using Blender [15] with 3D object models [20, 19] via the following two approaches: 1) *2D sticker image* (abbreviated to 2D): Inspired by the SHAPES dataset [4], we position individual 2D rendering

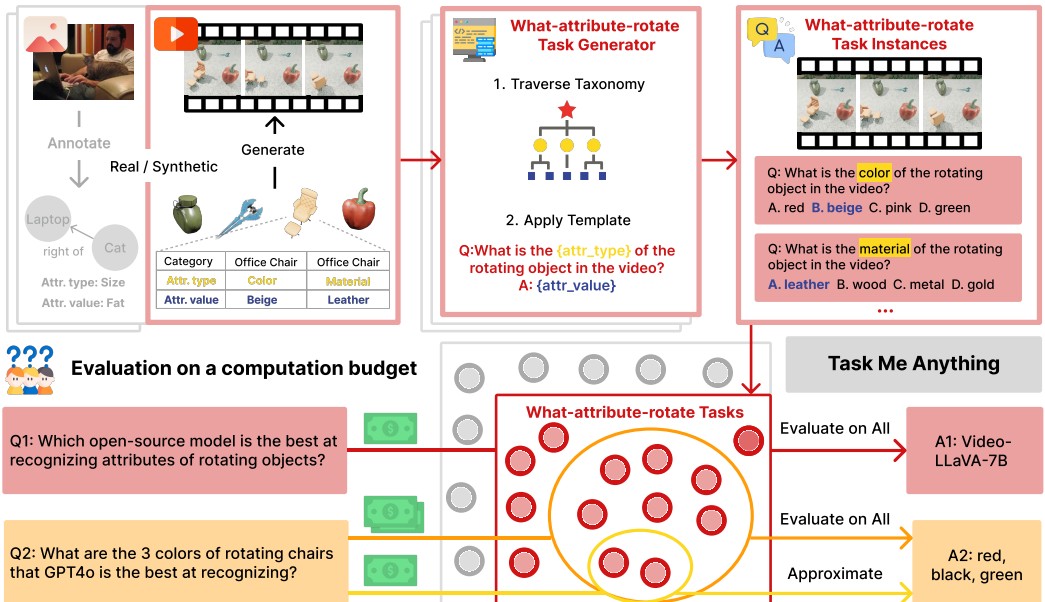

Figure 2: We present the key components in TASK-ME-ANYTHING. The top part illustrates the task generation process with an example video synthesized with 3D objects and their annotations, and the task generator for generating questions about rotating objects' attributes. The bottom part depicts the model evaluation process, which selects the relevant tasks based on the user's query and their budget and performs either full evaluation or results approximation to answer the query.

images of 3D object models in a grid (either 2x2 or 3x3) to compose an image, which is fast to generate but lack realism, *e.g.*, plausible object co-occurrences, lighting, shadows, *etc.* are absent. and 2) *3D tabletop scene* (abbreviated to 3D): To overcome the limitations of the 2D approach, we render tabletop scenes to generate images after placing the 3D object assets on the table [68]. Similarly, we generate videos and adjust the position and angle of the objects across different key frames to make objects move and rotate. Such rendered images/videos are more realistic since Blender also supports lightning and collision controls.

## 2.3 Support for different outputs

While many user queries can be simply addressed by identifying the relevant task generators and a subset of the taxonomy to generate task instances for model investigation, we additionally support 4 types of fine-grained user queries for investigations regarding individual tasks and taxonomy concepts:

① *Top-K queries* enable users to request the top-K taxonomy concepts or tasks (*e.g.*, "Return the top-10 colors/tasks that LLAVA-13B struggles with").

② *Threshold queries* allows users to query for taxonomy concepts or tasks where model performance surpasses or falls below a given threshold (*e.g.*, "Find all the object recognition tasks that both LLAVA-NEXT-34B and GPT4O perform below 30% accuracy?").

③ *Model comparison queries* identify where one model outperforms another by a specified margin, enabling comparative analysis (*e.g.*, "Which types of tasks does GPT4O outperform GEMINI-PRO?").

④ *Model debugging queries* identify where a model's performance deviates from its average by one standard deviation, facilitating the ability to uncover models' inconsistent behavior. (*e.g.*, What action does VIDEO-LLAMA-2-7B struggle to recognize compared to other actions?).

## 2.4 Evaluating on a computation budget

Given the millions of task instances that TASK-ME-ANYTHING can generate, it is computationally infeasible to evaluate even a single model on the entire task space. It would also take too long to

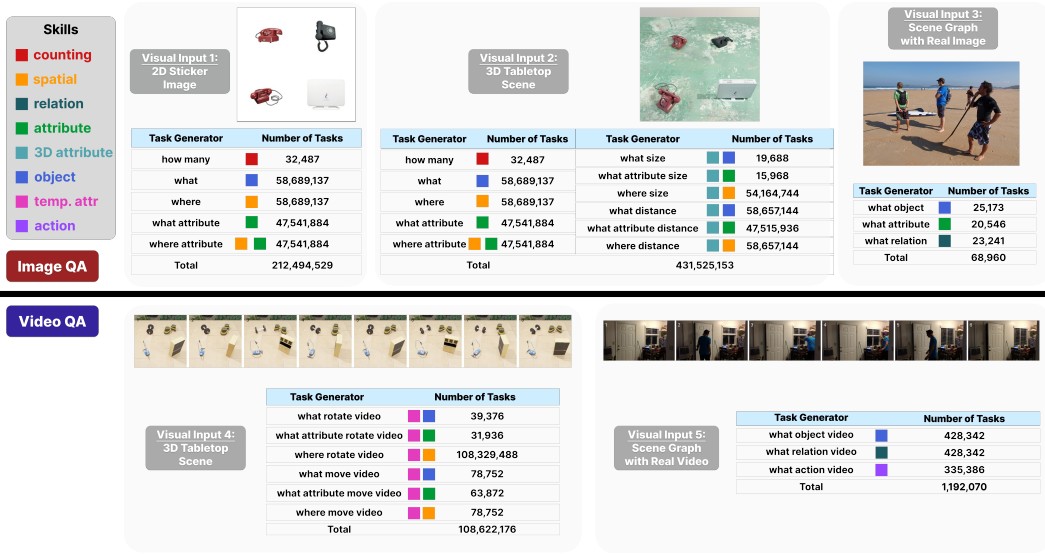

Figure 3: The statistics of generatable tasks of each task generator and example image/video in TASK-ME-ANYTHING.

be useful for everyday users. We draw on active learning literature [45], to implement 3 efficient approximation approaches:

①  *Random* randomly samples a subset of task instances from the total possible for that query. MLMs are evaluated on only this subset.

②  *Fitting* similarly samples a random subset and evaluates MLMs. The results are used to train an efficient function approximator for each MLM. This function approximator learns to predict an MLM's performance on a task, by featurizing the task-metadata—never actually generating the task instance itself. While many model choices are applicable, we adopt the Gaussian Process regressor throughout this work since it renders stable performance in preliminary studies. It uses this function to approximate the MLM's performance on the remaining task space.

③  *Active* is similar to *fitting* but iteratively trains each function approximator using active learning. Given a smaller subset, it trains an initial function, which is then used to sample the most uncertain task instances. MLMs are evaluated on these uncertain instances; the results are used to *re-train* the functions again.

## 2.5   Final benchmark engine

Although TASK-ME-ANYTHING supports many different kinds of reasoning tasks, it currently focuses on visual perception capabilities. We include 28 different task templates across 5 types of visual inputs: 2D sticker images (2D), 3D tabletop scene images/videos (3D), and real images/videos with manually-annotated scene graphs. In total, it can generate over 750 million possible VQA task instances (see Figure 3 for a breakdown). We draw image scene graphs from Visual Genome [47], and video spatio-temporal scene graphs from Action Genome [40]. We also include GQA [39] and AGQA [29] for their real VQA instances. For 2D and 3D scenes, we select $1,996$ high-quality 3D objects across 337 categories from Objaverse-LVIS, the subset of Objaverse 1.0 [20] that has been annotated with LVIS [30] categories. Each 3D object was manually annotated with attributes such as color, material, shape, and visible angles. More details can be found in Appendix E.

These 28 different task generators provide a comprehensive way to evaluate visual understanding capability including object recognition, attribute recognition, relation recognition, localization, spatial reasoning, temporal reasoning, action recognition, *etc.* (Figure 3). With this diversity of potential questions, TASK-ME-ANYTHING supports the evaluation at varying desired levels of granularity

For model users, TASK-ME-ANYTHING can help decide which model to use for their needs, and for model developers, it can identify the weaknesses of models to improve. For example, a model user wanting to find the best model for distinguishing different breeds of dogs can query: "What are the top 3 models for distinguishing dogs?" Similarly, a model developer might query: "Find the spatial

reasoning capabilities that all models lack?" to identify some general issues in current architecture. Or they might also query: "Which types of materials do LLAVA underperform on?" and then add the corresponding data into training to enhance LLAVA's material recognition performance.

This system is not only versatile but also scalable. By adding new task generators, assets like 3D object models, and software like Blender, DALL-E, etc., we can continuously expand its taxonomy. Updating a taxonomy of underlying capabilities is more scalable than collecting sufficient data for the rapid growth in use-cases for MLMs.

## 3    Validating and ablating TASK-ME-ANYTHING

We validate the accuracy of our generated evaluated data by measuring human performance on our tasks. Then, we evaluate the different approximation methods introduced in Section 2.4 to demonstrate their effectiveness.

**Validating with human evaluation.**    To gain an overview of existing MLMs' performance and validate TASK-ME-ANYTHING, we create a random subset of 300 tasks from each task generator, resulting in $5,700$ ImageQA task instances and $2,700$ VideoQA task instances, referred to as TASK-ME-ANYTHING-RANDOM, which we release as a benchmark. We first conduct a ($N = 2$) human evaluation to check the correctness of the tasks. In these random subsets, annotators achieve an accuracy of $92\% - 100\%$ for task instances from different task generators (specifically, humans perform 100% on the "how many" 2D image tasks while 92% on the "what rotate" 3D video tasks), indicating that our tasks are accurate and can be solved by humans. By contrast, GQA [39] and AGQA [29] report a human performance between $70\% - 84\%$.

**Ablating the approximation algorithms.**    We evaluate the proposed approximation algorithms on 1,137 queries across the 4 query types (Table 1). To obtain ground truth results for measuring the effectiveness of approximation, we generate over one million VQA task instances across all the task generators and evaluate 13 open-source MLM models on the generated tasks, leading to a total number of 24,240,780 <model, task instance> evaluation pairs; We refer to this set of evaluation results as TASK-ME-ANYTHING-DB, which we release for future study of query approximation algorithms. From Table 1, we can see that the *Active* method outperforms both the *Random* and *Fitting* methods across nearly all query types, yet there is still room for future improvement. More details of experiments and results are in Appendix H.

Table 1: The performance of query results approximation algorithms. Top-K query uses Mean Rank (MR, lower is better) and Hit Rate (HR, higher is better) as metrics, while other queries use Precision (P), Recall (R), and F1-score (F1).

| Method | Top-K Query | | Threshold Query | | | Model Compare Query | | | Model Debug Query | | |
|---|---|---|---|---|---|---|---|---|---|---|---|
| | MR | HR (%) | P (%) | R (%) | F1 (%) | P (%) | R (%) | F1 (%) | P (%) | R (%) | F1 (%) |
| *Random* | 46.81 | 42.30 | 46.88 | 42.48 | 44.05 | **100.00** | 24.58 | 37.28 | 93.39 | 23.27 | 35.04 |
| *Fitting* | 34.43 | 46.77 | **47.45** | 46.34 | 46.46 | 78.42 | 47.44 | 52.59 | 83.27 | 32.04 | 43.86 |
| *Active* | **10.79** | **70.55** | 47.39 | **46.83** | **46.55** | 89.94 | **54.88** | **61.87** | 89.95 | **43.84** | **56.44** |

## 4    Analysing MLMs with TASK-ME-ANYTHING

We use TASK-ME-ANYTHING to conduct multiple analyses to highlight its different use cases, while simultaneously drawing insights about today's MLMs (More details in Appendix I). Specifically, we evaluated 18 MLMs on TASK-ME-ANYTHING-RANDOM for Query 1 and 4 and reused the evaluation results of TASK-ME-ANYTHING-DB for Query 2, 3, and 5. Finally, we leverage TASK-ME-ANYTHING to provide an in-depth analysis on GPT4O as Query 6.

### 4.1    Query 1: How do models perform over a random subset of all possible questions?

We evaluated 18 MLMs on the TASK-ME-ANYTHING-RANDOM test set (Figure 4). To demonstrate the impact of different prompts, we used two types of prompts: a succinct prompt with basic questions and options, and a detailed prompt that includes more rules and instructions. Detailed

prompts typically yield better results; however, certain models, like GPT4V, perform much better with succinct prompts, indicating that current models are still prompt-sensitive.

For ImageQA tasks, the latest open-sourced models, such as INTERNVL-CHAT-1.5-24B and LLAVA-NEXT-34B, perform better than popular proprietary models, achieving state-of-the-art performance, which is also shown in recent benchmarking results [16]. Notably, models like INSTRUCTBLIP-7B and QWEN-VL perform significantly better with detailed prompt than succinct prompt. For VideoQA tasks, we also evaluated larger or proprietary ImageQA models, like GPT4V, by concatenating four frames of a video into a single picture. Notably, VIDEO-LLAVA-7B perform much better with succinct prompts than other small open-source models.

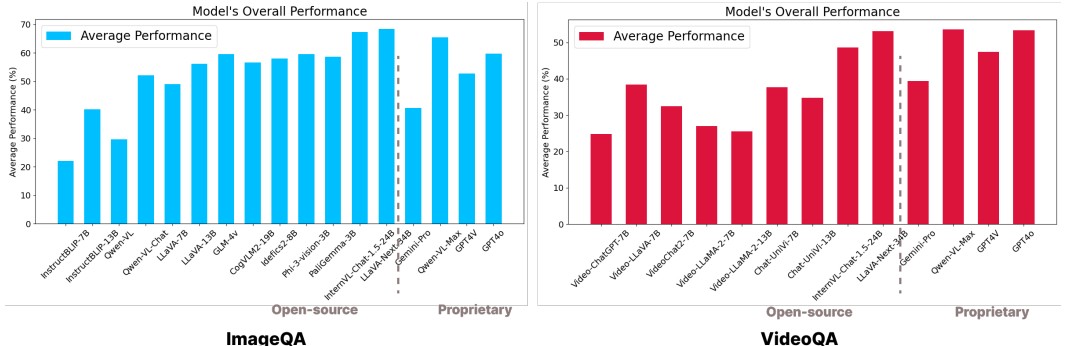

Figure 4: Model performance on the random subset of tasks from TASK-ME-ANYTHING.

## 4.2 Query 2: What skills are MLMs best and worst at?

We analyze performance across different perceptual capabilities to answer: what skills are all models good or bad at? We conduct this study for both ImageQA and VideoQA tasks respectively. We find that no specific skill appears to be the best or worst across (both image and video) models (Figure 5). We see that all models struggle in spatial reasoning, counting objects, and 3D attribute understanding on ImageQA tasks, and object recognition, temporal understanding on VideoQA tasks. They perform well on object, attribute, and other relationship recognition instances. Surprisingly, we find that most MLMs perform the best at relationship understanding between objects, scoring high if not perfectly on interactional relations such as "riding", "looking into", "lying next to" etc. On the other hand, these models struggle the most in spatial reasoning in synthetic images, performing poorly especially on questions that ask about objects in the "middle", "bottom" or "back" (for 3D images) part of the image. Nevertheless, some models behave differently. For example, LLAVA-13B is worst at recognizing 3D attributes, failing at identifying the "smallest" or "closest" 3D objects correctly. Meanwhile, LLAVA-7B is best at object recognition and worst at relation understanding, struggling to understand simple actions such as "touching" that other models perform well on.

Further, TASK-ME-ANYTHING also enables us to conduct analyses of models' fine-grained skills such as recognizing a specific type of object, attribute, or relation. For example, on ImageQA tasks, we find that on average models are better at recognizing plants, understanding mood and comprehending spatial relations between real-world objects (Figure 7). Nevertheless, some models might showcase different strengths: LLAVA-13B is better at recognizing animals (Figure 7 (a)), and INSTRUCTBLIP-7B is better at understanding emotional relationships (Figure 7 (c)).

## 4.3 Query 3: what is the best MLM for each specific skill?

LLAVA-13B stood out as the strongest model on ImageQA tasks, achieving the best performance on all skills except for relation understanding; and VIDEO-LLAVA-7B is the overall winner on VideoQA tasks, scoring the highest on action understanding and second or third elsewhere. Specifically, we find that LLAVA-13B performs consistently better than other multi-modal models on all skills except for relation understanding, where QWEN-VL-CHAT performs better (Figure 5 (a)). On VideoQA tasks, in addition to VIDEO-LLAVA-7B, CHAT-UNIVI-7B is also relatively well-rounded, positioning in the top 3 models across all skills except for Attribute understanding (Figure 5 (b)). On the other hand, while VIDEOCHAT2-7B specializes in object, attribute, and temporal attribute understanding, it falls short on Action and Relation reasoning (Figure 5 (b)).

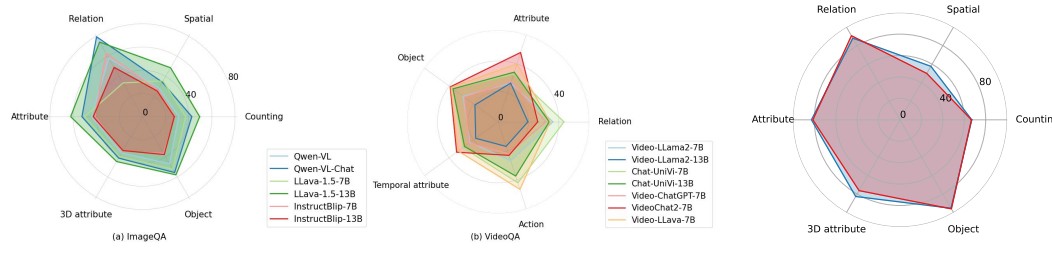

Figure 5: We plot models' performance on Image and VideoQA tasks across all skills. We learn that models are relatively good at object and attribute recognition in both Image and VideoQA and relation understanding in ImageQA but still struggle at others.

Figure 6: We plot the performance of the best open-source and proprietary model for each skill on ImageQA tasks.

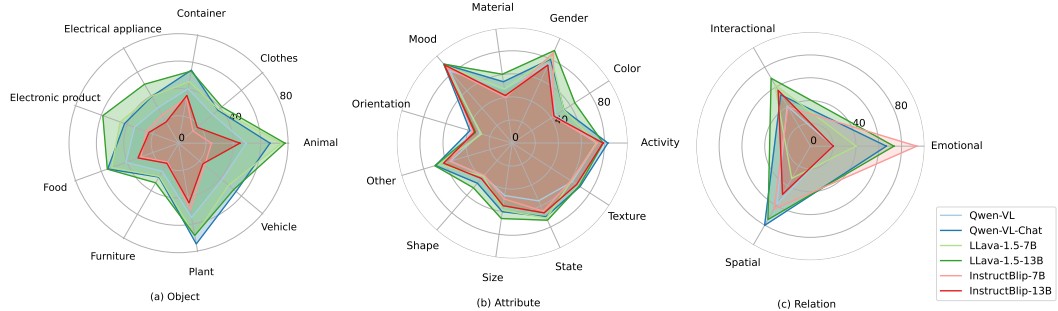

Figure 7: We also analyze models' performance on ImageQA tasks across fine-grained skills and find that models are good at recognizing plants, understanding mood, and comprehending spatial relations between real-world objects on average despite differences in individual models.

## 4.4 Query 4: How does the best open-source model compare against the best proprietary model across skills?

Moreover, we find that the best open-source model (LLAVA-NEXT-34B on object recognition, LLAVA-13B on relation understanding and INTERNVL-CHAT-1.5-24B else where) is on par with if not better than the best proprietary model (GPT4O on attribute recognition, GPT4V on counting and QWEN-VL-CHAT else where) for most skills (Figure 6). Notably, the best open-source model outperforms the best proprietary one on spatial reasoning by around 8% and 3D attribute by 7%.

## 4.5 Query 5: How do small models compare against large models?

We are also interested in the relative performance of small versus large models with the same skills. On ImageQA tasks, for example, we observe that large multi-modal models collectively perform better than smaller models on ImageQA tasks (Appendix I). Nevertheless, this finding might not always hold for individual models. Through t-tests with pairs of small and large models from the same source, we find one exception: INSTRUCTBLIP-7B ($\mu = 0.63$) significantly outperforms INSTRUCTBLIP-13B ($\mu = 0.49$) on relation understanding (with $p$-value $< 1e - 5$).

## 4.6 Query 6: What is today's popular proprietary model, GPT4O, bad at?

Finally, we investigate GPT4O, today's popular proprietary model: what *objects* are GPT4O bad at recognizing when rotating/moving? what *relations* are GPT4O bad at understanding? and what *attributes* of objects are GPT4O bad at recognizing? To answer these questions, we first identify task generators for each question that can generate relevant tasks to evaluate, based on which we provide both the object/relation/attribute categories and individuals that GPT4O are bad at.

**Answering with object/relation/attribute categories.** First, we answer these questions by comparing GPT4O's performance across different coarse-grained object/relation/attribute categories and their average, as shown in Figure 8. We can see that 1) GPT4O does not perform well in recognizing "interactional" relations in images and "spatial" relations in videos, 2) recognizing rotating/moving "furniture", "food", and "plant" is more challenging for GPT4O than other object categories such as animal and vehicle, and 3) GPT4O is worse at recognizing "color" than other attributes.

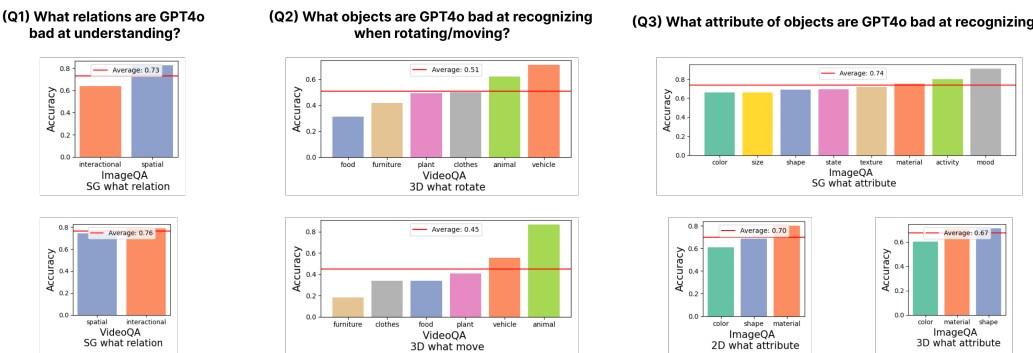

Figure 8: Answering Q1-Q3 with GPT4O performance on randomly generated task instances relating to coarse-grained object/relation/attribute categories.

**Answering with individual objects/relations/attributes.** To pinpoint the specific objects/relations/attributes that GPT4O can't do well, we convert each question to a Top-K query regarding individual objects/relations/attributes, and employ our *Active* method for query results approximation with a budget of GPT4O calls. We found that GPT4O's performance drops by a large margin ($-5\%$ to $-50\%$) on the Top-5 objects/relations/attributes founded by TASK-ME-ANYTHING, indicating they remain challenging for GPT4O. This example use case of TASK-ME-ANYTHING demonstrates how to leverage the system for locating the model weakness regarding fine-grained concepts.

## 5 TASK-ME-ANYTHING-2024 Benchmark

Finally, we introduce TASK-ME-ANYTHING-2024, a benchmark specifically designed to highlight tasks that popular MLMs are still struggling with. With TASK-ME-ANYTHING's user query benchmark generation process and query approximation algorithms, we can automatically find out the challenging tasks among 750 million task space under an extremely small budget, this benchmark can provide a comprehensive reflection of the capabilities and limitations of current MLMs. This, in turn, offers a clearer picture of their stages towards achieving human-level vision-language understanding.

### 5.1 TASK-ME-ANYTHING-2024 Generation Process.

We conducted Top-K queries with active approximation algorithms over popular open-source MLMs across all task types in TASK-ME-ANYTHING. In each task type, we only provide 300 budget (inference times) for each model to query the worst-performing tasks. For each task type (e.g., 3d-how-many), we collected the Top 10 worst-performing VQA questions for each model in both succinct and detailed prompts. By combining the worst-performing VQA questions of each model, we obtained 12,270 ImageQA and 3,567 VideoQA questions that current popular MLMs are struggling with as our TASK-ME-ANYTHING-2024 benchmark.

### 5.2 Results and analysis.

We then evaluate popular open-sourced models and GPT4O on TASK-ME-ANYTHING-2024 dataset (Figure 9). Comparing the models' performance on TASK-ME-ANYTHING-RANDOM and TASK-ME-ANYTHING-2024, we observed a 10-30% performance drop for each model. Importantly, GPT4O, which wasn't involved in the TASK-ME-ANYTHING-2024 generation process, also performed significantly worse. These results validate that TASK-ME-ANYTHING-2024 is more challenging than TASK-ME-ANYTHING-RANDOM, and our generation process indeed found the questions where popular visual models broadly suffer. With this procedure, we can automatically identify tasks that

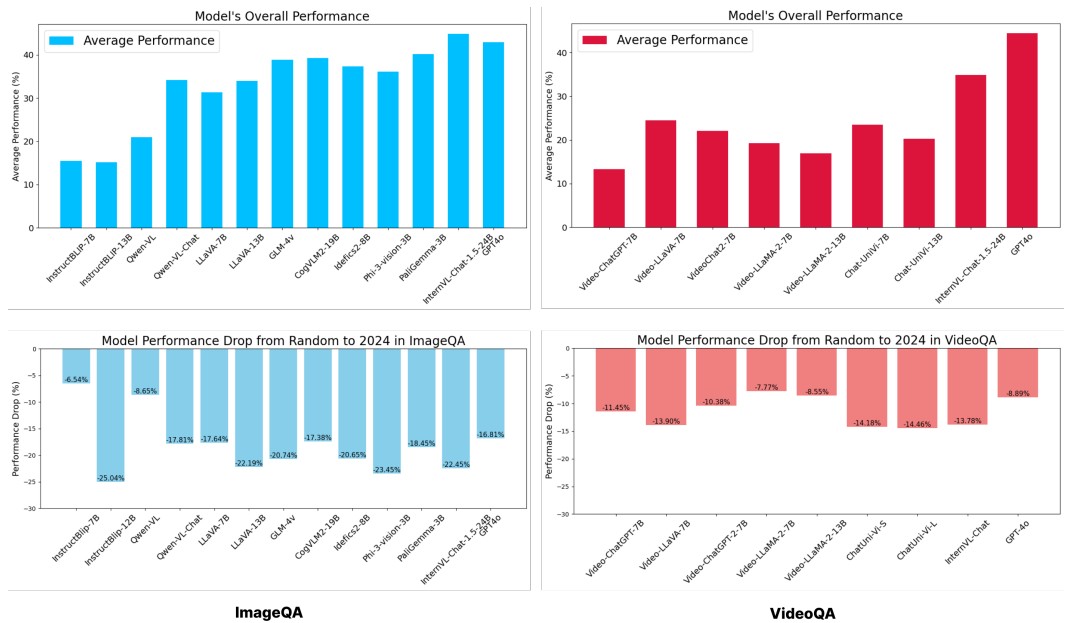

Figure 9: Model's overall performance on TASK-ME-ANYTHING-2024 and averaged model's performance drop from TASK-ME-ANYTHING-RANDOM to TASK-ME-ANYTHING-2024.

are challenging for models at any time, facilitating targeted improvements and helping researchers focus on specific weaknesses. This ensures that models are continuously evaluated and refined based on the most relevant and difficult tasks.

## 6   Conclusion

In this work, we introduce TASK-ME-ANYTHING, a task generation and evaluation system designed to address user queries with different evaluation objectives. We conduct various analyses and case studies based on TASK-ME-ANYTHING and existing MLMs, and offer many insights to the headroom for future model improvements. There are some limitations in this first version of TASK-ME-ANYTHING. For example, the current task space is more about models' perceptual capabilities and don't test for complex reasoning capabilities, which we plan to address in future versions by adding more task generators into TASK-ME-ANYTHING.

