## Acknowledgement

This project was partially funded by Toyota Motor Corporation.


# Contents

# A Discussion

## A.1 Related Work

We situate our work amongst existing work on large multimodal language models, programmatic task generation, and model-adaptive testing and debugging.

**Large multimodal language models (MLMs).** In recent years, large multimodal language models, by integrating visual encoders within various pretrained large languages models [94, 36, 11, 95, 83, 64, 86, 92, 56, 8, 12, 59, 75, 13, 80, 56, 63, 50, 85, 70, 5, 84], have progressively driven advancements in visual-language learning. With ubiquitous open-sourced LLM backbones and the increasing data for visual instruction tuning. Models like InstructBlip [18], QwenVL [6], LLaVA [58], InternVL [14], etc, have achieved unprecedented visual understanding performance in nearly all kind of visual tasks. Not only for static images, in the filed of video, by adding temporal information into the training and fine-tuning process. Models like VideoLLaMA [100], VideoChatGPT [65], ChatUnivi [42], VideoLLaVA [55], and VideoChat2 [53] have extended their capabilities to encompass video. These models, take both visual content and language as input and output language, are being considered as a new type of foundation model. The rise of large multimodal models has catalyzed the evolution of multimodal benchmarks [22, 93, 99, 49, 101, 78, 38, 66, 102, 87, 23, 9, 103, 17, 37, 27, 57, 71, 61], making them both broader and deeper. On the breadth axis, works such as MMBench[60], SEED-Bench [52, 51] and MMMU [97] provide comprehensive and integrated VQA benchmarks to evaluate a model's performance overall. On the depth axis, efforts like MathVista [62], Blink [24], MultipanelVQA [21], and Lance [77] focus on specific areas of visual tasks, such as spatial reasoning, multipanel images understanding, counterfactual images understanding, etc. To evaluate the models' ability in specific domains or tasks.

**Programmatic task generation.** Leveraging program to generate scalable and controllable benchmark data to evaluate models has been explored in various tasks, Within the task of VQA. Early attempts like the CLEVR [43] dataset, which generates simple 3D shapes to test models' visual reasoning, GQA dataset[39], using programs to generate questions from real images have achieved great success. The advent of vision models has given them the ability to tackle more complicated and compositional vision tasks, and the need for comprehensive and complex programmatic benchmarks has emerged. SimVQA[10], integrated 3D models and simulated 3D environments, to generate photo-realistic, multi-physics synthetic scenarios with questions. Moreover, leveraging the advantages of programmatic benchmark generation, such as those used in 3DB [48], allows for precise targeting and identification of subgroups where models underperform.

**Model-adaptive testing and debugging.** In the past decades, we used the static "training set, test set" paradigm to evaluate the model's performance. However, as the foundation models are all trained on a wide spectrum of datasets, this paradigm might face overfitting and data contamination issues, which makes it hard to evaluate the performance of a model fairly and truly. Model-adaptive testing and debugging, consequently, emerges to solve this problem. The key idea is 1): dynamically update the test data to prevent overfitting and data contamination. Dynabench [46], for instance, uses human and model collaboration to create challenging benchmarks. Additionally, LatestEval [54] uses the latest texts to evaluate the model, avoiding training data overlap, and [96] automates dataset updates through stylistically similar samples generated by LLMs. 2): adaptively identify subgroups where models underperform and adjust task ratios accordingly. AdaVision [26], an interactive tool for iterative testing and refinement of computer vision models, pinpoints and addresses their systematic failures with user involvement. Moreover, [88]'s 3S Testing employs synthetic data to focus evaluations on minority subgroups and distributional shifts. Lifelong Benchmarks [76] proposes dynamically expanding benchmarks and an innovative algorithm to handle the increasing data and evaluation demands efficiently.

## A.2 Limitation

**Programmatically generated tasks can be unrealistic and biased.** Programmatically generated tasks can lack the complexity and variability found in real-world data. These tasks might not capture the nuances of real-world scenarios, leading to models that perform well on synthetic data but fail in practical applications. The constraints and rules defined in the code may oversimplify the

tasks, making them easier for models to solve compared to real-world tasks. This can result in overestimating a model's capabilities. The rules and logic used to generate tasks can inadvertently introduce biases. For example, if the code disproportionately generates certain types of objects or scenarios, the model may not be adequately tested on a diverse range of tasks.

**Designing the task space is challenging.** Identifying and defining the relevant attributes for each task type (e.g., object recognition) requires deep domain knowledge and understanding of what aspects are critical for evaluating model performance. The task space must be comprehensive enough to cover various scenarios but not so complex that it becomes infeasible to manage or evaluate. Striking this balance is a significant challenge. The task space should be designed to ensure comprehensive coverage of all relevant scenarios and diversity in the types of tasks. This requires meticulous planning and consideration of all possible task variations.

**Adding new task generators requires coding skills.** Adding new task generators involves programming and understanding the underlying framework used for task generation. This requires technical expertise, which may not be available for all communities and can be a barrier for non-technical researchers who might have valuable insights and ideas for new tasks but lack the coding ability to implement them.

**Query results approximation can be inaccurate.** Efficient query results approximation within certain budgets might sometimes yield inaccurate results, especially when the budget limits are constrained. This inaccuracy can stem from several factors. First, the models that embed tasks into vectors may not fully capture all the details and nuances between different tasks. Second, the algorithms used for querying might have inherent limitations or room for improvement, affecting the precision of the results. Addressing these issues requires ongoing refinement of both the task embedding models and the query algorithms to enhance their ability to deliver accurate approximations under varying computational budgets.

## A.3   Potential negative social impact

**Misuse for malicious benchmarks.** TASK-ME-ANYTHING's ability to generate a vast number of tasks could be misused to create benchmarks specifically designed to trick or expose vulnerabilities in AI systems. Malicious actors might use this capability to create benchmarks that mislead researchers or lead to the development of AI models with undesirable biases or vulnerabilities.

**Reinforcement of biases and discrimination.** If TASK-ME-ANYTHING's task generators are not carefully designed and curated, they could inadvertently perpetuate existing biases present in the source data. This could lead to the development of AI models that are biased against certain groups of people or perpetuate harmful stereotypes.

**Overreliance on synthetic tasks.** The focus on synthetic task generation could lead to a disconnect between evaluation results and real-world performance. Overreliance on synthetic tasks might create a false sense of progress and hinder the development of AI models that can effectively address real-world challenges.

**Data contamination.** Fine-tuning models on synthetic tasks generated by TASK-ME-ANYTHING could lead to data contamination, where the model learns to exploit the specific patterns and biases of the synthetic data rather than generalizing to real-world scenarios. This could result in models that perform well on synthetic benchmarks but poorly in practical applications.

**Access and fairness.** While TASK-ME-ANYTHING aims to democratize AI evaluation, the technical expertise required to implement new task generators could create barriers for researchers and practitioners from underrepresented groups, leading to a lack of diverse perspectives and potentially reinforcing existing inequalities.

## A.4   Future work

**Supporting natural language user queries.** We plan to enable natural language queries, allowing users to specify evaluation needs in plain language. This will leverage language models to translate

instructions into actionable query commands, making the system more accessible and user-friendly. This enhancement will democratize access to model evaluation, streamline the process, and reduce barriers for non-technical users, fostering a more inclusive evaluation ecosystem.

**Expanding the TASK-ME-ANYTHING system.** To further enhance the capabilities of TASK-ME-ANYTHING, we plan to extend it across a broader range of scenarios and model types. This involves integrating support for various generative models, including language models and visual generative models, which can fine-tune the evaluation of generation quality. Also, by incorporating new types of source data, we aim to enrich the diversity and relevance of the tasks generated, ensuring that the evaluation framework remains robust and comprehensive as foundation model capabilities advance. Additionally, developing new task generators will enable the creation of tasks that capture emerging AI challenges and applications, facilitating continuous adaptation to the evolving landscape of AI. This expansion will empower users from different domains to evaluate models in ways that are highly specific to their needs, ultimately contributing to more targeted and effective deployment of AI technologies.

**A new workload for database study.** TASK-ME-ANYTHING presents new opportunities for the database community to develop efficient query execution techniques on conceptual relations containing model inference results (e.g., task accuracy of many models on many tasks) that are expensive to compute and often unmaterialized when a query is issued. The idea of pre-filtering to avoid expensive computation has been proven to be effective in some database problems, such as accelerating similarity joins [67, 41] and video analytics queries [44] where computing the similarity function or running model inference on videos is expensive during query execution. In a similar vein, recent work [34, 33, 90] has proposed efficient database indexing and query execution techniques to navigate the tradeoffs between storing the model inference results on disk and computing them on-the-fly at query time. Some other efforts [3] have also proposed trading off query result accuracy for query response time. Another direction for future work is query result diversification. When a practitioner explores a set of MLMs, datasets, and tasks, they may desire to examine a diverse set of result items, e.g., tasks that are dissimilar. It would be interesting to how query result diversification techniques [28, 35] could be adapted in TASK-ME-ANYTHING's setting.

# B  Details of Task Generation

In this section, we describe the details of the programmatic task generation process in TASK-ME-ANYTHING. We focus on tasks of multiple-choice visual questions answering, including both image question answering (ImageQA) and video question answering (VideoQA).

## B.1  Key concepts

First, we introduce several key concepts and definitions in our task generation process.

**Task instance, task, and task plan.**   A task instance is an image/video, question, options, and ground truth answer tuple that comprises a single evaluation test-case. A task is a conceptual abstraction consisting of all task instances that share the same question and answer. Tasks are specified via task plans, which contain the required task metadata and configurations to create the actual task instances. For example, in tasks involving counting, the task plan specifies the categories of objects, their total numbers in the scene, and their positions in the image—such as two apples, one on the top right and one on the bottom left. The task instance then features an actual image of the target objects and includes a specific question and answer that is consistent with the arrangement of these objects in the scene. One such task instance might be an image with two apples, the question: "How many apples are there in the image?", and the answer: "2". Multiple task instances can be generated from a single task plan because other elements such as the image background and types of distractor objects can be randomized, as they are not specified in the task plan.

**Source data.**   We refer to source data as the visual data and annotations that are used to generate task instances, *e.g.*, the 3D objects from Objaverse [20, 19] and their associated annotations or the real images and scene graphs from GQA [39, 47].

**Task generator.**   Each task generator is a program that, given source data as input, generates task instances of a certain type. It achieves three main purposes: 1) it defines the schema of the task plan; 2) it can enumerate all possible task plans given the available source data; and 3) given source data and a specific task plan, it can randomly generate a task instance belonging to the task family defined by the task plan.

## B.2  The generation process

Given the source data and a task generator, one can readily generate a large number of tasks. The overall generation process consists of the following steps:

**Step 1: enumerate the task plans.**   Once the task generator is implemented, one can use it to enumerate and return all the possible task plans based on the defined schema and the source data. As each task plan consists of just the metadata of the task rather than the actual task instances, it is efficient to enumerate all the task plans and store them as a single table. Note that enumerating all possible task plans is a one-time job, since the table of task plans can be stored and reused.

**Step 2: generate task instances of a task given its task plan.**   Another core functionality of the task generator is to generate one task instance given a valid task plan. Note that the task generator may generate many different task instances because of the randomness, *e.g.*, the negative choices can be randomly sampled from possible candidates, yet since they are all generated by the same task generator with the same task plan, they would share the question and ground truth answer and are considered belonging to the same task.

**Properties.**   This task generation process exhibits several key properties:

- **Reproducible:** With our task generation process, the tasks are produced as a combination of the source data and the programs, therefore one can reproduce identical task instances with the same source data and the random seed of the program.
- **Scalable:** This task generation process is scalable for two reasons. First, it is *memory-friendly*. One only needs to store the source data and the annotations, as well as our

codebase. Even when one aims to evaluate a model on millions of task instances, since the task instances are reproducible, one can choose to generate the task instances on the fly rather than beforehand. Secondly, it is *easy to expand* the space of task that can be generated. One can increase the number of possible tasks by either adding new source data or new task generators.

- **Easy to update:** Benchmarks can contain unexpected errors, *e.g.*, annotation error [72], so the task generation process must be easy to update once the error is caught. Since our task generation process is transparent to the users, once an error is caught, it can immediately be attributed to either the error of the source data or bugs in the code of the task generators, and then be fixed. We welcome the whole community to report any flaw in our task generation process.

- **Structured task space:** Finally, each task generated by our approach is associated with a task plan composed of its metadata. This design offers a natural structure for the tasks so that they can be grouped by certain specifications of task metadata. It enables users to navigate wanted tasks by querying the table of task plans as querying a normal database. Also, it facilitates the diagnosis of models according to the task metadata.

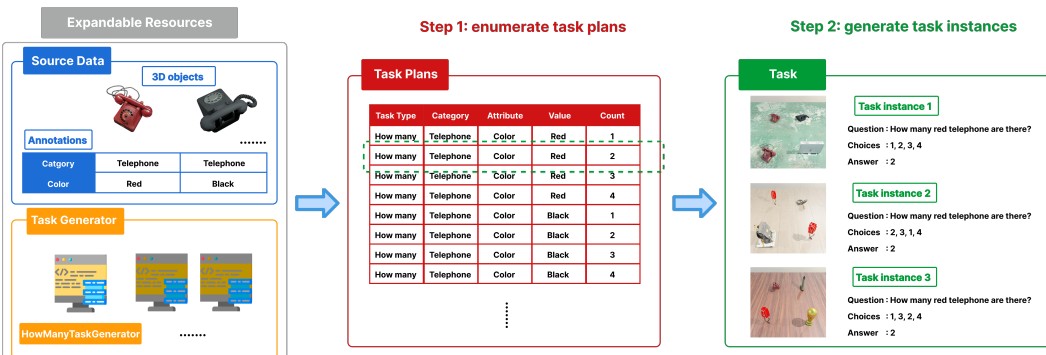

Figure 10: An illustration of core concepts and the task generation process.

# C    Details of Model Evaluation Protocol

In this section, we introduce how to evaluate MLMs against the task generated by our TASK-ME-ANYTHING system.

## C.1    Model's accuracy on a task.

We adopt the accuracy of the model on a task to capture the model's performance. However, one task can contain numerous concrete task instances. In practice, we randomly generate $n$ task instances for a task and then use the model's accuracy on the $n$ task instances as a proxy of the model's accuracy on the task.

## C.2    Prompt template and option extraction.

For prompt template, to fairly evaluate the model's performance and enhance the robustness of the results. We use two versions of prompts: a succinct prompt and a detailed prompt. The succinct version simply adds 'Select from the following choices' between the question and the options [24], while the detailed prompt includes more instructions such as: 'Based on the image/video", and also enclose the options within parentheses (e.g., "(A) camera (B) telephone")' and ends the prompt with 'Best Option: (' to guide the model to output the option only. [53] The exact prompt template can be found in Figure 11." For option extraction, we match the model output to three types of option representations: 1) option identifier, *e.g.*, "(A)", 2) option name, *e.g.*, "camera", and 3) option identifier and name, *e.g.*, "(A) camera" in order to increase the recall of the option extraction.

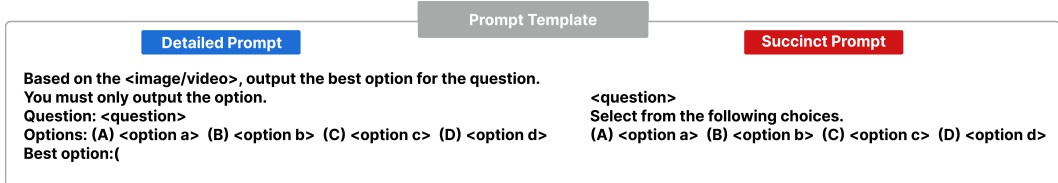

Figure 11: **Prompt Template.**

# D  Details of Fine-grained User Query and Query Approximation Algorithms

With TASK-ME-ANYTHING, most user queries regarding model performance can be simply addressed by identifying the relevant task generators and a subset of the task plans to generate task instances for model investigation. However, there is a special family of fine-grained user queries regarding individual tasks and taxonomy concepts that may require a large number of tasks to be appropriately addressed. For example, *the colors that the minimum performance of models M1, M2 is larger than 50%*; such a query involves tasks related to all the color attributes and concerns the models' performance on each individual color. In this section, we outline four types of such fine-grained user queries and discuss how to address them with efficient query results approximation.

## D.1  Fine-grained user query

We introduce four types of fine-grained user query. By default, the target of a query is the tasks, *e.g.*, Top K <task>; one can also query different task metadata or their products, *e.g.*, Top K <category> or Top K <category × attribute>.

**Top-K query.**  Users may be interested in knowing the tasks or task metadata (*e.g.*, object category) that the model(s) performs the best or the worst, which can be supported by a Top-K query. An example Top-K query in natural language is, *(E1) Top 10 "how many" tasks ranked by the maximum performance of the user-specified list of models (the user specifies all models in this case) in descending order*. This query finds the top 10 tasks that all models perform the best, measured by the maximum performance of the models on each task.

**Threshold query.**  Another useful type of query is the Threshold query, since users may want to know the tasks or task metadata on which the model's performance is larger or lower than a given threshold. An example in natural language is, *(E2) The color attributes on which the mean of the minimum performance of models M1, M2 is larger than 50%*. The query first groups tasks by their color value attribute and then aims to find the groups where the mean of the minimum performance of M1 and M2 across all tasks in the group is larger than 50%.

Built upon basic queries, one can develop new types of queries to fulfill specific needs, *e.g.*, comparing models or diagnosing the model. Here, we showcase two advanced queries based on the Threshold query: model compare and debug.

**Model Comparison query.**  A useful type of query is to support comparing a model to another. In contrast to the traditional way of comparing models by ranking based on their performance, our *Model Comparison Query* supports finding tasks or patterns where one model performs better than the other by a given threshold. An example query is *(E3) The task types on which the mean performance of model M1 is larger than model M2.*

**Model Debugging query.**  Model debugging is an important field of study for model evaluation, where the goal is to find patterns or subgroups where the model performs significantly worse or better than its average performance. To fulfill this need, we support *Model Debugging Queries* by leveraging the Threshold query with the threshold being a function of the model's average performance and a hyperparameter. For example, to find tasks where the model performs significantly worse than average, we can use the Threshold query and set the threshold to be $\mu - \sigma$, where $\mu$ is the averaged performance of the model and $\sigma$ is the standard deviation of the model performance. An example query is *(E4) The tasks on which the performance of model M1 is lower than its average performance of all tasks by a standard deviation.*

Note that these two types of query can be similarly defined based on the Top-K query, *e.g.*, the Model Debugging query can be the top k tasks that a model performs the worst, and how to define these queries depends on the user need.

## D.2  Query execution

We provide an example of the conceptual query execution process in Figure 12, which illustrates the steps required to execute query E2. Query E2 requires these steps:

1. Filter: the query filters the task plans related to "color".

2. Generate and evaluate: the query needs to generate the tasks given the obtained task plans and then evaluate model M1 and M2 against these tasks to collect their accuracy for each task.

3. Aggregate: once we obtain models' accuracy on every involved task, we perform some aggregate functions to collect the final results. We first compute the minimum accuracy of models M1 and M2 on each task. Then we average the obtained minimum accuracy over tasks within one color value group, to gather the final results for each color value group.

4. Select: for each group, the query checks whether the final result is greater than 0.5 and only keeps the groups where this filter condition holds.

## Query Execution

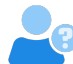

**The color attributes on which the minimum performances of models M1, M2 averaged over tasks within the group is larger than 0.5**

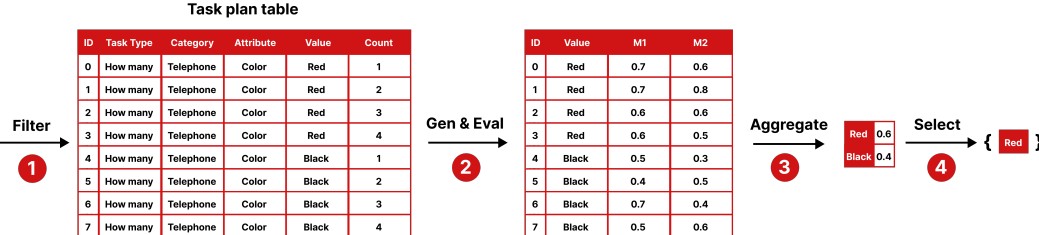

Figure 12: An illustration of the query execution process.

**Incorporating frequent pattern mining.** In practice, users may be more interested in knowing the patterns revealed by the returned tasks than the tasks themselves. Because each task in our system is associated with a task plan, one can apply frequent pattern mining [32, 91, 31] to extract frequent patterns from the set of task plans associated with the returned tasks. Note that frequent pattern mining can be applied to the results of any type of query as long as there is a set of associated task plans.

### D.3 Efficient Query Approximation Algorithms

As the fine-grained user queries may involve a large number of tasks to evaluate and therefore likely become computationally infeasible due to the compute-intensive nature of MLMs, we study three algorithms to approximate the query results given a budget of $B$ on the number of tasks to be evaluated.

**Subset proxy.** One straightforward approach to approximate the query results is to spend the budget randomly sampling $B$ tasks and then evaluate the models against them to obtain the results. Then, we use this sampled subset as a proxy of the whole set of tasks to perform the fine-grained user query.

**Fitting.** Built upon the subset proxy method, the fitting method uses the evaluation results of the $B$ randomly sampled tasks to train a model (referred to as *function approximator*) to approximate the function of interest, and then apply the model to the rest of the tasks to predict the results. In particular, the function of interest can be the model's accuracy function which inputs a task and predicts the model's accuracy, or the task aggregate function, *e.g.*, the minimum accuracy of two models as in query E2. Finally, we perform the query over all the tasks, with both actual evaluation results on $B$ sampled tasks and values of the remaining predicted by the function approximator.

**Active evaluation.** The third approach, active evaluation, builds upon the fitting method but enhances it by strategically selecting tasks to improve the approximation of query results, as opposed to relying on random sampling. This method utilizes an iterative process, where each step involves selecting a batch of unevaluated tasks based on predictions made by the current function approximator.

These tasks are then evaluated, and the results are used to re-fit the function approximator with both existing and new data until the evaluation budget is exhausted. Ultimately, the query is executed using a combination of actual results from evaluated tasks and predicted results, similar to the fitting method. The task selection criteria are tailored to the specific type of query. For the Top-K query, it selects the top-K tasks most likely to fulfill the user's inquiry based on the predicted values, because these tasks are predicted to have the most significant impact on the outcome of the query, and focusing on them could help learn a function approximator with more accurate predictions in areas that are likely relevant to the actual query results. For the Threshold query, it selects the tasks whose predicted values are closest to the threshold, because these tasks are most likely to influence the decision boundary of the function approximator and thus are critical for accurately determining the boundary's position within the task space.

**Implementation details.**    To learn a function approximator to predict the value of interest, we first need a representation of each task as the input of the approximator. We construct such representation using the task plan, question, and answer associated with each task. In particular, we convert these elements into a piece of formulated text and leverage pre-trained embedding models to calculate the text embedding as the task embedding. We adopt Gaussian Process regressor[2] because of its stable performance in our preliminary experiments, while any regression model is applicable.

---

[2]`https://scikit-learn.org/stable/modules/generated/sklearn.gaussian_process.`
`GaussianProcessRegressor.html`

# E    Details of TASK-ME-ANYTHING 1.0

In this section, we introduce the task generators implemented in the first version of TASK-ME-ANYTHING. Inspired by the model cards for model reporting [69], we make a task generator card for each implemented task generator, including information such as task type, task plan schema, *etc.*, available in the appendix, and the template can be found in Figure 13.

---

## Task Generator Card Template

- **Basic Information**.
    - **Task Type**. The target type of task, *e.g.*, ImageQA
    - **Question Type**. The type of generated question, *e.g.*, "how many"
    - **Answer Type**. The answer type *e.g.*, integer number or object category
    - **Data Type**. The type of visual content, like real images, rendering videos, etc.
- **Source Data**. The source data and annotations it requires
- **Task Plan Schema**. The schema of the associated task plans
- **Partitions**. The partition of the task space.
    - **Partition 1**.
        * **Template**. Template used to generate question if available
        * **Example**. An example of generated test case
- **Limitations**
- **Recommendations**

---

Figure 13: Summary of task generator card sections and suggested prompts for each. Task generator cards for all the included task generators can be found in Appendix M.

## E.1    Source data

**3D objects with annotations.**    We start by selecting objects from Objaverse-LVIS, the subset of Objaverse 1.0 [20] that has been annotated with LVIS [30] categories. From the set of 47K objects spanning 1,230 categories that comprise Objaverse-LVIS, we select 1,996 objects spanning 337 categories. These objects were manually chosen for their high quality and strong category alignment. We use Blender [15], an open-source ray-tracing software, to render each object from a uniform set of surrounding viewpoints and, following manual verification, only keep renderings where the object's category and attributes are discernible. This gives us a set of viewpoint annotations that we also use when constructing 3D scenes, as they allow us to ensure that the object's category and attributes are perceivable from the camera.

**Real images and videos with *Scene Graph*.**    We also collect real images and videos with scene graph [47] as part of our source data. In particular, we collect real images with scene graphs from the GQA dataset [47, 39] and real videos with scene graphs from the AGQA dataset [40, 81].

Additionally, we normalized the object terms across all source data and built a taxonomy containing 927 concepts and 965 edges using Wikidata and human filtering to avoid concept conflicts in options, such as listing both "apple" and "fruit" as choices.

## E.2    Task generators for different scenarios

### E.2.1    2D sticker image

The first scenario of TASK-ME-ANYTHING is *2D sticker image*, where we compose task instance images by compositing pre-rendered object images into a 2x2 or 3x3 grid. Such a simple type of image already enables the generation of basic types of visual questions regarding recognizing object categories and attributes, spatial relations, and counting. For example, one task could be *how many*

*red telephones are there in the image?*. We list the task generators implemented for *2D sticker image* and the statistics in Table 2.

Table 2: *2D sticker image*

| Task generator | Example question | Example answer | # of tasks |
|---|---|---|---|
| how many | How many blue objects are there in the image? | 2 | 494 |
| | How many tables are there in the image? | 4 | 6,136 |
| | How many pink beverages are there in the image? | 2 | 27,027 |
| what | What is the object in the bottom middle part of the image? | folding chair | 33,163 |
| | What is the object to the left of the telephone? | table lamp | 61,648,184 |
| where | Where is the apple in the image? | back left | 33,163 |
| | Where is the vacuum cleaner with respect to the backpack? | left | 61,648,184 |
| what attribute | What is the material of the object in the middle part of the image? | plastic | 27,027 |
| | What is the color of the object to the left of the silverware? | gold | 50,175,008 |
| where attribute | Where is the white object in the image? | top right | 27,027 |
| | Where is the gray object with respect to the lollipop? | top | 50,175,008 |
| **Total number of tasks: 223,800,421** | | | |

## E.2.2  3D tabletop scene

Although *2D sticker image* is a useful setting for generating task instances with speed, the artificial way in which the scenes are constructed through image compositing limits their realism. A real-world scene would come from objects existing in a shared 3D space that is rendered through the perspective of a single camera. As such, in *2D sticker image* we are unable to understand the effects of depth, lighting and occlusion on image understanding. To remedy this, we introduce *3D tabletop scene*, a setting analogous to *2D sticker image*, wherein objects are arranged on a plane in a shared 3D scene and rendered from a fixed camera viewpoint. This allows us to port all of the task generators from *2D sticker image* while also allowing us to test 3D-specific capabilities such as relative depth.

**ImageQA.**  Another way to generate similar yet more realistic images is to compose a 3D tabletop scene using the objects, and then render a 2D image [43]. For this *3D tabletop scene*, we can reuse task generators of *2D sticker image* with some minor modifications regarding the spatial relations. For example, the spatial relation of "in the bottom of" would become "in front of". In addition, we identify two families of task generators unique to 3D scenes: tasks regarding the size and distance of objects, which are not suitable for the 2D scenario discussed above. We list the task generators implemented for ImageQA of *3D tabletop scene* and the statistics in Table 3.

**VideoQA.**  In addition to the aforementioned ImageQA tasks, we also build VideoQA tasks for *3D tabletop scene*. We leverage two temporal attributes, rotation and movement, which can only be identified via video, to construct video-specific task generators and evaluate the models' performance in understanding temporal dynamics. To generate these videos, we keep the same layout of the 3D tabletop scene as ImageQA, but change the positions and angles of the objects across different frames of the video to make the objects move and rotate. Our task generators then target the model's ability to understand these temporal changes in object position and orientation. We list the task generators implemented for VideoQA of *3D tabletop scene* and the statistics in Table 4.

## E.2.3  Real images/videos with scene graphs

We also leverage existing manually-annotated scene graph data, *i.e.*, GQA and AGQA, to construct task generators. For ImageQA, because there are three types of nodes in the scene graph for images, *i.e.*, object, relation, and attribute, we accordingly implement three task generators to evaluate models' capability in recognizing these basic visual elements. Similarly, the scene graph for videos consists of three types of nodes, *i.e.*, object, relation, and action, we implement three task generators regarding these visual elements. We list the task generators implemented for ImageQA and VideoQA leveraging scene graphs and the statistics in Table 5&6.

Table 3: *3D tabletop scene* with images

| Task generator | Example question | Example answer | # of tasks |
|---|---|---|---|
| how many | How many blue objects are there in the image?
How many plates are there in the image?
How many black furnitures are there in the image? | 6
5
4 | 494
6,136
27,027 |
| what | What is the object in the front right part of the image?
What is the object to the right of the mobile computer? | scale
bucket | 33,163
61,648,184 |
| where | Where is the vacuum cleaner in the image?
Where is the vacuum cleaner with respect to the wine glass? | back left
left | 33,163
61,648,184 |
| what attribute | What is the color of the object in the back left part of the image?
What is the material of the object behind the plate? | red
wood | 27,027
50,175,008 |
| where attribute | Where is the wood object in the image?
Where is the white object with respect to the trophy? | front right
left | 27,027
50,175,008 |
| what size | What is the smallest object in the image? | spatula | 20,408 |
| what attribute size | What is the color of the smallest object in the image? | black | 16,632 |
| where size | Where is the largest object in the image?
Where is the smallest object in the image with respect to the car? | back left
front | 20,408
56,906,016 |
| what distance | What is the object that is farthest from the optical instrument? | juice | 61,648,184 |
| what attribute distance | What is the color of the object that is closest to the statue? | beige | 50,175,008 |
| where distance | Where is the object that is farthest from the bread in the image? | middle | 61,648,184 |
| **Total number of tasks: 454,235,261** | | | |

Table 4: *3D tabletop scene* with videos

| Task generator | Example question | Example answer | # of tasks |
|---|---|---|---|
| what rotate video | What is the object that is rotating counterclockwise in the video?
What is the rotating object in the video? | pants
jewelry | 20,408
20,408 |
| what attribute rotate video | What is the color of the object that is rotating clockwise in the video?
What is the color of the rotating object in the video? | beige
yellow | 16,632
16,632 |
| where rotate video | Where is the stepladder with respect to the rotating object in the video?
Where is the object that is rotating counterclockwise with respect to the microscope in the video? | back
front left | 51,631,112
62,221,736 |
| what move video | What is the object that is moving left in the video?
What is the moving object in the video? | serving tray
barrel | 40,816
40,816 |
| what attribute move video | What is the color of the object that is moving left in the video?
What is the color of the moving object in the video? | black
white | 33,264
33,264 |
| where move video | Where is the object that is moving down located in the video?
Where is the moving object located in the video? | back right
back right | 40,816
40,816 |
| **Total number of tasks: 114,176,720** | | | |

Table 5: Real images with *Scene Graph*

| Task generator | Example question | Example answer | # of tasks |
|---|---|---|---|
| what object | What is the flat object that is on the brown and wood table? | paper | 25,169 |
| what attribute | What is the material of the smooth object that is to the right of the yellow container? | plastic | 20,554 |
| what relation | What is the relation from the standing object, which the colorful and long snowboard is to the right of, to the blue and long object, which is to the left of the patterned skis? | holding | 23,241 |
| **Total number of tasks: 68,964** | | | |

Table 6: Real videos with *Scene Graph*.

| Task generator | Example question | Example answer | # of tasks |
|---|---|---|---|
| what object video | What is the spatial relation of the person to the closet while the person closing a closet? | floor | 428,342 |
| what relation video | What is the object that the person is behind after the person watching something in a mirror?
What is the person doing to the blanket before the person putting a phone somewhere? | behind
touching | 211,983
216,359 |
| what action video | What action is the person doing while laughing at something? | sitting at a table | 335,386 |
| **Total number of tasks: 1,192,070** | | | |

## E.3 TASK-ME-ANYTHING-UI

The ultimate goal of our query-centric model evaluation framework is to allow diverse users, including ML practitioners and non-technical users, to understand foundation models' capabilities and

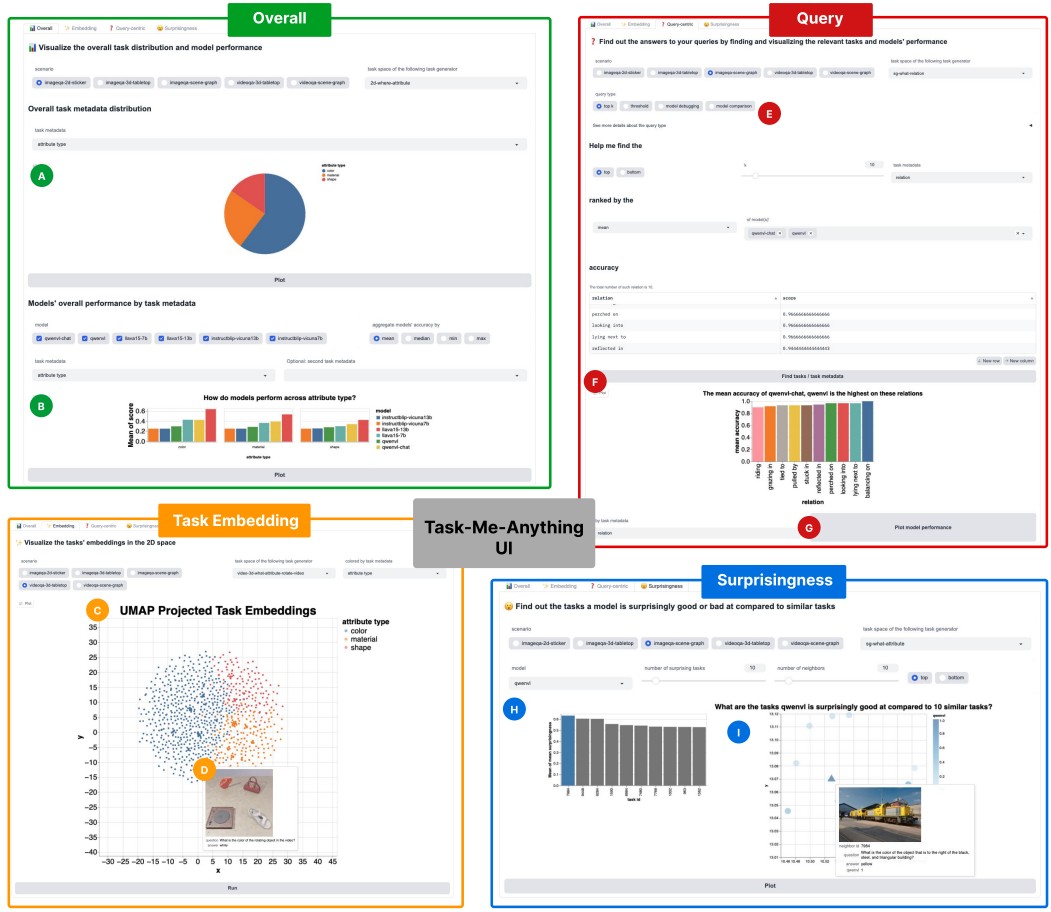

Figure 14: TASK-ME-ANYTHING-UI Interface.

limitations comprehensively and dynamically by answering their various case-specific queries. To achieve this overarching goal, we further break it down into three subgoals and aim to design an interactive end-user interface to achieve these goals:

**G1**: Support understanding of the overall task space and model performance;

**G2**: Enable deeper understanding of models through query-centric visualization of model performance (especially for common queries);

**G3**: Facilitate model debugging via discovery of surprising results.

To achieve these goals, we implemented a graphical user interface[3] with Gradio's [1] framework and used Altair [89, 79] for all the visualizations. In this section, we describe our interface in detail and how its components aim to address our design goals. Then, we present several case studies using this interface in the next section. Our interface consists of four major components organized as different tabs:

**Overall.** As the name suggests, the Overall tab is designed to help users understand the overall task distribution and model performance (**G1**). It consists of two horizontal sections for visualizing overall task distribution (Ⓐ) and models' overall performance (Ⓑ) respectively. Section A displays a pie chart of the distribution of all tasks by metadata based on user's choice of task metadata, while Section B visualizes certain models' aggregated performance in either a bar plot or heat map according to user-selected models, aggregation method and task metadata. We choose these common chart types in hopes of supporting straightfoward understanding of the overall task space and model performance.

---

[3]`https://huggingface.co/spaces/zixianma/TaskMeAnything-UI`

**Task embedding.**   In addition to the overall task distribution, we also include the Task Embedding tab to allow users to visualize all tasks at once in a 2D embedding space (**G1**). Concretely, the Embedding tab plots the 2D embeddings of all tasks reduced by UMAP as dots in a scatter plot (ⓒ). Further, we add a descriptive tooltip for each dot that displays an example image or video along with the corresponding question-answer pair for this task (ⓓ). By visualizing all tasks in one plot and enabling detail of individual tasks on demand at the same time, we hope the interface can help users understand the entire task space well on both high and low levels.

**Fine-grained user query.**   Most importantly, our interface supports query-centric visualizations of model performance under the Query-centric tab. While the space of possible user queries can be infinite, we define four common user queries: top k, threshold, model comparison and model debugging (Section 2.3) and support corresponding visualizations (Ⓔ). As these queries involve selecting a subset of tasks for visualization, we include a "Find tasks/task metadata" button to first select the relevant tasks based on the user query and return these tasks in a table (Ⓕ). If the user selects task metadata, they will have the option to visualize models' performance on the selected task metadata (Ⓖ). If the user chooses to find individual tasks however, they can additionally visualize the task distribution by some metadata, or find frequent patterns among tasks. By specifying a query first and visualizing models' performance only on selected tasks/task metadata, users can gain a more targted understanding of models based on what they are interested in (**G1**). In particular, the model debugging query can help the user find buggy model behaviors by identifying tasks/task metadata where the model's performance is lower than its global average accuracy by a large margin i.e. one standard deviation (**G2**).

**Surprisingness.**   Last but not least, we include the Surprisingness tab to help users uncover tasks where models achieve surprisingly good or bad performance compared to their performance on similar tasks (**G3**). We define the "surprisingness" of a model $M$ on a particular task $T_i$ as the following: For a task, $T_i$ and its $K$ nearest neighbors tasks $\{T_j'\}$, we compute the surprisingness score as

$$s_i^M = \frac{1}{K} \sum_{j=1}^{K} \left( \text{sim}(T_i, T_j') \times (f(T_i, M) - f(T_j', M)) \right) \tag{1}$$

A higher score indicates the model $M$ is much better at task $T_i$ than the neighbor tasks, while a lower score means $M$ is worse at $T_i$ than the neighbors.

Under the Surprisingness tab, we display the tasks where the model achieves the highest surprisingness scores in a bar chart (Ⓗ). We also make the bar chart interactive so that the user can select a particular surprising task. Then, the scatter plot on the side visualizes this model's performance on the user-selected task accordingly along with the k most similar tasks in the 2D embedding space (Ⓘ). With this interactive visualization of surprising tasks, we hope to allow users to uncover unexpected model behaviors quickly.

# F  Details of Model and Human Performance on Random Task Instances

In this section, we present the full results of our evaluation on TASK-ME-ANYTHING-RANDOM with 18 MLMs and human anntators.

## F.1  Raw results of Figure 4

Table 7: **TASK-ME-ANYTHING-RANDOM-ImageQA**. The model performance on random subsets of ImageQA tasks using both the detailed prompt and the succinct prompt. Numbers in parentheses are the number of task instances for each set.

| | 2D sticker image (1,500) | | 3D tabletop scene (3,300) | | Scene Graph (900) | |
|---|---|---|---|---|---|---|
| | Detailed prompt | Succinct prompt | Detailed prompt | Succinct prompt | Detailed prompt | Succinct prompt |
| **Human** | 99.40 | | 99.73 | | 97.33 | |
| INSTRUCTBLIP-7B | 28.27 | 0.60 | 34.48 | 0.45 | 68.33 | 0.11 |
| INSTRUCTBLIP-13B | 28.34 | 23.87 | 33.12 | 24.73 | 65.22 | 66.11 |
| QWEN-VL | 33.40 | 13.33 | 33.48 | 15.91 | 68.78 | 12.56 |
| QWEN-VL-CHAT | 40.40 | 35.87 | 38.88 | 39.36 | 78.33 | 79.45 |
| LLAVA-7B | 37.93 | 41.87 | 37.55 | 39.24 | 62.00 | 75.22 |
| LLAVA-13B | 45.60 | 43.20 | 43.97 | 42.39 | 79.22 | 82.78 |
| INTERNVL-CHAT-1.5-24B | 58.60 | 57.40 | 61.06 | 59.64 | 84.67 | 82.33 |
| LLAVA-NEXT-34B | 62.80 | 62.33 | 56.33 | 58.06 | 85.66 | 84.89 |
| GEMINI-PRO | 30.60 | 31.47 | 33.03 | 31.09 | 56.78 | 60.89 |
| QWEN-VL-MAX | 55.46 | 53.33 | 53.49 | 55.06 | 85.67 | 89.33 |
| GPT4V | 34.60 | 52.40 | 36.73 | 47.55 | 73.44 | 71.78 |
| GPT4O | 45.33 | 54.80 | 46.00 | 58.61 | 76.33 | 77.34 |

Table 8: **TASK-ME-ANYTHING-RANDOM-VideoQA**. The model performance on random subsets of VideoQA tasks using both the detailed prompt and the succinct prompt. Numbers in parentheses are the number of task instances for each set.

| | 3D tabletop scene (1,800) | | Scene Graph (900) | |
|---|---|---|---|---|
| | Detailed prompt | Succinct prompt | Detailed prompt | Succinct prompt |
| **Human** | 98.33 | | 99.33 | |
| VIDEO-CHATGPT-7B | 21.44 | 21.39 | 30.45 | 25.67 |
| VIDEO-LLAVA-7B | 26.00 | 38.78 | 32.11 | 56.67 |
| VIDEOCHAT2-7B | 30.61 | 28.55 | 37.89 | 32.89 |
| VIDEO-LLAMA-2-7B | 23.78 | 16.33 | 36.34 | 31.67 |
| VIDEO-LLAMA-2-13B | 22.67 | 20.23 | 30.78 | 28.45 |
| CHAT-UNIVI-7B | 29.72 | 25.95 | 50.11 | 45.00 |
| CHAT-UNIVI-13B | 28.17 | 25.67 | 45.22 | 39.89 |
| INTERNVL-CHAT-1.5-24B | 38.33 | 31.67 | 68.11 | 56.33 |
| LLAVA-NEXT-34B | 40.06 | 41.17 | 67.55 | 63.44 |
| GEMINI-PRO | 31.78 | 30.11 | 50.00 | 45.78 |
| QWEN-VL-MAX | 38.89 | 39.39 | 69.11 | 66.78 |
| GPT4V | 30.95 | 36.83 | 59.11 | 62.67 |
| GPT4O | 35.67 | 41.72 | 69.56 | 66.22 |

Table 9: *2D sticker image*

| | how many | | what | | what attribute | | where | | where attribute | |
|---|---|---|---|---|---|---|---|---|---|---|
| | DP | SP | DP | SP | DP | SP | DP | SP | DP | SP |
| **Human** | 100.00 | | 98.00 | | 100.00 | | 100.00 | | 99.00 | |
| INSTRUCTBLIP-7B | 23.67 | 0.00 | 24.33 | 0.00 | 39.67 | 0.00 | 27.00 | 1.00 | 26.67 | 2.00 |
| INSTRUCTBLIP-13B | 26.67 | 30.67 | 23.67 | 24.33 | 41.67 | 40.67 | 23.67 | 22.00 | 26.00 | 1.67 |
| QWEN-VL | 30.67 | 9.00 | 36.67 | 9.00 | 47.00 | 17.67 | 27.33 | 15.00 | 25.33 | 16.00 |
| QWEN-VL-CHAT | 39.67 | 24.67 | 42.67 | 42.67 | 54.67 | 52.00 | 31.67 | 33.00 | 33.33 | 27.00 |
| LLAVA-7B | 42.00 | 40.67 | 40.00 | 45.67 | 48.67 | 49.67 | 31.00 | 39.00 | 28.00 | 34.33 |
| LLAVA-13B | 49.33 | 48.33 | 46.00 | 46.67 | 58.33 | 55.33 | 39.67 | 32.67 | 34.67 | 33.00 |
| INTERNVL-CHAT-1.5-24B | 57.67 | 60.67 | 62.00 | 55.00 | **75.33** | **72.33** | **51.33** | 49.33 | 46.67 | **49.67** |
| LLAVA-NEXT-34B | **68.33** | 64.67 | 63.33 | 62.67 | 72.00 | 70.67 | **57.33** | 58.33 | 53.00 | 55.33 |
| GEMINI-PRO | 33.33 | 34.33 | 32.67 | 38.00 | 32.33 | 33.00 | 26.67 | 28.33 | 28.00 | 23.67 |
| QWEN-VL-MAX | 58.33 | 45.00 | 57.00 | 59.67 | 71.33 | 68.33 | 48.33 | 47.33 | 42.33 | 46.33 |
| GPT4V | 40.00 | **68.67** | 40.67 | 50.33 | 41.00 | 60.33 | 25.67 | 42.67 | 25.67 | 40.00 |
| GPT4O | 44.67 | 53.67 | 50.33 | **62.33** | 60.00 | 67.00 | 36.00 | 45.67 | 35.67 | 45.33 |

Table 10: *3D tabletop scene* part 1

| | how many | | what | | what attribute | | where | | where attribute | |
|---|---|---|---|---|---|---|---|---|---|---|
| | DP | SP | DP | SP | DP | SP | DP | SP | DP | SP |
| **Human** | 99.00 | | 100.00 | | 100.00 | | 99.00 | | 100.00 | |
| INSTRUCTBLIP-7B | 32.67 | 0.00 | 28.00 | 0.00 | 45.00 | 0.00 | 25.67 | 1.00 | 27.00 | 2.33 |
| INSTRUCTBLIP-13B | 32.00 | 32.33 | 22.67 | 23.33 | 42.67 | 0.00 | 28.67 | 25.33 | 23.00 | 24.67 |
| QWEN-VL | 32.33 | 11.00 | 28.00 | 8.67 | 50.67 | 19.67 | 22.67 | 18.33 | 24.67 | 15.00 |
| QWEN-VL-CHAT | 45.00 | 33.33 | 32.33 | 33.33 | 55.00 | 57.00 | 21.67 | 24.00 | 29.67 | 32.33 |
| LLAVA-7B | 38.67 | 39.33 | 32.67 | 40.33 | 57.00 | 54.00 | 27.00 | 27.67 | 26.00 | 26.00 |
| LLAVA-13B | 46.67 | 48.33 | 40.67 | 41.00 | 60.33 | 56.00 | 34.33 | 32.67 | 36.00 | 32.67 |
| INTERNVL-CHAT-1.5-24B | **67.00** | 67.00 | 60.33 | 56.33 | 68.33 | 65.67 | **54.67** | 55.67 | 46.67 | 46.00 |
| LLAVA-NEXT-34B | 63.67 | 63.33 | 49.67 | 50.67 | **71.33** | **71.33** | 48.33 | 51.00 | 40.33 | **49.00** |
| GEMINI-PRO | 40.00 | 38.67 | 32.67 | 25.00 | 31.33 | 34.67 | 28.00 | 31.00 | 27.67 | 28.00 |
| QWEN-VL-MAX | 65.00 | 60.67 | 54.67 | 55.33 | 63.67 | 61.33 | 42.33 | 44.00 | 32.67 | 37.33 |
| GPT4V | 41.67 | **66.67** | 31.67 | 37.67 | 41.33 | 54.67 | 25.00 | 39.00 | 25.67 | 28.33 |
| GPT4O | 45.00 | 64.33 | 47.33 | **58.67** | 57.33 | **68.67** | 37.67 | 45.33 | 30.67 | 44.33 |

Table 11: *3D tabletop scene* part 2

| | what distance | | where distance | | what attribute distance | | what size | | where size | | what attribute size | |
|---|---|---|---|---|---|---|---|---|---|---|---|---|
| | DP | SP | DP | SP | DP | SP | DP | SP | DP | SP | DP | SP |
| **Human** | 100.00 | | 99.00 | | 100.00 | | 100.00 | | 100.00 | | 100.00 | |
| INSTRUCTBLIP-7B | 17.67 | 0.00 | 38.33 | 0.00 | 51.00 | 0.00 | 30.33 | 0.00 | 32.33 | 1.67 | 51.33 | 0.00 |
| INSTRUCTBLIP-13B | 23.67 | 24.33 | 29.33 | 29.00 | 48.00 | 1.67 | 35.67 | 37.00 | 25.33 | 24.00 | 53.33 | 50.33 |
| QWEN-VL | 25.33 | 8.67 | 26.33 | 14.00 | 50.33 | 19.67 | 34.67 | 14.00 | 21.33 | 19.00 | 52.00 | 27.00 |
| QWEN-VL-CHAT | 25.00 | 24.00 | 25.67 | 28.33 | 56.67 | 56.00 | 43.00 | 48.67 | 31.00 | 30.67 | 62.67 | 65.33 |
| LLAVA-7B | 28.00 | 30.67 | 26.33 | 25.67 | 49.67 | 48.67 | 43.00 | 44.67 | 29.33 | 34.67 | 55.33 | 60.00 |
| LLAVA-13B | 33.67 | 29.33 | 26.00 | 23.67 | 57.67 | 55.33 | 48.33 | 48.33 | 34.67 | 35.67 | 65.33 | 63.33 |
| INTERNVL-CHAT-1.5-24B | **52.33** | 36.00 | 39.00 | **47.00** | 69.67 | 68.67 | **73.33** | 73.67 | 57.67 | 57.67 | **82.67** | 82.33 |
| LLAVA-NEXT-34B | 48.00 | 45.33 | 34.33 | **40.67** | 75.00 | **74.00** | 62.33 | 62.00 | 49.00 | **52.67** | 77.67 | **78.67** |
| GEMINI-PRO | 39.33 | 31.00 | 25.33 | 24.33 | 38.33 | 36.00 | 34.33 | 29.67 | 26.67 | 26.67 | 39.67 | 37.00 |
| QWEN-VL-MAX | 39.00 | **53.00** | 2.67 | 35.67 | 65.00 | 66.67 | 72.33 | 69.67 | 45.33 | 50.00 | 75.67 | 72.00 |
| GPT4V | 39.33 | 46.67 | 21.67 | 19.00 | 43.33 | 64.33 | 46.00 | 54.00 | 22.33 | 37.67 | 66.00 | 75.00 |
| GPT4O | 44.67 | **62.33** | 24.00 | **41.67** | 58.33 | 65.33 | 57.67 | **73.00** | 32.33 | 44.67 | 71.00 | 76.33 |

Table 12: Real images with *Scene Graph*

| | what attribute | | what object | | what relation | |
|---|---|---|---|---|---|---|
| | **DP** | **SP** | **DP** | **SP** | **DP** | **SP** |
| **Human** | 96.00 | | 99.00 | | 97.00 | |
| INSTRUCTBLIP-7B | 65.67 | 0.00 | 79.00 | 0.00 | 60.33 | 0.33 |
| INSTRUCTBLIP-13B | 66.33 | 68.67 | 84.33 | 80.00 | 45.00 | 49.67 |
| QWEN-VL | 64.00 | 4.33 | 83.33 | 8.67 | 59.00 | 24.67 |
| QWEN-VL-CHAT | 69.67 | 69.00 | 87.00 | 86.67 | 78.33 | 82.67 |
| LLAVA-7B | 70.00 | 65.33 | 85.00 | 84.33 | 31.00 | 76.00 |
| LLAVA-13B | 72.67 | 70.33 | 90.00 | 90.00 | 75.00 | **88.00** |
| INTERNVL-CHAT-1.5-24B | **80.00** | 77.33 | **94.67** | 92.00 | 79.33 | 77.67 |
| LLAVA-NEXT-34B | **78.33** | 75.33 | 93.33 | **95.33** | 85.33 | 84.00 |
| GEMINI-PRO | 51.00 | 50.67 | 71.00 | 68.67 | 48.33 | 63.33 |
| QWEN-VL-MAX | 76.67 | **81.33** | 93.67 | **96.00** | **86.67** | **90.67** |
| GPT4V | 69.33 | 67.00 | 82.67 | 79.33 | 68.33 | 69.00 |
| GPT4O | 68.00 | 67.67 | 83.00 | 81.67 | 78.00 | 82.67 |

## F.3 A breakdown of Table 8

Table 13: *3D tabletop scene*

| | what attribute move | | what attribute rotate | | what move | | what rotate | | where move | | where rotate | |
|---|---|---|---|---|---|---|---|---|---|---|---|---|
| | DP | SP | DP | SP | DP | SP | DP | SP | DP | SP | DP | SP |
| **Human** | 100.00 | | 100.00 | | 98.00 | | 92.00 | | 100.00 | | 100.00 | |
| Video-ChatGPT-7B | 27.00 | 24.33 | 27.00 | 28.33 | 18.33 | 19.00 | 15.67 | 18.67 | 27.33 | 26.33 | 13.33 | 11.67 |
| Video-LLaVA-7B | 28.33 | 54.00 | 25.00 | 49.33 | 26.00 | **34.00** | 26.67 | **35.33** | 25.00 | 31.33 | 25.00 | 28.67 |
| VideoChat2-7B | 46.67 | 48.33 | 41.33 | 47.67 | 29.00 | 22.33 | 27.67 | 19.67 | 17.00 | 14.00 | 22.00 | 19.33 |
| Video-LLaMA-2-7B | 28.67 | 24.00 | 27.67 | 25.00 | 22.33 | 19.00 | 23.33 | 16.00 | 20.00 | 7.33 | 20.67 | 6.67 |
| Video-LLaMA-2-13B | 29.67 | 26.67 | 32.33 | 32.00 | 18.33 | 17.67 | 19.33 | 17.67 | 17.67 | 14.67 | 18.67 | 12.67 |
| Chat-UniVi-7B | 36.67 | 27.67 | 35.33 | 39.67 | 27.67 | 20.33 | 28.33 | 24.00 | 25.67 | 24.00 | 24.67 | 20.00 |
| Chat-UniVi-13B | 33.67 | 31.33 | 33.67 | 37.00 | 24.33 | 22.67 | 29.33 | 28.00 | 25.33 | 16.33 | 22.67 | 18.67 |
| InternVL-Chat-1.5-24B | 52.33 | 43.00 | 56.00 | 49.33 | 26.67 | 21.00 | 31.33 | 22.67 | 31.67 | 28.00 | 32.00 | 26.00 |
| LLaVA-Next-34B | **57.67** | **56.67** | 59.00 | **62.67** | 28.00 | 29.33 | 30.67 | 29.67 | **32.33** | **32.33** | **32.67** | 36.33 |
| Gemini-Pro | 39.33 | 38.67 | 40.33 | 37.67 | **30.67** | 28.67 | 27.33 | 25.33 | 27.67 | 29.67 | 25.33 | 20.67 |
| Qwen-VL-Max | **56.33** | 52.67 | **67.33** | 67.00 | 29.00 | 30.00 | 34.00 | **35.33** | 26.00 | 25.00 | 20.67 | 26.33 |
| GPT4V | 43.67 | 51.00 | 46.67 | 57.33 | 28.00 | 29.33 | 29.67 | 32.00 | 22.00 | 26.00 | 15.67 | 25.33 |
| GPT4o | 47.67 | 46.00 | 54.67 | 62.67 | 27.33 | **31.00** | 34.33 | 38.67 | 27.00 | 36.33 | 23.00 | **35.67** |

Table 14: Real videos with *Scene Graph*

| | what action | | what object | | what relation | |
|---|---|---|---|---|---|---|
| | DP | SP | DP | SP | DP | SP |
| **Human** | 100.00 | | 98.00 | | 100.00 | |
| Video-ChatGPT-7B | 19.67 | 16.33 | 37.00 | 29.67 | 34.67 | 31.00 |
| Video-LLaVA-7B | 29.67 | 58.33 | 31.33 | 62.67 | 35.33 | 49.00 |
| VideoChat2-7B | 36.33 | 26.33 | 44.33 | 42.67 | 33.00 | 29.67 |
| Video-LLaMA-2-7B | 33.67 | 21.33 | 37.67 | 40.00 | 37.67 | 33.67 |
| Video-LLaMA-2-13B | 30.33 | 23.67 | 39.00 | 36.00 | 23.00 | 25.67 |
| Chat-UniVi-7B | 44.67 | 37.67 | 57.33 | 47.67 | 48.33 | 49.67 |
| Chat-UniVi-13B | 38.33 | 25.00 | 58.67 | 52.00 | 38.67 | 42.67 |
| InternVL-Chat-1.5-24B | **72.33** | 52.33 | **73.00** | 54.33 | 59.00 | 62.33 |
| LLaVA-Next-34B | 67.00 | 60.00 | 67.33 | 65.33 | 68.33 | 65.00 |
| Gemini-Pro | 54.33 | 39.67 | 55.00 | 53.00 | 40.67 | 44.67 |
| Qwen-VL-Max | **67.33** | **68.67** | **69.67** | **68.00** | 70.33 | 63.67 |
| GPT4V | 53.67 | 56.67 | 57.67 | 58.67 | 66.00 | **72.67** |
| GPT4o | 64.67 | 62.33 | 66.00 | 60.00 | **78.00** | **76.33** |

# G  Details of Model Performance on TaskMeAnything 2024 benchmark

In this section, we present the full results of our evaluation on TASK-ME-ANYTHING-2024 with 18 MLMs.

## G.1  Raw results of Figure 9

Table 15: **TASK-ME-ANYTHING-2024-ImageQA**. The model performance on the ImageQA split of TASK-ME-ANYTHING-2024. Numbers in parentheses are the number of task instances for each set.

| | 2D sticker image (1,500) | | 3D tabletop scene (3,300) | | Scene Graph (900) | |
|---|---|---|---|---|---|---|
| | Detailed prompt | Succinct prompt | Detailed prompt | Succinct prompt | Detailed prompt | Succinct prompt |
| INSTRUCTBLIP-7B | 25.34 | 0.31 | 26.89 | 0.06 | 38.17 | 0.33 |
| INSTRUCTBLIP-13B | 19.46 | 0.26 | 20.13 | 0.10 | 41.91 | 1.09 |
| QWEN-VL | 22.06 | 10.26 | 22.05 | 11.16 | 37.61 | 10.40 |
| QWEN-VL-CHAT | 21.81 | 19.93 | 20.07 | 24.03 | 45.63 | 48.99 |
| LLAVA-7B | 25.25 | 23.97 | 21.62 | 20.59 | 40.74 | 37.74 |
| LLAVA-13B | 22.94 | 19.75 | 22.44 | 17.93 | 41.31 | 36.75 |
| INTERNVL-CHAT-1.5-24B | 19.46 | 19.06 | 20.27 | 21.78 | 38.51 | 35.75 |
| LLAVA-NEXT-34B | 19.61 | 20.28 | 20.46 | 22.08 | 43.11 | 43.19 |
| GEMINI-PRO | 22.79 | 21.85 | 23.04 | 22.85 | 30.55 | 32.16 |
| QWEN-VL-MAX | | | | | | |
| GPT4V | 20.11 | 21.86 | 20.99 | 21.34 | 39.93 | 38.79 |
| GPT4O | 21.06 | 26.93 | 18.52 | 29.73 | 38.07 | 41.57 |

Table 16: **TASK-ME-ANYTHING-2024-VideoQA**. The model performance on the VideoQA split of TASK-ME-ANYTHING-2024. Numbers in parentheses are the number of task instances for each set.

| | 3D tabletop scene | | Scene Graph | |
|---|---|---|---|---|
| | Detailed prompt | Succinct prompt | Detailed prompt | Succinct prompt |
| VIDEO-CHATGPT-7B | 16.10 | 15.20 | 15.84 | 15.80 |
| VIDEO-LLAVA-7B | 25.19 | 22.35 | 29.03 | 20.98 |
| VIDEOCHAT2-7B | 23.00 | 20.20 | 20.29 | 13.47 |
| VIDEO-LLAMA-2-7B | 21.25 | 13.87 | 22.13 | 14.50 |
| VIDEO-LLAMA-2-13B | 20.49 | 16.94 | 19.42 | 14.33 |
| CHAT-UNIVI-7B | 21.57 | 18.99 | 24.74 | 18.82 |
| CHAT-UNIVI-13B | 21.78 | 18.22 | 24.54 | 19.04 |
| INTERNVL-CHAT-1.5-24B | 22.07 | 20.26 | 17.58 | 13.99 |
| LLAVA-NEXT-34B | 19.11 | 20.00 | 16.56 | 18.07 |
| GEMINI-PRO | 20.38 | 20.31 | 17.36 | 15.97 |
| QWEN-VL-MAX | | | | |
| GPT4V | 20.07 | 21.86 | 15.71 | 14.34 |
| GPT4O | 20.25 | 21.73 | 14.01 | 14.05 |

## G.2 A breakdown of Table 15

Table 17: *2D sticker image*

|  | how many | | what | | what attribute | | where | | where attribute | |
| --- | --- | --- | --- | --- | --- | --- | --- | --- | --- | --- |
|  | DP | SP | DP | SP | DP | SP | DP | SP | DP | SP |
| INSTRUCTBLIP-7B | 28.71 | 0.00 | 10.91 | 0.12 | 32.86 | 0.00 | 26.39 | 1.04 | 27.84 | 0.37 |
| INSTRUCTBLIP-13B | 17.64 | 0.00 | 8.54 | 0.12 | 31.57 | 0.00 | 22.34 | 0.93 | 17.22 | 0.24 |
| QWEN-VL | 29.68 | 4.01 | 12.22 | 5.10 | 31.46 | 12.79 | 20.60 | 13.66 | 16.36 | 15.75 |
| QWEN-VL-CHAT | 30.54 | 20.80 | 11.03 | 11.51 | 31.22 | 33.45 | 20.49 | 16.20 | 15.75 | 17.70 |
| LLAVA-7B | 31.87 | 32.12 | 9.96 | 11.03 | 29.93 | 24.06 | 26.04 | 24.19 | 28.45 | 28.45 |
| LLAVA-13B | 34.79 | 39.17 | 11.27 | 9.61 | 31.46 | 24.41 | 16.90 | 12.85 | 20.27 | 12.70 |
| **InternVL-Chat-24B** | 24.21 | 29.08 | 12.46 | 9.49 | 36.62 | 28.17 | 11.57 | 17.94 | 12.45 | 10.62 |
| **LLaVA-Next-34B** | 27.98 | 25.18 | 9.61 | 8.90 | 35.45 | 40.14 | 11.34 | 14.58 | 13.68 | 12.58 |
| **Gemini-Pro** | 29.68 | 30.66 | 16.37 | 16.37 | 26.53 | 26.88 | 20.25 | 17.13 | 21.12 | 18.19 |
| QWEN-VL-MAX | | | | | | | | | | |
| GPT4V | 25.18 | 35.04 | 11.15 | 10.20 | 27.82 | 35.45 | 20.02 | 17.25 | 16.36 | 11.36 |
| GPT4O | 28.47 | 39.78 | 11.86 | 16.49 | 32.04 | 39.20 | 20.72 | 24.77 | 12.21 | 14.41 |

Table 18: *3D tabletop scene* part 1

|  | how many | | what | | what attribute | | where | | where attribute | |
| --- | --- | --- | --- | --- | --- | --- | --- | --- | --- | --- |
|  | DP | SP | DP | SP | DP | SP | DP | SP | DP | SP |
| INSTRUCTBLIP-7B | 27.14 | 0.00 | 23.08 | 0.00 | 13.24 | 0.00 | 29.10 | 0.00 | 28.33 | 0.34 |
| INSTRUCTBLIP-13B | 14.64 | 0.00 | 22.03 | 0.00 | 12.41 | 0.00 | 21.53 | 0.23 | 18.32 | 0.11 |
| QWEN-VL | 32.26 | 10.24 | 20.63 | 5.48 | 26.36 | 11.35 | 16.15 | 12.94 | 15.02 | 13.20 |
| QWEN-VL-CHAT | 38.45 | 28.81 | 18.07 | 22.14 | 26.24 | 32.98 | 8.93 | 14.66 | 10.47 | 17.18 |
| LLAVA-7B | 19.64 | 17.14 | 15.85 | 21.45 | 15.01 | 19.62 | 25.66 | 17.98 | 23.09 | 15.13 |
| LLAVA-13B | 29.76 | 33.69 | 19.81 | 16.20 | 16.08 | 15.60 | 21.53 | 9.05 | 22.07 | 10.58 |
| **InternVL-Chat-24B** | 20.71 | 29.05 | 10.02 | 7.11 | 22.70 | 31.32 | 13.75 | 5.96 | 17.06 | 9.10 |
| **LLaVA-Next-34B** | 31.55 | 34.88 | 22.73 | 22.61 | 16.08 | 16.90 | 8.25 | 11.00 | 8.42 | 13.08 |
| **Gemini-Pro** | 33.10 | 35.12 | 17.02 | 18.18 | 23.17 | 25.77 | 20.85 | 18.33 | 21.05 | 20.02 |
| QWEN-VL-MAX | | | | | | | | | | |
| GPT4V | 27.50 | 32.74 | 20.98 | 18.88 | 17.61 | 26.60 | 15.58 | 5.96 | 18.66 | 6.71 |
| GPT4O | 24.52 | 40.00 | 13.99 | 14.80 | 14.30 | 30.85 | 7.90 | 14.20 | 8.65 | 17.41 |

Table 19: *3D tabletop scene* part 2

|  | what distance | | where distance | | what attribute distance | | what size | | where size | | what attribute size | |
| --- | --- | --- | --- | --- | --- | --- | --- | --- | --- | --- | --- | --- |
|  | DP | SP | DP | SP | DP | SP | DP | SP | DP | SP | DP | SP |
| INSTRUCTBLIP-7B | 21.36 | 0.23 | 37.90 | 0.00 | 26.89 | 0.00 | 19.10 | 0.00 | 35.19 | 0.12 | 34.43 | 0.00 |
| INSTRUCTBLIP-13B | 20.90 | 0.11 | 30.37 | 0.68 | 15.24 | 0.00 | 15.21 | 0.00 | 20.86 | 0.00 | 29.91 | 0.00 |
| QWEN-VL | 15.25 | 8.47 | 24.54 | 14.95 | 22.68 | 9.54 | 19.22 | 7.18 | 18.40 | 15.93 | 31.99 | 13.43 |
| QWEN-VL-CHAT | 15.25 | 24.07 | 16.89 | 22.15 | 19.45 | 19.21 | 19.34 | 22.02 | 14.44 | 26.67 | 33.21 | 34.43 |
| LLAVA-7B | 16.38 | 22.71 | 27.17 | 19.06 | 17.84 | 21.93 | 16.30 | 16.79 | 31.85 | 26.91 | 29.06 | 27.72 |
| LLAVA-13B | 26.44 | 23.50 | 16.55 | 9.93 | 19.21 | 19.58 | 17.88 | 18.49 | 28.40 | 12.84 | 29.06 | 27.72 |
| **InternVL-Chat-24B** | 11.86 | 10.96 | 20.89 | 31.05 | 22.18 | 31.35 | 23.11 | 26.89 | 18.27 | 11.23 | 42.37 | 45.54 |
| **LLaVA-Next-34B** | 33.45 | 29.72 | 14.16 | 15.18 | 18.46 | 19.83 | 23.36 | 25.79 | 11.85 | 13.70 | 36.75 | 40.17 |
| **Gemini-Pro** | 25.20 | 22.49 | 16.55 | 20.21 | 33.33 | 29.12 | 19.22 | 19.10 | 19.38 | 18.15 | 24.54 | 24.91 |
| QWEN-VL-MAX | | | | | | | | | | | | |
| GPT4V | 21.69 | 19.66 | 12.79 | 11.19 | 25.90 | 42.26 | 17.40 | 20.32 | 15.43 | 7.28 | 37.36 | 43.10 |
| GPT4O | 17.85 | 22.26 | 8.22 | 23.40 | 33.21 | 55.27 | 19.34 | 33.70 | 14.44 | 26.05 | 41.27 | 49.08 |

Table 20: *Scene Graph*

| | what attribute | | what object | | what relation | |
|---|---|---|---|---|---|---|
| | **DP** | **SP** | **DP** | **SP** | **DP** | **SP** |
| **Human** | 96.00 | | 99.00 | | 97.00 | |
| INSTRUCTBLIP-7B | 40.21 | 0.12 | 31.09 | 0.26 | 43.21 | 0.60 |
| INSTRUCTBLIP-13B | 43.06 | 0.47 | 37.55 | 0.53 | 45.12 | 2.26 |
| QWEN-VL | 38.91 | 2.25 | 42.03 | 3.95 | 31.90 | 25.00 |
| QWEN-VL-CHAT | 46.38 | 48.04 | 43.48 | 48.22 | 47.02 | 50.71 |
| LLAVA-7B | 44.72 | 35.23 | 50.72 | 44.53 | 26.79 | 33.45 |
| LLAVA-13B | 40.21 | 33.21 | 52.17 | 45.85 | 31.55 | 31.19 |
| **InternVL-Chat-24B** | 26.81 | 25.74 | 44.66 | 34.26 | 44.05 | 47.26 |
| **LLaVA-Next-34B** | 27.40 | 29.30 | 55.73 | 54.68 | 46.19 | 45.60 |
| **Gemini-Pro** | 27.88 | 27.28 | 26.88 | 26.35 | 36.90 | 42.86 |
| QWEN-VL-MAX | | | | | | |
| GPT4V | 34.40 | 32.03 | 34.78 | 30.17 | 50.60 | 54.17 |
| GPT4O 28.94 | 30.25 | 37.42 | 37.68 | 47.86 | 56.79 | |

## G.3 A breakdown of Table 16

Table 21: *3D tabletop scene*

| | what attribute move | | what attribute rotate | | what move | | what rotate | | where move | | where rotate | |
|---|---|---|---|---|---|---|---|---|---|---|---|---|
| | DP | SP | DP | SP | DP | SP | DP | SP | DP | SP | DP | SP |
| VIDEO-CHATGPT-7B | 23.03 | 20.21 | 20.80 | 19.98 | 8.87 | 7.07 | 12.28 | 13.33 | 19.29 | 19.17 | 12.31 | 11.47 |
| VIDEO-LLAVA-7B | 26.10 | 36.01 | 22.81 | 24.47 | 23.86 | 11.75 | 23.86 | 16.14 | 27.84 | 20.15 | 26.64 | 25.57 |
| VIDEOCHAT2-7B | 29.72 | 30.39 | 29.67 | 25.18 | 16.07 | 12.71 | 15.20 | 10.76 | 18.56 | 21.00 | 28.79 | 21.15 |
| VIDEO-LLAMA-2-7B | 22.89 | 21.02 | 24.11 | 17.73 | 17.87 | 11.63 | 18.95 | 12.75 | 21.00 | 10.50 | 22.70 | 9.56 |
| VIDEO-LLAMA-2-13B | 26.91 | 21.15 | 25.30 | 17.73 | 14.39 | 13.55 | 16.61 | 15.56 | 20.63 | 16.36 | 19.12 | 17.32 |
| CHAT-UNIVI-7B | 28.65 | 26.24 | 23.76 | 21.28 | 14.51 | 11.51 | 14.39 | 12.40 | 20.63 | 21.98 | 27.48 | 20.55 |
| CHAT-UNIVI-13B | 26.77 | 24.63 | 24.70 | 20.45 | 16.55 | 12.35 | 18.25 | 12.51 | 19.66 | 19.05 | 24.73 | 20.31 |
| INTERNVL-CHAT-1.5-24B | 34.00 | 32.93 | 40.31 | 30.73 | 10.19 | 8.75 | 11.93 | 9.36 | 12.94 | 16.61 | 23.06 | 23.18 |
| LLAVA-NEXT-34B | 31.33 | 33.33 | 35.82 | 36.05 | 11.39 | 12.11 | 13.68 | 15.09 | 10.38 | 10.26 | 12.07 | 13.14 |
| GEMINI-PRO | 31.46 | 34.14 | 26.83 | 24.70 | 8.03 | 11.51 | 13.33 | 11.93 | 19.78 | 18.93 | 22.82 | 20.67 |
| QWEN-VL-MAX | | | | | | | | | | | | |
| GPT4V | 27.58 | 33.60 | 27.42 | 36.41 | 11.99 | 12.11 | 14.39 | 12.75 | 22.34 | 20.15 | 16.73 | 16.13 |
| GPT4O | 30.39 | 33.33 | 33.22 | 47.99 | 9.47 | 7.55 | 8.65 | 6.32 | 22.83 | 18.32 | 16.97 | 16.85 |

Table 22: *Scene Graph*

| | what action | | what object | | what relation | |
|---|---|---|---|---|---|---|
| | DP | SP | DP | SP | DP | SP |
| VIDEO-CHATGPT-7B | 6.83 | 9.31 | 10.09 | 10.46 | 30.59 | 27.63 |
| VIDEO-LLAVA-7B | 28.27 | 11.19 | 24.23 | 22.63 | 34.59 | 29.11 |
| VIDEOCHAT2-7B | 16.49 | 11.07 | 24.85 | 14.39 | 19.52 | 14.95 |
| VIDEO-LLAMA-2-7B | 18.26 | 9.66 | 19.93 | 10.33 | 28.20 | 23.52 |
| VIDEO-LLAMA-2-13B | 16.37 | 12.84 | 18.94 | 15.99 | 22.95 | 14.16 |
| CHAT-UNIVI-7B | 23.32 | 18.49 | 20.42 | 12.30 | 30.48 | 25.68 |
| CHAT-UNIVI-13B | 22.73 | 17.31 | 25.22 | 15.50 | 25.68 | 24.32 |
| INTERNVL-CHAT-1.5-24B | 16.49 | 11.90 | 11.93 | 6.77 | 24.32 | 23.29 |
| LLAVA-NEXT-34B | 12.25 | 14.02 | 14.02 | 14.15 | 23.40 | 26.03 |
| GEMINI-PRO | 11.54 | 12.25 | 22.39 | 18.08 | 18.15 | 17.58 |
| QWEN-VL-MAX | | | | | | |
| GPT4V | 10.95 | 10.25 | 15.87 | 14.15 | 20.32 | 18.61 |
| GPT4O | 9.89 | 12.37 | 13.41 | 15.50 | 18.72 | 14.27 |

# H  Details of Experiments on Query Results Approximation Algorithms

To experiment with different query results approximation approaches, we first conduct extensive experiments to evaluate a set of representative models against a subset of tasks for each task generator. Then, we build an Oracle database with the obtained evaluation results, referred to as TASK-ME-ANYTHING-DB, and study different query results approximation methods with this Oracle database to verify their effectiveness. We will release the TASK-ME-ANYTHING-DB for future studies of query results approximation or model performance prediction.

## H.1  Experiment details

**Setup.**  For image question answering tasks, We select 6 representative open-sourced large multi-modal language models (MLMs) from 3 model families: INSTRUCTBLIP-7B and INSTRUCTBLIP-13B from INSTRUCTBLIP [18], QWEN-VL and QWEN-VL-CHAT from QWEN-VL [6], and LLAVA-7B and LLAVA-13B from LLAVA [58]. For video question answering tasks, We select 7 representative open-sourced Large Video Language Models from 5 model families: VIDEO-LLAMA-2-7B and VIDEO-LLAMA-2-13B from VIDEO-LLAMA-2 [100], VIDEO-CHATGPT-7B from VIDEO-CHATGPT [65], CHAT-UNIVI-7B and CHAT-UNIVI-13B from CHAT-UNIVI [42], VIDEO-LLAVA-7B from VIDEO-LLAVA [55], and VIDEOCHAT2-7B from VIDEOCHAT2 [53]. We evaluate the models against a subset of tasks whose statistics can be found in Table 23. Since we generate 15 task instances for each task and involve multiple models, these lead to a total number of 24,240,780 <model, task instance> pairs in evaluation. We evaluate the query results approximation methods on a series of query instances for each type of query. These query instances cover all the subsets of tasks and models we evaluate, leading to a set of 1137 query instances in total (741 for ImageQA and 396 for VideoQA). We set the budget to 2,000 task evaluations.

Table 23: **Statistics of evaluated tasks.** For each task, we generate 15 task instances for evaluation.

|  | Scenerio | Task generator | # of tasks |
|---|---|---|---|
| **ImageQA** | *2D sticker image* | how many | 17,238 |
|  |  | what | 12,740 |
|  |  | where | 12,740 |
|  |  | what attribute | 12,740 |
|  |  | where attribute | 12,740 |
|  | *3D tabletop scene* | how many | 17,238 |
|  |  | what | 12,740 |
|  |  | where | 12,740 |
|  |  | what attribute | 12,740 |
|  |  | where attribute | 12,740 |
|  |  | what size | 10,304 |
|  |  | what attribute size | 7,840 |
|  |  | where size | 10,304 |
|  |  | what distance | 6,160 |
|  |  | what attribute distance | 6,000 |
|  |  | where distance | 6,160 |
|  | **real image w** *Scene Graph* | what object | 10,000 |
|  |  | what attribute | 10,000 |
|  |  | what relation | 10,000 |
| | **Total number of tasks: 144,966** | | |
| **VideoQA** | *3D tabletop scene* | what rotate video | 2,464 |
|  |  | what attribute rotate video | 7,840 |
|  |  | where rotate video | 2,464 |
|  |  | what distance video | 4,928 |
|  |  | what attribute distance video | 15,680 |
|  |  | where distance video | 4,928 |
|  | **Real video w** *Scene Graph* | what object video | 10,000 |
|  |  | what action video | 10,000 |
|  |  | what relation video | 10,000 |
| | **Total number of tasks: 106,608** | | |

**Evaluation metrics.** To evaluate the query results approximation methods, we adopt different evaluation metrics for different types of queries. For Top-K queries, we report the Mean Rank and the Hit Rate: Mean Rank is the average of the ground truth rank of the K items returned by the query results approximation method, so a lower Mean Rank indicates the returned items are actually ranked higher and the query results approximation method is better; Hit Rate measures the percentage of the K returned items are actual Top-K items, so the higher is the better. For the Threshold query and its variants (Model Comparison and Model Debugging query), we can treat them as a binary classification problem and adopt the Prediction, Recall, and F1-score as evaluation metrics.

## H.2 Experiments on approximations under different budgets.

To evaluate the performance of approximation algorithms under different budgets, we conducted an experiment using QWEN-VL-CHAT as the target model on 2D how-many tasks. We tested three query approximation algorithms on four types of queries: Top-K query, Threshold query, Model comparison query, and Model debugging query. The experiments were performed under budgets of $1,000$, $2,000$, and $3,000$. The results of the experiment can be found in Table 24, 25, 26, and 27.

The results demonstrate that the *Active* approximation algorithm consistently outperforms the *Random* and *Fitting* algorithms across all query types and budget levels. In particular, for the Model Compare query, *Active* achieves better results with a 2,000 budget than baselines with larger budgets. Also, we can see the performance increase rapidly with more budget, indicating that users could have more accurate results when using a larger budget

Table 24: The performance of Top-K query results approximation algorithms. Mean Rank (MR, lower is better) and Hit Rate (HR, higher is better) are the metrics.

| Budget | Random | | Fitting | | Active | |
|---|---|---|---|---|---|---|
| | MR | HR (%) | MR | HR (%) | MR | HR (%) |
| 1,000 | 137.1 | 0.0 | 143.3 | 10.0 | 44.3 | 20.0 |
| 2,000 | 116.6 | 0.0 | 121.8 | 0.0 | 32.2 | 20.0 |
| 3,000 | 110.3 | 10.0 | 121.4 | 10.0 | 21.4 | 20.0 |

Table 25: The performance of Threshold query results approximation algorithms. Precision (P), Recall (R), and F1-score (F1) are the metrics.

| Budget | Random | | | Fitting | | | Active | | |
|---|---|---|---|---|---|---|---|---|---|
| | P (%) | R (%) | F1 (%) | P (%) | R (%) | F1 (%) | P (%) | R (%) | F1 (%) |
| 1,000 | 42.61 | 31.82 | 36.43 | 48.48 | 10.39 | 17.11 | 45.0 | 11.69 | 18.56 |
| 2,000 | 43.90 | 35.06 | 38.99 | 43.44 | 34.42 | 38.41 | 43.44 | 34.42 | 38.41 |
| 3,000 | 45.38 | 38.31 | 41.55 | 45.89 | 43.51 | 44.67 | 50.93 | 71.43 | 59.46 |

Table 26: The performance of Model comparison query results approximation algorithms. Precision (P), Recall (R), and F1-score (F1) are the metrics.

| Budget | Random | | | Fitting | | | Active | | |
|---|---|---|---|---|---|---|---|---|---|
| | P (%) | R (%) | F1 (%) | P (%) | R (%) | F1 (%) | P (%) | R (%) | F1 (%) |
| 1,000 | 100.0 | 5.86 | 11.08 | 88.34 | 6.73 | 12.51 | 61.22 | 28.71 | 39.09 |
| 2,000 | 100.0 | 11.37 | 20.42 | 62.88 | 31.82 | 42.26 | 75.18 | 41.44 | 53.43 |
| 3,000 | 100.0 | 17.41 | 29.66 | 69.74 | 43.19 | 53.35 | 82.81 | 52.30 | 64.11 |

Table 27: The performance of Model debugging query results approximation algorithms. Precision (P), Recall (R), and F1-score (F1) are the metrics.

| Budget | Random | | | Fitting | | | Active | | |
|---|---|---|---|---|---|---|---|---|---|
| | P (%) | R (%) | F1 (%) | P (%) | R (%) | F1 (%) | P (%) | R (%) | F1 (%) |
| 1,000 | 100.0 | 6.34 | 11.92 | 100.0 | 6.34 | 11.92 | 100.0 | 6.93 | 12.96 |
| 2,000 | 100.0 | 13.50 | 23.79 | 97.18 | 13.58 | 23.83 | 100.0 | 15.0 | 26.09 |
| 3,000 | 100.0 | 18.82 | 31.68 | 95.29 | 19.13 | 31.87 | 100.0 | 22.01 | 36.08 |

## H.3 Query results approximation experiments in ImageQA

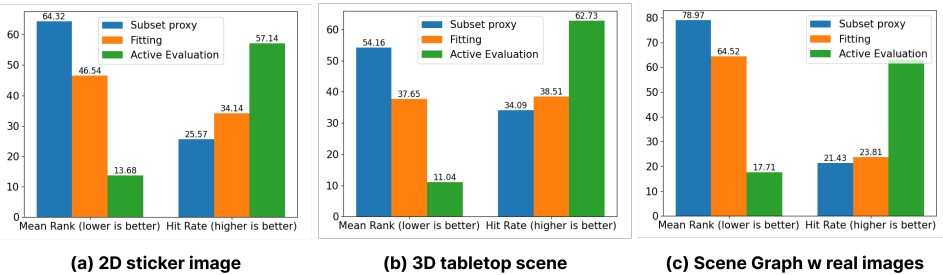

Figure 15: **Top-K Query.** These three bar graphs display the performance of three query approximation methods in Top-K Query, measured by Mean Rank and Hit Rate.

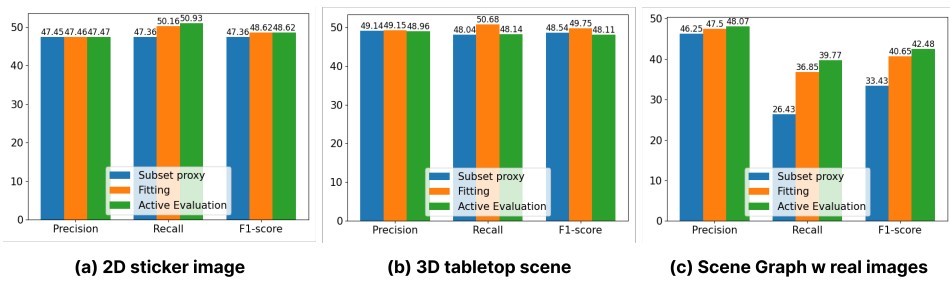

Figure 16: **Threshold Query.** These three bar graphs display the performance of three query approximation methods in Threshold Query, measured by Precision, Recall, and F1-score.

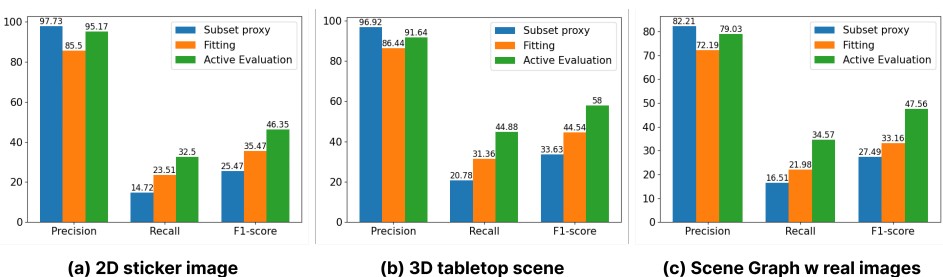

Figure 17: **Model Debugging Query.** These three bar graphs display the performance of three query approximation methods in Model Debugging Query, measured by Precision, Recall, and F1-score.

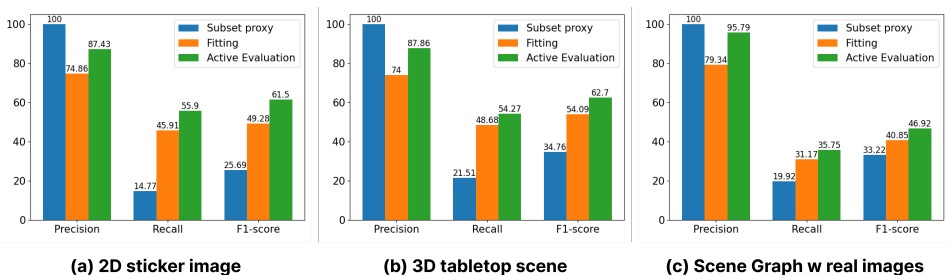

Figure 18: **Model Comparison Query.** These three bar graphs display the performance of three query approximation methods in Model Comparison Query, measured by Precision, Recall, and F1-score.

## H.4 Query results approximation experiments in VideoQA

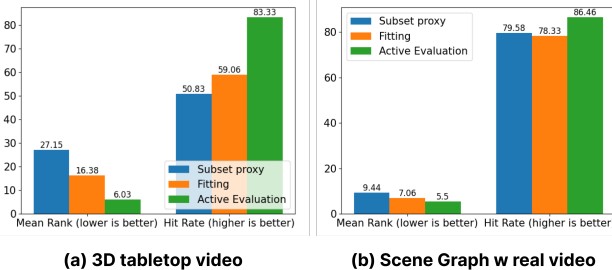

**(a) 3D tabletop video**     **(b) Scene Graph w real video**

Figure 19: **Top-K Query in VideoQA.** These three bar graphs display the performance of three query approximation methods in Top-K Query, measured by Mean Rank and Hit Rate.

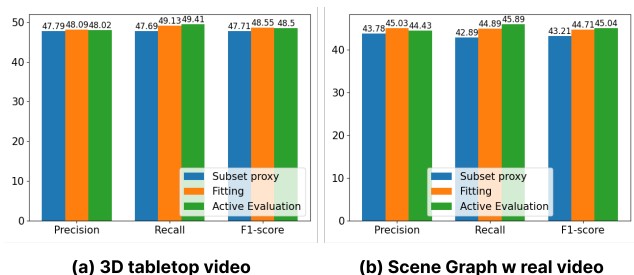

**(a) 3D tabletop video**     **(b) Scene Graph w real video**

Figure 20: **Threshold Query in VideoQA.** These three bar graphs display the performance of three query approximation methods in Threshold Query, measured by Precision, Recall, and F1-score.

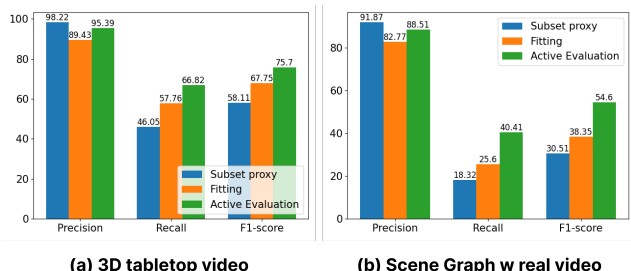

**(a) 3D tabletop video**     **(b) Scene Graph w real video**

Figure 21: **Model Debugging Query in VideoQA.** These three bar graphs display the performance of three query approximation methods in Model Debugging Query, measured by Precision, Recall, and F1-score.

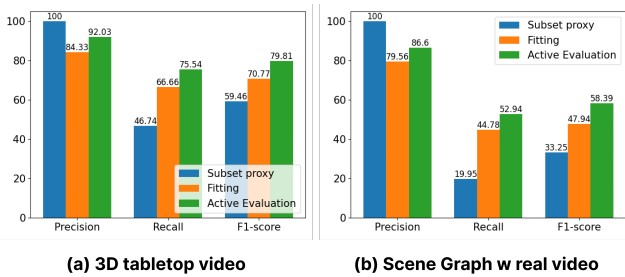

**(a) 3D tabletop video**     **(b) Scene Graph w real video**

Figure 22: **Model Comparison Query in VideoQA.** These three bar graphs display the performance of three query approximation methods in Model Comparison Query, measured by Precision, Recall, and F1-score.

# I Details of Analysis and Case Study

## I.1 What task metadata are models good or bad at?

To obtain a more finegrained understanding of models' skill sets, we also leverage our interface to examine the top and bottom task metadata related to models' best and worst skills. For example, as QWEN-VL-CHAT performs the best on relation understanding across models and skills, we identify the top 20 relations where QWEN-VL-CHAT achieves the highest accuracies (Figure 23) and find that they are mostly actions. Similarly, on VideoQA tasks related to attribute understanding, we are also able to find the attribute values VIDEOCHAT2-7B is the best at and learn that they are mostly associated with color instead of shape or material (Figure 24). On the other hand, we learn that INSTRUCTBLIP-13B does terribly on spatial understanding especially when the object's absolute position is in the back, followed by front right or left (Figure 25); and among the actions VIDEO-LLAMA-2-13B performs the worst on, most involve "putting" or "throwing" something (Figure 26).

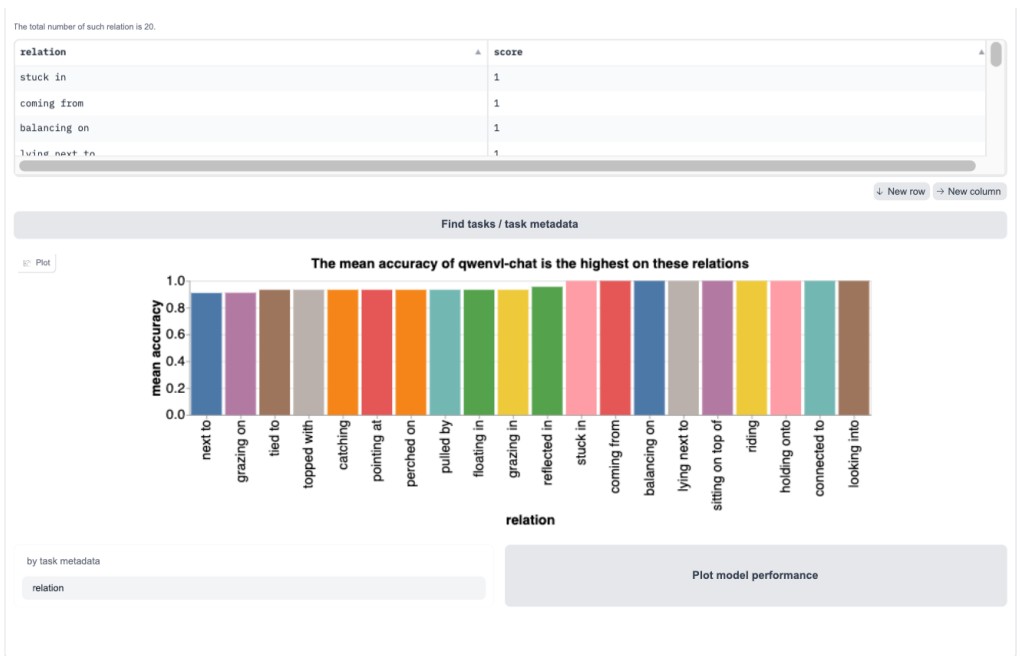

Figure 23: ImageQA: Best relations

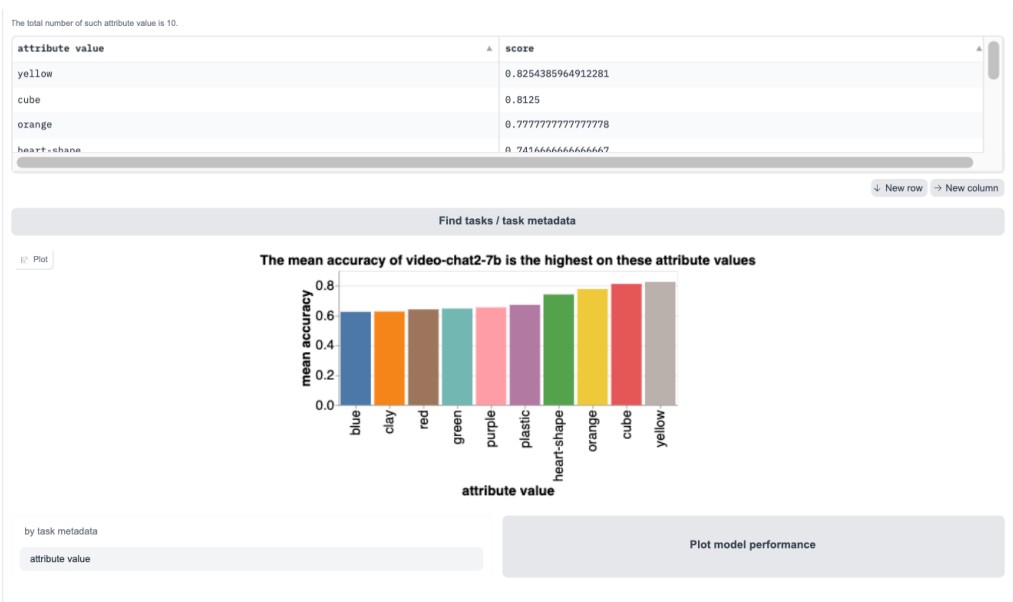

Figure 24: VideoQA: Best attributes

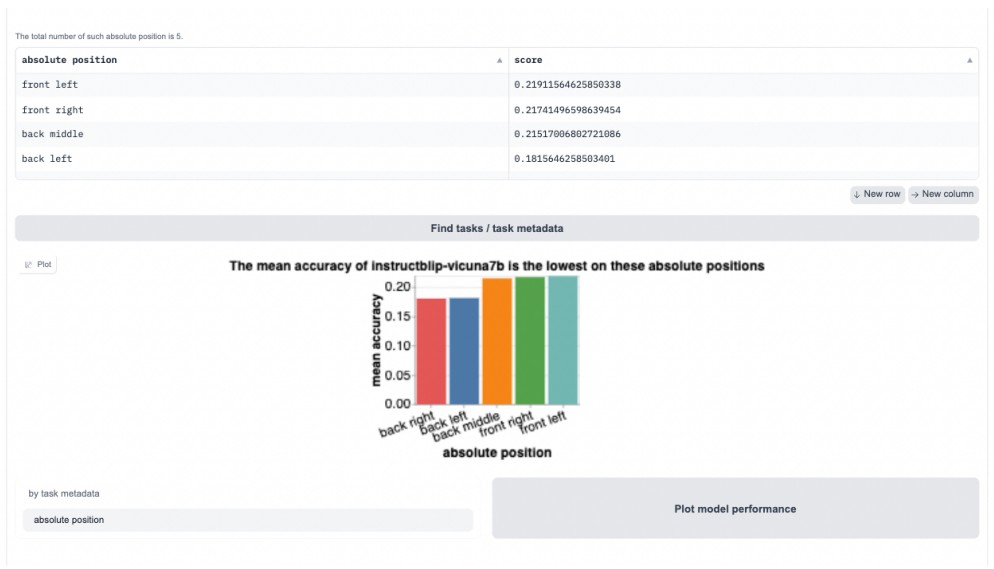

Figure 25: ImageQA: Worst positions

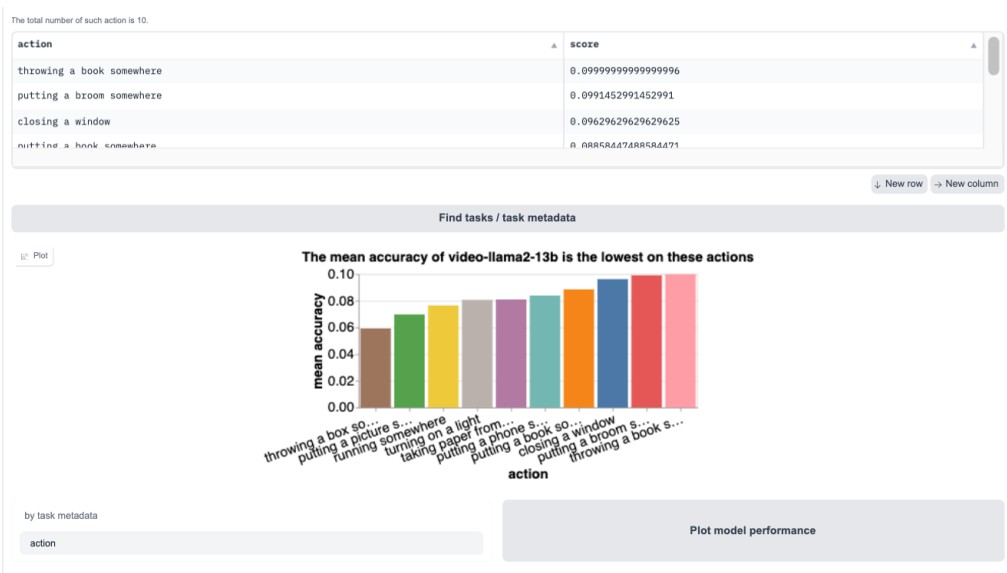

Figure 26: VideoQA: Worst actions

## I.2    Are models' strengths and weaknesses consistent across visual inputs?

Further, we are curious if the models' strong and weak skills are consistent across visual inputs. To this end, we look at models' performance across visual inputs for object, attribute, spatial understanding, and counting as these skills involve tasks in multiple visual inputs such as 2D and 3D. We find that for the same skill, the rankings of models remain largely consistent across visual inputs (Figure 27). We observe strong correlations (with Spearman coefficients of 0.77-0.94) between models' accuracy scores for different visual inputs in the same skill with only one exception: the video models' performance on object understanding in 3D tabletop tasks is only weakly correlated (coefficient = 0.64) with their performance in scene graph tasks. This finding suggests our definition of skills is orthogonal to visual inputs and enables us to find models' inherent strengths and weaknesses.

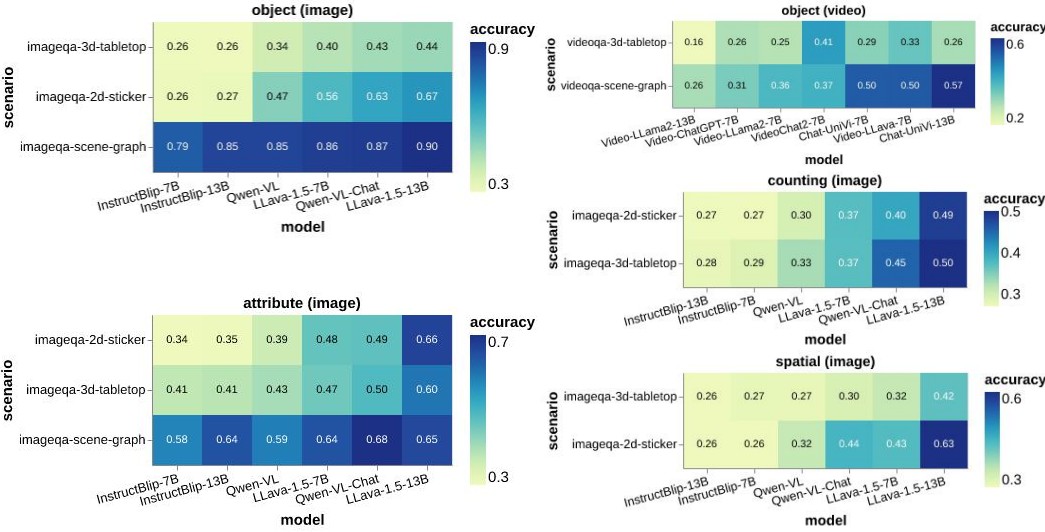

Figure 27: We present models' performance for each skill across visual inputs.

### I.3 How do small models compare against large models? (continued)

As discussed in the main paper, we observe that large multi-modal models collectively perform better than smaller models on ImageQA tasks (Figure 28). Nevertheless, this finding might not always hold for individual models. Through t-tests with pairs of small and large models from the same source, we find one exception: INSTRUCTBLIP-7B ($\mu = 0.63$) significantly outperforms INSTRUCTBLIP-13B ($\mu = 0.49$) on relation understanding (with p-value = 0) (Figure 28).

Further, upon a closer look with our interface, we identify a few relations where INSTRUCTBLIP-7B outperforms INSTRUCTBLIP-13B by a large margin e.g. 50% (Figure 28). Similarly, we also retrieve a few actions and objects where VIDEO-LLAMA-2-7B performs much better e.g. by 20% than VIDEO-LLAMA-2-13B (Figures 29 and 30).

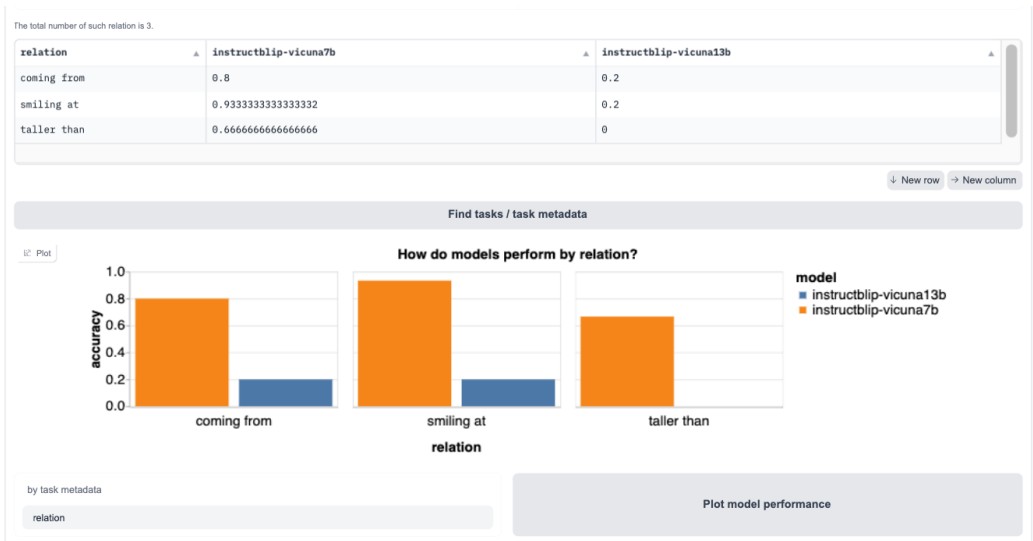

Figure 28: INSTRUCTBLIP-7B vs. INSTRUCTBLIP-13B relations

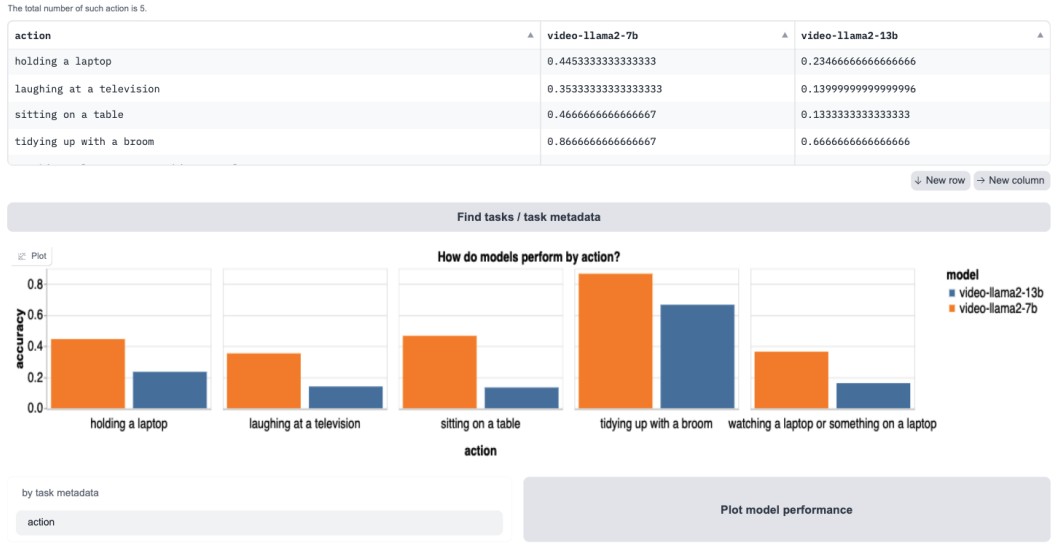

Figure 29: VIDEO-LLAMA-2-7B vs. VIDEO-LLAMA-2-13B actions

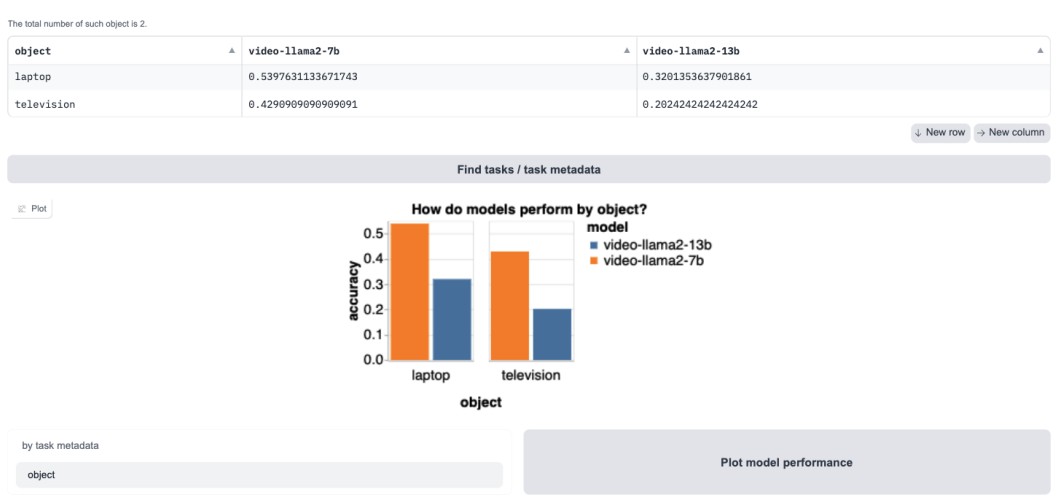

Figure 30: VIDEO-LLAMA-2-7B vs. VIDEO-LLAMA-2-13B objects

## I.4 A case study for synthetic data vs. real data in TASK-ME-ANYTHING

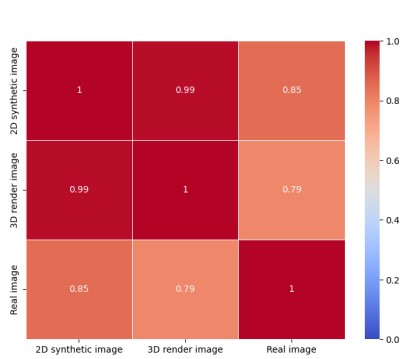

Figure 31: Correlations of synthetic data with real data in TASK-ME-ANYTHING.

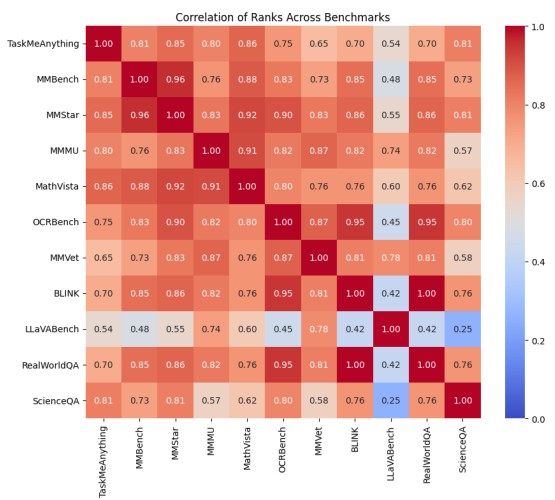

Figure 32: Correlations of TASK-ME-ANYTHING with other popular MLM benchmarks.

TASK-ME-ANYTHING leverages both synthetic and real images/videos for evaluating MLM models. While synthetic data are higly controllable, low-cost, its effectiveness compared to real data remains unclear, particularly given that most benchmarks rely heavily on real datasets. To quantitatively justify the transferability between evaluations on synthetic and real-world images, we compared 18 models' rankings on synthetic vs. realistic images and obsered strong positive correlations between them. Across the three different image sources in Task-Me-Anything, we found that the correlation between models' rankings on 2D and 3D synthetic images is the strongest (r=0.99). While the correlations between models' rankings on real and 2D images, and between real and 3D images are slightly smaller (r=0.85 and 0.79 respectively), these numbers still suggest strong positive correlations between models' performance on synthetic images and on real images (Figure 31).

## I.5 Do TASK-ME-ANYTHING yield results similar to existing benchmarks?

**Comparison with existing benchmarks.** According to Figure 32, we found that there is a strong positive correlation between models' rankings on TaskMeAnything and on other benchmarks, with strong correlations (>=0.8) on the most commonly used ones such as MMMU and MMBench. Notably, the average correlation between TaskMeAnything and the other benchmarks is 0.77, which is greater than that of some other benchmarks such as ScienceQA and LLaVABench (0.70 and 0.57). These results suggest that the evaluation results on our benchmark align with those on existing benchmarks.

**Fine-grained analysis: TallyQA vs. TASK-ME-ANYTHING-RANDOM in MLMs' counting evaluations.** We also conducted a more fine-grained case study testing six open-source models on both the well-known TallyQA Counting benchmark [2] (we selected 10,000 simple questions and 10,000 complex from the whole set) and 2D how-many and 3D how-many tasks in TASK-ME-ANYTHING-RANDOM. (Table 28), the results demonstrate a notable correlation. For instance, the LLAVA-13B is the best-performing model in both TallyQA and how-many tasks in TASK-ME-ANYTHING-RANDOM. The Spearman ranking coefficient for the correlation between the 2D how-many tasks and TallyQA is 0.714 (p-value = 0.111), while for the 3D how-many tasks, it is 0.543 (p-value = 0.266). These results indicate positive correlations of model performance between our tasks and existing ones, validating that TASK-ME-ANYTHING can effectively reflect model performance in a manner similar to existing benchmark.

Table 28: Models performance on TallyQA Counting benchmark and 2D how-many and 3D how-many in our TASK-ME-ANYTHING-RANDOM

| Model | TallyQA | 2D How Many | 3D How Many |
|---|---|---|---|
| LLAVA-7B | 35.90 | 42.00 | 38.67 |
| LLAVA-13B | **38.33** | **49.33** | **46.67** |
| QWEN-VL | 18.79 | 30.67 | 32.33 |
| QWEN-VL-CHAT | 32.07 | 39.67 | 45.00 |
| INSTRUCTBLIP-7B | 29.92 | 23.67 | 32.67 |
| INSTRUCTBLIP-13B | 33.22 | 26.67 | 32.00 |

## I.6 Results of Query 6

Table 29: Answering Q1-Q3 with Top-K query regarding individual objects/relations/attributes. We also present the GPT4O performance drop ($\Delta$ Perf. (%)) on task instances involving found task elements as ground truth answers compared to random task instances, and show that performance drops by a large margin.

| Question | Task generator | Top-K objects/relations/attributes | $\Delta$ Perf. (%) |
|---|---|---|---|
| what objects are GPT4O bad at recognizing when rotating/moving? | VideoQA 3D what rotate | *fermentation product, hamper, tool, computer keyboard, mathematical instrument* | -21.67 |
| | VideoQA 3D what move | *towel, bathtub, furniture, air conditioner, desk* | -19.33 |
| what relations are GPT4O bad at understanding? | ImageQA SG what relation | *taller than, exiting, pushing, pushed by, between* | -51.05 |
| | VideoQA SG what relation | *beneath, covered by, carrying, above, standing on* | -16.66 |
| what attributes are GPT4O bad at recognizing? | ImageQA 2D what attribute | *purple, brown, red, gray, beige* | -5.33 |
| | ImageQA 3D what attribute | *stone, rubber, textile, leather, plastic* | -10.67 |
| | ImageQA SG what attribute | *crooked, power, lower, steep, glowing* | -45.45 |

## J  Datasheet for TASK-ME-ANYTHING-RANDOM

### J.1  Motivation

1. **For what purpose was the dataset created?**
   TASK-ME-ANYTHING-RANDOM is created as a randomly selected subset of TASK-ME-ANYTHING-V1.0 to provide an overview of TASK-ME-ANYTHING.

2. **Who created the dataset and on behalf of which entity?**
   It was created by the authors of this paper.

3. **Who funded the creation of the dataset?**
   The creation of the dataset was funded by the institute to which the authors belong.

### J.2  Composition

1. **What do the instances that comprise the dataset represent (e.g., documents, photos, people, countries?)**
   The dataset consists of 2D and 3D synthetic images, videos, and real images and videos, each accompanied by corresponding task plans, questions, options, and ground truths.

2. **How many instances are there in total (of each type, if appropriate)?**
   ImageQA: 5,700 instances (19 types of task generators, each with 300 instances per split per generator type). VideoQA: 2,700 instances (9 types of task generators, each with 300 instances per split per generator type).

3. **Does the dataset contain all possible instances, or is it a sample of instances from a larger set?**
   This dataset is a randomly selected subset from the TASK-ME-ANYTHING-V1.0 task space. Additional tasks can be generated by users based on their needs.

4. **Is there a label or target associated with each instance?**
   Yes, each instance includes both input and targets.

5. **Is any information missing from individual instances?**
   No.

6. **Are there recommended data splits (e.g., training, development/validation, testing)?**
   For ImageQA, there are 19 splits, each containing 300 instances from a specific task type. For VideoQA, there are 9 splits, each also containing 300 instances from a specific task type.

7. **Are there any errors, sources of noise, or redundancies in the dataset?**
   For real images and videos, the scene graphs may contain a small amount of noise due to human annotation bias. However, this does not have a significant impact on the research.

8. **Is the dataset self-contained, or does it link to or otherwise rely on external resources (e.g., websites, tweets, other datasets)?**
   The 3D objects used in the 2D sticker and 3D table scenarios are sourced from Objaverse. The real image scenarios are derived from the GQA versions of Visual Genome (VG), and the real videos are obtained from AGQA.

9. **Does the dataset contain data that might be considered confidential?**
   No.

10. **Does the dataset contain data that, if viewed directly, might be offensive, insulting, threatening, or might otherwise cause anxiety?**
    No.

## K  Datasheet for TASK-ME-ANYTHING-2024

### K.1  Motivation

1. **For what purpose was the dataset created?**
   TASK-ME-ANYTHING-2024 is created as an automatically selected subset of TASK-ME-ANYTHING-V1.0 that contains the tasks that current MLMs are still struggling with.

2. **Who created the dataset and on behalf of which entity?**
   It was created by the authors of this paper.

3. **Who funded the creation of the dataset?**
The creation of the dataset was funded by the institute to which the authors belong.

## K.2 Composition

1. **What do the instances that comprise the dataset represent (e.g., documents, photos, people, countries?)**
The dataset consists of 2D and 3D synthetic images, videos, and real images and videos, each accompanied by corresponding task plans, questions, options, and ground truths.

2. **How many instances are there in total (of each type, if appropriate)?**
ImageQA: 12,270 instances (19 types of task generators). VideoQA: 3,567 instances (9 types of task generators).

3. **Does the dataset contain all possible instances, or is it a sample of instances from a larger set?**
This dataset is for reflecting the current progress of MLMs by automatically finding tasks that popular MLMs struggle with using the TaskMeAnything Top-K query and query approximation algorithms.

4. **Is there a label or target associated with each instance?**
Yes, each instance includes both input and targets.

5. **Is any information missing from individual instances?**
No.

6. **Are there recommended data splits (e.g., training, development/validation, testing)?**
For ImageQA, there are 19 splits, each containing instances from a specific task type. For VideoQA, there are 9 splits, each also containing instances from a specific task type.

7. **Are there any errors, sources of noise, or redundancies in the dataset?**
For real images and videos, the scene graphs may contain a small amount of noise due to human annotation bias. However, this does not have a significant impact on the research.

8. **Is the dataset self-contained, or does it link to or otherwise rely on external resources (e.g., websites, tweets, other datasets)?**
The 3D objects used in the 2D sticker and 3D table scenarios are sourced from Objaverse. The real image scenarios are derived from the GQA versions of Visual Genome (VG), and the real videos are obtained from AGQA.

9. **Does the dataset contain data that might be considered confidential?**
No.

10. **Does the dataset contain data that, if viewed directly, might be offensive, insulting, threatening, or might otherwise cause anxiety?**
No.

## K.3 Collection Process

1. **How was the data associated with each instance acquired?**
The 3D objects used in the 2D sticker and 3D table scenarios are sourced from Objaverse. The real image scenarios are derived from the GQA versions of VG, while the real videos are from AGQAs. References are provided in Section 3 of the main text.

2. **What mechanisms or procedures were used to collect the data (e.g., hardware apparatus or sensor, manual human curation, software program, software API)?**
We used multiple NVIDIA A6000 and A100 GPUs to run Blender for rendering the synthetic scenes. Questions, options, and ground truth were generated by task generators (Python code).

3. **Who was involved in the data collection process (e.g., students, crowdworkers, contractors) and how were they compensated (e.g., how much were crowdworkers paid)?**
The authors of this paper were directly involved in the data collection process, annotating the attributes of 3d objects and build the taxonomy themselves.

4. **Over what timeframe was the data collected?**
The final version of the dataset was generated in August, 2024.

### K.4 Uses

1. **Has the dataset been used for any tasks already?**
   No, this dataset has not been used for any tasks yet.

2. **What (other) tasks could the dataset be used for?**
   This data can also be used in various computer vision tasks, such as localization, object detection, etc.

3. **Is there anything about the composition of the dataset or the way it was collected and preprocessed/cleaned/labeled that might impact future uses?**
   No.

4. **Are there tasks for which the dataset should not be used?**
   No.

### K.5 Distribution

1. **Will the dataset be distributed to third parties outside of the entity (e.g., company, institution, organization) on behalf of which the dataset was created?**
   Yes, the dataset is open to the public.

2. **How will the dataset be distributed (e.g., tarball on website, API, GitHub)?**
   You can access our dataset via the links below:
   Dataset (ImageQA): `https://huggingface.co/datasets/weikaih/TaskMeAnything-v1-videoqa-2024`
   Dataset (VideoQA): `https://huggingface.co/datasets/weikaih/TaskMeAnything-v1-videoqa-2024`
   Code: `https://github.com/JieyuZ2/TaskMeAnything`

3. **Have any third parties imposed IP-based or other restrictions on the data associated with the instances?**
   No.

4. **Do any export controls or other regulatory restrictions apply to the dataset or to individual instances?**
   No.

### K.6 Maintenance

1. **Who will be supporting/hosting/maintaining the dataset?**
   The authors of this paper will support, host, and maintain the dataset.

2. **How can the owner/curator/manager of the dataset be contacted (e.g., email address)?**
   The owner/curator/manager(s) of the dataset can be contacted through the following email: Jieyu Zhang (jieyuz2@cs.washington.edu)

3. **Is there an erratum?**
   No. If errors are found in the future, we will release errata on the GitHub repo for the dataset: (https://github.com/JieyuZ2/TaskMeAnything).

4. **Will the dataset be updated (e.g., to correct labeling errors, add new instances, delete instances)?**
   Yes, the datasets will be updated whenever necessary to ensure accuracy, and announcements will be made accordingly. These updates will be posted on the GitHub repo for the dataset: (https://github.com/JieyuZ2/TaskMeAnything).

5. **If the dataset relates to people, are there applicable limits on the retention of the data associated with the instances (e.g., were the individuals in question told that their data would be retained for a fixed period of time and then deleted?)**
   N/A

6. **Will older versions of the dataset continue to be supported/hosted/maintained?**
   Yes. Older versions of the dataset will continue to be maintained and hosted.

7. **If others want to extend/augment/build on/contribute to the dataset, is there a mechanism for them to do so?**
   Yes, one can extend the dataset by simply adding more source data and task generators, or by generating more instances from the existing task space.

# L    Author Statement for TASK-ME-ANYTHING-RANDOM

We, the authors of this dataset, hereby declare that we bear all responsibility in case of any violation of rights or any other legal issues arising from the use of this dataset. We confirm that we have obtained the necessary permissions for the use of any external data incorporated within this dataset and have adhered to all relevant data protection and privacy regulations.

We also confirm that this dataset is licensed under the Apache license 2.0, which permits its use, distribution, and modification under the terms specified in the license. Users of this dataset are required to comply with the conditions set forth in the license. For detailed information on the license, please refer to the dataset repository or contact us at the provided email address: Jieyu Zhang (jieyuz2@cs.washington.edu).

By accessing or using this dataset, you agree to abide by the terms and conditions of the Apache license 2.0 and acknowledge that any misuse or violation of these terms is solely your responsibility.

# M    Task Generator Cards

# WhatGridTaskGenerator

- **Basic Information**.
  - **Task Type**. ImageQA
  - **Question Type**. what object
  - **Answer Type**. object category
  - **Data Type**. 2D sticker image
- **Source Data**.
  - Rendering images of objects from Objaverse.
  - Annotations regarding object category, material, color, visable angles, and shape.
- **Task Plan Schema**.
  - **question type**: `string`. The question type of these tasks will be "what".
  - **grid number**: `integer`. The number of diagonal grids of the image, $N$ indicates there are $N \times N$ grids in the image. Support $\{2, 3\}$.
  - **target category**: `string`. The category name of the target object.
  - **absolute position**: `string`. The absolute position of the target object in the grid. It is a number ranging from 0 to 3 (grid number = 2) or 0 to 8 (grid number = 3).
  - **reference category**: `string`. The category name of the object that is used to reference the target object.
  - **reference position**: `string`. The relative position of the target object from the reference object.
  - **attribute type**: `string`. The type of attributes of the target object, currently include: `color`, `material`, and `shape`.
  - **attribute value**: `string`. The value of the attributes of the target object.
- **Partitions**.
  - **Partition 1**.
    * **Template**
      · **Q**: What is the object in the <absolute pos> part of the image?
      · **A**: <target category>
    * **Example**
      · **Q**: What is the object in the bottom middle part of the image?
      · **A**: folding chair
  - **Partition 2**.
    * **Template**.
      · **Q**: What is the object <reference pos> the <reference category>?
      · **A**: <target category>
    * **Example**
      · **Q**: What is the object to the left of the telephone?
      · **A**: table lamp
- **Recommendations**: This task generator is well-suited for evaluating a model's capability in object recognition, both with and without reference objects, using annotated 2D sticker images from Objaverse.

# WhereGridTaskGenerator

- **Basic Information**.
  - **Task Type**. ImageQA
  - **Question Type**. what object
  - **Answer Type**. object category
  - **Data Type**. 2D sticker image
- **Source Data**.
  - Rendering images of objects from Objaverse.
  - Annotations regarding object category, material, color, visable angles, and shape.
- **Task Plan Schema**.
  - **question type**: `string`. The question type of these tasks will be "what".
  - **grid number**: `integer`. The number of diagonal grids of the image, $N$ indicates there are $N \times N$ grids in the image. Support $\{2, 3\}$.
  - **target category**: `string`. The category name of the target object.
  - **absolute position**: `string`. The absolute position of the target object in the grid. It is a number ranging from 0 to 3 (grid number = 2) or 0 to 8 (grid number = 3).
  - **reference category**: `string`. The category name of the object that is used to reference the target object.
  - **reference position**: `string`. The relative position of the target object from the reference object.
  - **attribute type**: `string`. The type of attributes of the target object, currently include: `color`, `material`, and `shape`.
  - **attribute value**: `string`. The value of the attributes of the target object.
- **Partitions**.
  - **Partition 1**.
    * **Template**
      · **Q**: Where is the <target category> in the image?
      · **A**: <absolute position>
    * **Example**
      · **Q**: Where is the apple in the image?
      · **A**: back left
  - **Partition 2**.
    * **Template**.
      · **Q**: Where is the <target category> with respect to the <reference category>?
      · **A**: <reference position>
    * **Example**
      · **Q**: Where is the vacuum cleaner with respect to the backpack?
      · **A**: left
- **Recommendations**: This task generator is designed to evaluate models on their spatial reasoning and object localization capabilities in images, leveraging annotated 2D sticker images from Objaverse.

# WhatAttributeGridTaskGenerator

- **Basic Information**.
  - **Task Type**. ImageQA
  - **Question Type**. what object
  - **Answer Type**. object category
  - **Data Type**. 2D sticker image
- **Source Data**.
  - Rendering images of objects from Objaverse.
  - Annotations regarding object category, material, color, visable angles, and shape.
- **Task Plan Schema**.
  - **question type**: `string`. The question type of these tasks will be "what attribute".
  - **grid number**: `integer`. The number of diagonal grids of the image, $N$ indicates there are $N \times N$ grids in the image. Support $\{2, 3\}$.
  - **target category**: `string`. The category name of the target object.
  - **absolute position**: `string`. The absolute position of the target object in the grid. It is a number ranging from 0 to 3 (grid number = 2) or 0 to 8 (grid number = 3).
  - **reference category**: `string`. The category name of the object that is used to reference the target object.
  - **reference position**: `string`. The relative position of the target object from the reference object.
  - **attribute type**: `string`. The type of attributes of the target object, currently include: `color`, `material`, and `shape`.
  - **attribute value**: `string`. The value of the attributes of the target object.
- **Partitions**.
  - **Partition 1**.
    * **Template**
      · **Q**: What is the <attribute type> of the object in the <absolute position> part of the image?
      · **A**: <attribute value>
    * **Example**
      · **Q**: What is the material of the object in the middle part of the image?
      · **A**: plastic
  - **Partition 2**.
    * **Template**.
      · **Q**: What is the <attribute type> of the object to the left of the <reference category>?
      · **A**: <attribute value>
    * **Example**
      · **Q**: What is the color of the object to the left of the silverware?
      · **A**: gold
- **Recommendations**: This task generator is ideal for evaluating a model's ability to recognize object attributes, including color, material, and shape, within both absolute and relative spatial contexts using annotated 2D sticker images from Objaverse.

# WhereAttributeGridTaskGenerator

- **Basic Information**.
    - **Task Type**. ImageQA
    - **Question Type**. what object
    - **Answer Type**. object category
    - **Data Type**. 2D sticker image
- **Source Data**.
    - Rendering images of objects from Objaverse.
    - Annotations regarding object category, material, color, visable angles, and shape.
- **Task Plan Schema**.
    - **question type**: `string`. The question type of these tasks will be "where attribute".
    - **grid number**: `integer`. The number of diagonal grids of the image, $N$ indicates there are $N \times N$ grids in the image. Support {2, 3}.
    - **target category**: `string`. The category name of the target object.
    - **absolute position**: `string`. The absolute position of the target object in the grid. It is a number ranging from 0 to 3 (grid number = 2) or 0 to 8 (grid number = 3).
    - **reference category**: `string`. The category name of the object that is used to reference the target object.
    - **reference position**: `string`. The relative position of the target object from the reference object.
    - **attribute type**: `string`. The type of attributes of the target object, currently include: `color`, `material`, and `shape`.
    - **attribute value**: `string`. The value of the attributes of the target object.
- **Partitions**.
    - **Partition 1**.
        * **Template**
            · **Q**: Where is the <attribute value> object in the image?
            · **A**: <absolute position>
        * **Example**
            · **Q**: Where is the white object in the image?
            · **A**: top right
    - **Partition 2**.
        * **Template**.
            · **Q**: Where is the <attribute value> object with respect to the <reference category>?
            · **A**: <absolute position>
        * **Example**
            · **Q**: Where is the gray object with respect to the lollipop?
            · **A**: top
- **Recommendations**: This task generator is effective for evaluating a model's spatial reasoning and object localization capabilities, specifically in identifying the positions of objects based on their attributes, using annotated 2D sticker images from Objaverse.

# HowManyGridTaskGenerator

- **Basic Information**.
  - **Task Type**. ImageQA
  - **Question Type**. what object
  - **Answer Type**. object category
  - **Data Type**. 2D sticker image
- **Source Data**.
  - Rendering images of objects from Objaverse.
  - Annotations regarding object category, material, color, visable angles, and shape.
- **Task Plan Schema**.
  - **question type**: `string`. The question type of these tasks will be "how many".
  - **grid number**: `integer`. The number of diagonal grids of the image, $N$ indicates there are $N \times N$ grids in the image. Support $\{2, 3\}$.
  - **target category**: `string`. The category name of the target object.
  - **count** `integer`. The total number of the target objects in the image.
  - **attribute type**: `string`. The type of attributes of the target object, currently include: `color`, `material`, and `shape`.
  - **attribute value**: `string`. The value of the attributes of the target object.
- **Partitions**.
  - **Partition 1**.
    * **Template**
      · **Q**: How many <attribute value> objects are there in the image?
      · **A**: <count>
    * **Example**
      · **Q**: How many blue objects are there in the image?
      · **A**: 2
  - **Partition 2**.
    * **Template**.
      · **Q**: How many <target category> are there in the image?
      · **A**: <count>
    * **Example**
      · **Q**: How many tables are there in the image?
      · **A**: 4
  - **Partition 3**.
    * **Template**.
      · **Q**: How many <attribute value> <target category> are there in the image?
      · **A**: <count>
    * **Example**
      · **Q**: How many pink beverages are there in the image?
      · **A**: 2
- **Recommendations**: This task generator is optimal for evaluating a model's ability to count objects and assess numerical reasoning, using annotated 2D sticker images from Objaverse to determine quantities based on attributes and categories

# What3DGridTaskGenerator

- **Basic Information**.
  - **Task Type**. ImageQA
  - **Question Type**. what object
  - **Answer Type**. object category
  - **Data Type**. 3D tabletop image
- **Source Data**.
  - Rendering images of objects from Objaverse.
  - Annotations regarding object category, material, color, visable angles, and shape.
- **Task Plan Schema**.
  - **question type**: `string`. The question type of these tasks will be "what".
  - **grid number**: `integer`. The number of diagonal grids of the image, $N$ indicates there are $N \times N$ grids in the image. Support $\{2, 3\}$.
  - **target category**: `string`. The category name of the target object.
  - **absolute position**: `string`. The absolute position of the target object in the grid. It is a number ranging from 0 to 3 (grid number = 2) or 0 to 8 (grid number = 3).
  - **reference category**: `string`. The category name of the object that is used to reference the target object.
  - **reference position**: `string`. The relative position of the target object from the reference object.
  - **attribute type**: `string`. The type of attributes of the target object, currently include: `color`, `material`, and `shape`.
  - **attribute value**: `string`. The value of the attributes of the target object.
- **Partitions**.
  - **Partition 1**.
    * **Template**
      · **Q**: What is the object <absolute pos> part of the image?
      · **A**: <target category>
    * **Example**
      · **Q**: What is the object in the front right part of the image?
      · **A**: scale
  - **Partition 2**.
    * **Template**.
      · **Q**: What is the object <reference pos> the <reference category>?
      · **A**: <target category>
    * **Example**
      · **Q**: What is the object to the right of the mobile computer?
      · **A**: bucket
- **Recommendations**: This task generator is excellent for evaluating a model's capability in recognizing objects in 3D tabletop images, both with and without reference objects, using annotated images from Objaverse.

# Where3DGridTaskGenerator

- **Basic Information**.
  - **Task Type**. ImageQA
  - **Question Type**. what object
  - **Answer Type**. object category
  - **Data Type**. 3D tabletop image
- **Source Data**.
  - Rendering images of objects from Objaverse.
  - Annotations regarding object category, material, color, visable angles, and shape.
- **Task Plan Schema**.
  - **question type**: `string`. The question type of these tasks will be "where".
  - **grid number**: `integer`. The number of diagonal grids of the image, $N$ indicates there are $N \times N$ grids in the image. Support $\{2, 3\}$.
  - **target category**: `string`. The category name of the target object.
  - **absolute position**: `string`. The absolute position of the target object in the grid. It is a number ranging from 0 to 3 (grid number = 2) or 0 to 8 (grid number = 3).
  - **reference category**: `string`. The category name of the object that is used to reference the target object.
  - **reference position**: `string`. The relative position of the target object from the reference object.
  - **attribute type**: `string`. The type of attributes of the target object, currently include: `color`, `material`, and `shape`.
  - **attribute value**: `string`. The value of the attributes of the target object.
- **Partitions**.
  - **Partition 1**.
    * **Template**
      · **Q**: Where is the <target category> in the image?
      · **A**: <absolute position>
    * **Example**
      · **Q**: Where is the vacuum cleaner in the image?
      · **A**: back left
  - **Partition 2**.
    * **Template**.
      · **Q**: Where is the <target category> with respect to the <reference category>?
      · **A**: <reference position>
    * **Example**
      · **Q**: Where is the vacuum cleaner with respect to the wine glass?
      · **A**: left
- **Recommendations**: This task generator is ideal for evaluating a model's spatial reasoning and object localization abilities in 3D tabletop images, using annotated images from Objaverse to determine the positions of objects both in absolute and relative terms.

# WhatAttribute3DGridTaskGenerator

- **Basic Information**.
  - **Task Type**. ImageQA
  - **Question Type**. what object
  - **Answer Type**. object category
  - **Data Type**. 3D tabletop image
- **Source Data**.
  - Rendering images of objects from Objaverse.
  - Annotations regarding object category, material, color, visable angles, and shape.
- **Task Plan Schema**.
  - **question type**: `string`. The question type of these tasks will be "what attribute".
  - **grid number**: `integer`. The number of diagonal grids of the image, $N$ indicates there are $N \times N$ grids in the image. Support $\{2, 3\}$.
  - **target category**: `string`. The category name of the target object.
  - **absolute position**: `string`. The absolute position of the target object in the grid. It is a number ranging from 0 to 3 (grid number = 2) or 0 to 8 (grid number = 3).
  - **reference category**: `string`. The category name of the object that is used to reference the target object.
  - **reference position**: `string`. The relative position of the target object from the reference object.
  - **attribute type**: `string`. The type of attributes of the target object, currently include: `color`, `material`, and `shape`.
  - **attribute value**: `string`. The value of the attributes of the target object.
- **Partitions**.
  - **Partition 1**.
    * **Template**
      · **Q**: What is the <attribute type> of the object in the <absolute position> part of the image?
      · **A**: <attribute value>
    * **Example**
      · **Q**: What is the color of the object in the back left part of the image?
      · **A**: red
  - **Partition 2**.
    * **Template**.
      · **Q**: What is the <attribute type> of the object to the left of the <reference category>?
      · **A**: <attribute value>
    * **Example**
      · **Q**: What is the material of the object behind the plate?
      · **A**: wood
- **Recommendations**: This task generator is suitable for evaluating a model's ability to recognize and attribute specific characteristics, such as color, material, and shape, within 3D tabletop images, using annotated images from Objaverse.

# WhereAttribute3DGridTaskGenerator

- **Basic Information**.
  - **Task Type**. ImageQA
  - **Question Type**. what object
  - **Answer Type**. object category
  - **Data Type**. 3D tabletop image
- **Source Data**.
  - Rendering images of objects from Objaverse.
  - Annotations regarding object category, material, color, visable angles, and shape.
- **Task Plan Schema**.
  - **question type**: `string`. The question type of these tasks will be "where attribute".
  - **grid number**: `integer`. The number of diagonal grids of the image, $N$ indicates there are $N \times N$ grids in the image. Support $\{2, 3\}$.
  - **target category**: `string`. The category name of the target object.
  - **absolute position**: `string`. The absolute position of the target object in the grid. It is a number ranging from 0 to 3 (grid number = 2) or 0 to 8 (grid number = 3).
  - **reference category**: `string`. The category name of the object that is used to reference the target object.
  - **reference position**: `string`. The relative position of the target object from the reference object.
  - **attribute type**: `string`. The type of attributes of the target object, currently include: `color`, `material`, and `shape`.
  - **attribute value**: `string`. The value of the attributes of the target object.
- **Partitions**.
  - **Partition 1**.
    * **Template**
      · **Q**: Where is the <attribute value> object in the image?
      · **A**: <absolute position>
    * **Example**
      · **Q**: Where is the wood object in the image?
      · **A**: front right
  - **Partition 2**.
    * **Template**.
      · **Q**: Where is the <attribute value> object with respect to the <reference category>?
      · **A**: <absolute position>
    * **Example**
      · **Q**: Where is the white object with respect to the trophy?
      · **A**: left
- **Recommendations**: This task generator is designed to assess a model's ability to identify and locate objects based on their attributes within 3D tabletop images, leveraging annotated data from Objaverse for precise spatial reasoning and attribute recognition.

# HowMany3DGridTaskGenerator

- **Basic Information**.
  - **Task Type**. ImageQA
  - **Question Type**. what object
  - **Answer Type**. object category
  - **Data Type**. 3D tabletop image
- **Source Data**.
  - Rendering images of objects from Objaverse.
  - Annotations regarding object category, material, color, visable angles, and shape.
- **Task Plan Schema**.
  - **question type**: `string`. The question type of these tasks will be "how many".
  - **grid number**: `integer`. The number of diagonal grids of the image, $N$ indicates there are $N \times N$ grids in the image. Support $\{2, 3\}$.
  - **target category**: `string`. The category name of the target object.
  - **count** `integer`. The total number of the target objects in the image.
  - **attribute type**: `string`. The type of attributes of the target object, currently include: `color`, `material`, and `shape`.
  - **attribute value**: `string`. The value of the attributes of the target object.
- **Partitions**.
  - **Partition 1**.
    * **Template**
      · **Q**: How many <attribute value> objects are there in the image?
      · **A**: <count>
    * **Example**
      · **Q**: How many blue objects are there in the image?
      · **A**: 6
  - **Partition 2**.
    * **Template**.
      · **Q**: How many <target category> are there in the image?
      · **A**: <count>
    * **Example**
      · **Q**: How many plates are there in the image?
      · **A**: 5
  - **Partition 3**.
    * **Template**.
      · **Q**: How many <attribute value> <target category> are there in the image?
      · **A**: <count>
    * **Example**
      · **Q**: How many black furnitures are there in the image?
      · **A**: 4
- **Recommendations**: This task generator is well-suited for evaluating a model's numerical reasoning and counting abilities within 3D tabletop images, using annotated images from Objaverse to determine quantities based on attributes and categories.

# WhatDistance3DGridTaskGenerator

- **Basic Information**.
  - **Task Type**. ImageQA
  - **Question Type**. what object
  - **Answer Type**. object category
  - **Data Type**. 3D tabletop image
- **Source Data**.
  - Rendering images of objects from Objaverse.
  - Annotations regarding object category, material, color, visable angles, and shape.
- **Task Plan Schema**.
  - **question type**: `string`. The question type of these tasks will be "what distance".
  - **distance type**: `string`. The type of the distance between target object and the reference object, indicates whether it pertains to the "farthest" or "closest" distance.
  - **grid number**: `integer`. The number of diagonal grids of the image, $N$ indicates there are $N \times N$ grids in the image. Support $\{2, 3\}$.
  - **target category**: `string`. The category name of the target object.
  - **absolute position**: `string`. The absolute position of the target object in the grid. It is a number ranging from 0 to 3 (grid number = 2) or 0 to 8 (grid number = 3).
  - **reference category**: `string`. The category name of the object that is used to reference the target object.
  - **reference position**: `string`. The relative position of the target object from the reference object.
  - **attribute type**: `string`. The type of attributes of the target object, currently include: `color`, `material`, and `shape`.
  - **attribute value**: `string`. The value of the attributes of the target object.
- **Partitions**.
  - **Partition 1**.
    * **Template**
      · **Q**: What is the object that is <distance type> from the <reference category>?
      · **A**: <target category>
    * **Example**
      · **Q**: What is the object that is farthest from the optical instrument?
      · **A**: juice
- **Recommendations**: This task generator is ideal for evaluating a model's ability to understand and determine relative distances between objects within 3D tabletop images, using annotated data from Objaverse to identify the closest or farthest objects from a given reference.

# WhereDistance3DGridTaskGenerator

- **Basic Information**.
    - **Task Type**. ImageQA
    - **Question Type**. what object
    - **Answer Type**. object category
    - **Data Type**. 3D tabletop image
- **Source Data**.
    - Rendering images of objects from Objaverse.
    - Annotations regarding object category, material, color, visable angles, and shape.
- **Task Plan Schema**.
    - **question type**: `string`. The question type of these tasks will be "where distance".
    - **distance type**: `string`. The type of the distance between target object and the reference object, indicates whether it pertains to the "farthest" or "closest" distance.
    - **grid number**: `integer`. The number of diagonal grids of the image, $N$ indicates there are $N \times N$ grids in the image. Support $\{2, 3\}$.
    - **target category**: `string`. The category name of the target object.
    - **absolute position**: `string`. The absolute position of the target object in the grid. It is a number ranging from 0 to 3 (grid number = 2) or 0 to 8 (grid number = 3).
    - **reference category**: `string`. The category name of the object that is used to reference the target object.
    - **reference position**: `string`. The relative position of the target object from the reference object.
    - **attribute type**: `string`. The type of attributes of the target object, currently include: `color`, `material`, and `shape`.
    - **attribute value**: `string`. The value of the attributes of the target object.
- **Partitions**.
    - **Partition 1**.
        * **Template**
            · **Q**: Where is the object that is <distance type> from the <reference category> in the image?
            · **A**: <reference position>
        * **Example**
            · **Q**: Where is the object that is farthest from the bread in the image?
            · **A**: middle
- **Recommendations**: This task generator is effective for evaluating a model's ability to identify the spatial positions of objects relative to a given reference, specifically focusing on the closest or farthest distances within 3D tabletop images, using annotated data from Objaverse.

# WhatAttributeDistance3DGridTaskGenerator

- **Basic Information**.
    - **Task Type**. ImageQA
    - **Question Type**. what object
    - **Answer Type**. object category
    - **Data Type**. 3D tabletop image
- **Source Data**.
    - Rendering images of objects from Objaverse.
    - Annotations regarding object category, material, color, visable angles, and shape.
- **Task Plan Schema**.
    - **question type**: `string`. The question type of these tasks will be "what attribute distance".
    - **distance type**: `string`. The type of the distance between target object and the reference object, indicates whether it pertains to the "farthest" or "closest" distance.
    - **grid number**: `integer`. The number of diagonal grids of the image, $N$ indicates there are $N \times N$ grids in the image. Support $\{2, 3\}$.
    - **target category**: `string`. The category name of the target object.
    - **absolute position**: `string`. The absolute position of the target object in the grid. It is a number ranging from 0 to 3 (grid number = 2) or 0 to 8 (grid number = 3).
    - **reference category**: `string`. The category name of the object that is used to reference the target object.
    - **reference position**: `string`. The relative position of the target object from the reference object.
    - **attribute type**: `string`. The type of attributes of the target object, currently include: `color`, `material`, and `shape`.
    - **attribute value**: `string`. The value of the attributes of the target object.
- **Partitions**.
    - **Partition 1**.
        * **Template**
            · **Q**: What is the <attribute type> of the object that is <distance type> to the <target category>?
            · **A**: <attribute value>
        * **Example**
            · **Q**: What is the color of the object that is closest to the statue?
            · **A**: beige
- **Recommendations**: This task generator is well-suited for evaluating a model's ability to determine and attribute specific characteristics, such as color, material, and shape, to objects that are either the closest or farthest from a given reference within 3D tabletop images, using annotated data from Objaverse.

# WhatSize3DGridTaskGenerator

- **Basic Information**.
    - **Task Type**. ImageQA
    - **Question Type**. what object
    - **Answer Type**. object category
    - **Data Type**. 3D tabletop image
- **Source Data**.
    - Rendering images of objects from Objaverse.
    - Annotations regarding object category, material, color, visable angles, and shape.
- **Task Plan Schema**.
    - **question type**: `string`. The question type of these tasks will be "what size".
    - **size**: `string`. The type of the size of the target object, indicates whether it pertains to the "largest" or "smallest" in all the objects.
    - **grid number**: `integer`. The number of diagonal grids of the image, $N$ indicates there are $N \times N$ grids in the image. Support $\{2, 3\}$.
    - **target category**: `string`. The category name of the target object.
    - **absolute position**: `string`. The absolute position of the target object in the grid. It is a number ranging from 0 to 3 (grid number = 2) or 0 to 8 (grid number = 3).
    - **attribute type**: `string`. The type of attributes of the target object, currently include: `color`, `material`, and `shape`.
    - **attribute value**: `string`. The value of the attributes of the target object.
- **Partitions**.
    - **Partition 1**.
        * **Template**
            · **Q**: What is the <size> object in the image?
            · **A**: <target category>
        * **Example**
            · **Q**: What is the smallest object in the image?
            · **A**: spatula
- **Recommendations**: This task generator is optimal for evaluating a model's ability to determine and identify the relative sizes of objects, specifically the largest or smallest items within 3D tabletop images, using annotated data from Objaverse.

# WhereSize3DGridTaskGenerator

- **Basic Information**.
    - **Task Type**. ImageQA
    - **Question Type**. what object
    - **Answer Type**. object category
    - **Data Type**. 3D tabletop image
- **Source Data**.
    - Rendering images of objects from Objaverse.
    - Annotations regarding object category, material, color, visable angles, and shape.
- **Task Plan Schema**.
    - **question type**: `string`. The question type of these tasks will be "where size".
    - **size**: `string`. The type of the size of the target object, indicates whether it pertains to the "largest" or "smallest" in all the objects.
    - **grid number**: `integer`. The number of diagonal grids of the image, $N$ indicates there are $N \times N$ grids in the image. Support $\{2, 3\}$.
    - **target category**: `string`. The category name of the target object.
    - **absolute position**: `string`. The absolute position of the target object in the grid. It is a number ranging from 0 to 3 (grid number = 2) or 0 to 8 (grid number = 3).
    - **reference category**: `string`. The category name of the object that is used to reference the target object.
    - **reference position**: `string`. The relative position of the target object from the reference object.
    - **attribute type**: `string`. The type of attributes of the target object, currently include: `color`, `material`, and `shape`.
    - **attribute value**: `string`. The value of the attributes of the target object.
    - **target-reference order**: `string`. Define the target object goes first or not in the question. It is related to grammar.
- **Partitions**.
    - **Partition 1**.
        * **Template**
            · **Q**: Where is the <size> object in the image?
            · **A**: <absolute position>
        * **Example**
            · **Q**: Where is the largest object in the image?
            · **A**: middle
    - **Partition 2**.
        * **Template**
            · **Q**: Where is the <size> object in the image with respect to the <reference category>?
            · **A**: <reference position>
        * **Example**
            · **Q**: Where is the smallest object in the image with respect to the car?
            · **A**: middle
- **Recommendations**: This task generator is designed for evaluating a model's ability to locate objects based on their size, such as identifying the largest or smallest objects and their positions relative to other objects within 3D tabletop images, using annotated data from Objaverse.

# WhatAttributeSize3DGridTaskGenerator

- **Basic Information**.
    - **Task Type**. ImageQA
    - **Question Type**. what object
    - **Answer Type**. object category
    - **Data Type**. 3D tabletop image
- **Source Data**.
    - Rendering images of objects from Objaverse.
    - Annotations regarding object category, material, color, visable angles, and shape.
- **Task Plan Schema**.
    - **question type**: `string`. The question type of these tasks will be "what attribute size".
    - **size**: `string`. The type of the size of the target object, indicates whether it pertains to the "largest" or "smallest" in all the objects.
    - **grid number**: `integer`. The number of diagonal grids of the image, $N$ indicates there are $N \times N$ grids in the image. Support $\{2, 3\}$.
    - **target category**: `string`. The category name of the target object.
    - **absolute position**: `string`. The absolute position of the target object in the grid. It is a number ranging from 0 to 3 (grid number = 2) or 0 to 8 (grid number = 3).
    - **attribute type**: `string`. The type of attributes of the target object, currently include: `color`, `material`, and `shape`.
    - **attribute value**: `string`. The value of the attributes of the target object.
- **Partitions**.
    - **Partition 1**.
        * **Template**
            · **Q**: What is the <attribute type> of the <size> object in the image?
            · **A**: <attribute value>
        * **Example**
            · **Q**: What is the color of the smallest object in the image?
            · **A**: black
- **Recommendations**: This task generator is ideal for evaluating a model's ability to recognize attributes such as color, material, and shape of objects based on their size, particularly identifying the largest or smallest objects within 3D tabletop images, using annotated data from Objaverse.

# WhatMovementVideoGridTaskGenerator

- **Basic Information**.
  - **Task Type**. VideoQA
  - **Question Type**. what object
  - **Answer Type**. object category
  - **Data Type**. 3D tabletop video
- **Source Data**.
  - Rendering images of objects from Objaverse.
  - Annotations regarding object category, material, color, visable angles, and shape.
- **Task Plan Schema**.
  - **question type**: `string`. The question type of these tasks will be "what move video".
  - **grid number**: `integer`. The number of diagonal grids of the image, $N$ indicates there are $N \times N$ grids in the image. Support $\{2, 3\}$.
  - **target category**: `string`. The category name of the target object.
  - **absolute position**: `string`. The absolute position of the target object in the grid. It is a number ranging from 0 to 3 (grid number = 2) or 0 to 8 (grid number = 3).
  - **attribute type**: `string`. The type of attributes of the target object, currently include: `color`, `material`, and `shape`.
  - **attribute value**: `string`. The value of the attributes of the target object.
  - **moving direction**: `string`. The moving direction of the target object, can be either 'left', 'right', 'up', or 'down'.
  - **are other objects moving**: `string`. Indicates that other objects in the video are moving or not, can be "Yes" or "No". If it is "Yes" moving, it should not be in the same direction of the target object's moving direction.
- **Partitions**.
  - **Partition 1**.
    * **Template**
      · **Q**: What is the object that is moving <moving direction> in the video?
      · **A**: <target category>
    * **Example**
      · **Q**: What is the object that is moving left in the video?
      · **A**: serving tray
  - **Partition 2**.
    * **Template**
      · **Q**: What is the moving object in the video?
      · **A**: <target category>
    * **Example**
      · **Q**: What is the moving object in the video?
      · **A**: barrel
- **Recommendations**: This task generator is effective for evaluating a model's capability to recognize objects and their movements within 3D tabletop videos, using annotated data from Objaverse to assess the identification of moving objects and their directions.

# WhereMovementVideoGridTaskGenerator

- **Basic Information**.
  - **Task Type**. VideoQA
  - **Question Type**. what object
  - **Answer Type**. object category
  - **Data Type**. 3D tabletop video
- **Source Data**.
  - Rendering images of objects from Objaverse.
  - Annotations regarding object category, material, color, visable angles, and shape.
- **Task Plan Schema**.
  - **question type**: `string`. The question type of these tasks will be "where move video".
  - **grid number**: `integer`. The number of diagonal grids of the image, $N$ indicates there are $N \times N$ grids in the image. Support {2, 3}.
  - **target category**: `string`. The category name of the target object.
  - **absolute position**: `string`. The absolute position of the target object in the grid. It is a number ranging from 0 to 3 (grid number = 2) or 0 to 8 (grid number = 3).
  - **attribute type**: `string`. The type of attributes of the target object, currently include: `color`, `material`, and `shape`.
  - **attribute value**: `string`. The value of the attributes of the target object.
  - **moving direction**: `string`. The moving direction of the target object, can be either 'left', 'right', 'up', or 'down'.
  - **are other objects moving**: `string`. Indicates that other objects in the video are moving or not, can be "Yes" or "No". If it is "Yes" moving, it should not be in the same direction of the target object's moving direction.
- **Partitions**.
  - **Partition 1**.
    * **Template**
      · **Q**: Where is the object that is moving down located in the video?
      · **A**: <absolute position>
    * **Example**
      · **Q**: Where is the object that is moving down located in the video?
      · **A**: back right
  - **Partition 2**.
    * **Template**
      · **Q**: Where is the moving object located in the video?
      · **A**: <absolute position>
    * **Example**
      · **Q**: Where is the moving object located in the video?
      · **A**: back right
- **Recommendations**: This task generator is ideal for evaluating a model's ability to locate moving objects within 3D tabletop videos, focusing on both the movement direction and the absolute position of objects using annotated data from Objaverse.

# WhatAttributeMovementVideoGridTaskGenerator

- **Basic Information**.
  - **Task Type**. VideoQA
  - **Question Type**. what object
  - **Answer Type**. object category
  - **Data Type**. 3D tabletop video
- **Source Data**.
  - Rendering images of objects from Objaverse.
  - Annotations regarding object category, material, color, visable angles, and shape.
- **Task Plan Schema**.
  - **question type**: `string`. The question type of these tasks will be "what attribute move video".
  - **size**: `string`. The type of the size of the target object, indicates whether it pertains to the "largest" or "smallest" in all the objects.
  - **grid number**: `integer`. The number of diagonal grids of the image, $N$ indicates there are $N \times N$ grids in the image. Support $\{2, 3\}$.
  - **target category**: `string`. The category name of the target object.
  - **absolute position**: `string`. The absolute position of the target object in the grid. It is a number ranging from 0 to 3 (grid number = 2) or 0 to 8 (grid number = 3).
  - **attribute type**: `string`. The type of attributes of the target object, currently include: `color`, `material`, and `shape`.
  - **attribute value**: `string`. The value of the attributes of the target object.
- **Partitions**.
  - **Partition 1**.
    * **Template**
      · **Q**: What is the <attribute type> of the object that is moving <moving direction> in the video?
      · **A**: <attribute value>
    * **Example**
      · **Q**: What is the color of the object that is moving left in the video?
      · **A**: black
  - **Partition 2**.
    * **Template**
      · **Q**: Where is the <attribute type> of the moving object in the video?
      · **A**: <attribute value>
    * **Example**
      · **Q**: What is the color of the moving object in the video?
      · **A**: white
- **Recommendations**: This task generator is effective for evaluating a model's capability to recognize attributes such as color, material, and shape of moving objects within 3D tabletop videos, using annotated data from Objaverse to assess the identification of attributes for objects in motion.

# WhatRotationVideoGridTaskGenerator

- **Basic Information**.
  - **Task Type**. VideoQA
  - **Question Type**. what object
  - **Answer Type**. object category
  - **Data Type**. 3D tabletop video
- **Source Data**.
  - Rendering images of objects from Objaverse.
  - Annotations regarding object category, material, color, visable angles, and shape.
- **Task Plan Schema**.
  - **question type**: `string`. The question type of these tasks will be "what rotate video".
  - **size**: `string`. The type of the size of the target object, indicates whether it pertains to the "largest" or "smallest" in all the objects.
  - **grid number**: `integer`. The number of diagonal grids of the image, $N$ indicates there are $N \times N$ grids in the image. Support $\{2, 3\}$.
  - **target category**: `string`. The category name of the target object.
  - **absolute position**: `string`. The absolute position of the target object in the grid. It is a number ranging from 0 to 3 (grid number = 2) or 0 to 8 (grid number = 3).
  - **attribute type**: `string`. The type of attributes of the target object, currently include: `color`, `material`, and `shape`.
  - **attribute value**: `string`. The value of the attributes of the target object.
- **Partitions**.
  - **Partition 1**.
    * **Template**
      · **Q**: What is the <size> object in the image?
      · **A**: <target category>
    * **Example**
      · **Q**: What is the smallest object in the image?
      · **A**: spatula
- **Recommendations**: This task generator is suitable for evaluating a model's capability to recognize objects undergoing rotation within 3D tabletop videos, using annotated data from Objaverse to identify objects based on their rotation and attributes.

# WhereRotationVideoGridTaskGenerator

- **Basic Information**.
    - **Task Type**. VideoQA
    - **Question Type**. what object
    - **Answer Type**. object category
    - **Data Type**. 3D tabletop video
- **Source Data**.
    - Rendering images of objects from Objaverse.
    - Annotations regarding object category, material, color, visable angles, and shape.
- **Task Plan Schema**.
    - **question type**: `string`. The question type of these tasks will be "where rotate video".
    - **size**: `string`. The type of the size of the target object, indicates whether it pertains to the "largest" or "smallest" in all the objects.
    - **grid number**: `integer`. The number of diagonal grids of the image, $N$ indicates there are $N \times N$ grids in the image. Support $\{2, 3\}$.
    - **target category**: `string`. The category name of the target object.
    - **absolute position**: `string`. The absolute position of the target object in the grid. It is a number ranging from 0 to 3 (grid number = 2) or 0 to 8 (grid number = 3).
    - **reference category**: `string`. The category name of the object that is used to reference the target object.
    - **reference position**: `string`. The relative position of the target object from the reference object.
    - **attribute type**: `string`. The type of attributes of the target object, currently include: `color`, `material`, and `shape`.
    - **attribute value**: `string`. The value of the attributes of the target object.
    - **target-reference order**: `string`. Define the target object goes first or not in the question. It is related to grammar.
- **Partitions**.
    - **Partition 1**.
        * **Template**
            · **Q**: Where is the <size> object in the image?
            · **A**: <absolute position>
        * **Example**
            · **Q**: Where is the largest object in the image?
            · **A**: middle
    - **Partition 2**.
        * **Template**
            · **Q**: Where is the <size> object in the image with respect to the <reference category>?
            · **A**: <reference position>
        * **Example**
            · **Q**: Where is the smallest object in the image with respect to the car?
            · **A**: middle
- **Recommendations**: This task generator is ideal for evaluating a model's ability to locate rotating objects within 3D tabletop videos, focusing on identifying the positions of the largest or smallest objects relative to other objects, using annotated data from Objaverse.

# WhatAttributeRotationVideoGridTaaskGenerator

- **Basic Information**.
    - **Task Type**. VideoQA
    - **Question Type**. what object
    - **Answer Type**. object category
    - **Data Type**. 3D tabletop video
- **Source Data**.
    - Rendering images of objects from Objaverse.
    - Annotations regarding object category, material, color, visable angles, and shape.
- **Task Plan Schema**.
    - **question type**: `string`. The question type of these tasks will be "what attribute rotate video".
    - **size**: `string`. The type of the size of the target object, indicates whether it pertains to the "largest" or "smallest" in all the objects.
    - **grid number**: `integer`. The number of diagonal grids of the image, $N$ indicates there are $N \times N$ grids in the image. Support $\{2, 3\}$.
    - **target category**: `string`. The category name of the target object.
    - **absolute position**: `string`. The absolute position of the target object in the grid. It is a number ranging from 0 to 3 (grid number = 2) or 0 to 8 (grid number = 3).
    - **attribute type**: `string`. The type of attributes of the target object, currently include: `color`, `material`, and `shape`.
    - **attribute value**: `string`. The value of the attributes of the target object.
- **Partitions**.
    - **Partition 1**.
        * **Template**
            · **Q**: What is the <attribute type> of the <size> object in the image?
            · **A**: <attribute value>
        * **Example**
            · **Q**: What is the color of the smallest object in the image?
            · **A**: black
- **Recommendations**: This task generator is effective for evaluating a model's ability to recognize attributes such as color, material, and shape of rotating objects within 3D tabletop videos, using annotated data from Objaverse to assess the identification of attributes for objects in rotation.

## WhatObjectSceneGraphTaskGenerator

- **Basic Information**.
    - **Task Type**. ImageQA
    - **Question Type**. what object
    - **Answer Type**. object category
    - **Data Type**. 3D tabletop image
- **Source Data**.
    - Real images from GQA versions of Visual Genome with its corresponding scene graphs
    - scene graph contains the objects, relations, and attributes in the image.
- **Task Plan Schema**.
    - **question type**: `string`. The question type of these tasks will be "what object".
    - **object** : `string`. The target object node of the question.
    - **subgraph** : `string`. The subgraph with the target object node as its root, used to reference the target object node.
    - **scene graph id** : `string`. The identifier of the scene graph.
    - **answers**: `list`. A list of object nodes in the scene graph that share the same subgraph structure, except the target object node and itself.
- **Partitions**.
    - **Partition 1**.
        * **Template**
            · **Q**: What is the <object and its attributes in the subgraph> that <obj reference(other reference objects, attributes, and relations in the subgraph)>?
            · **A**: <target category>
        * **Example**
            · **Q**: What is the flat object that is on the brown and wood table?
            · **A**: paper
- **Recommendations**: This task generator is well-suited for evaluating a model's ability to recognize objects and their relationships within complex scenes, using real images and corresponding scene graphs from GQA versions of Visual Genome to provide detailed contextual information for object identification.

# WhatAttributeSceneGraphTaskGenerator

- **Basic Information**.
  - **Task Type**. ImageQA
  - **Question Type**. what attribute
  - **Answer Type**. attribute
  - **Data Type**. real image
- **Source Data**.
  - Real images from GQA versions of Visual Genome with its corresponding scene graphs
  - scene graph contains the objects, relations, and attributes in the image.
- **Task Plan Schema**.
  - **question type**: `string`. The question type of these tasks will be "what attribute".
  - **attribute type** : `string`. The type of the target attribute.
  - **attribute** : `string`. The target attribute node of the question.
  - **subgraph** : `string`. The subgraph with the target attribute node as its root.
  - **scene graph id** : `string`. The identifier of the scene graph.
  - **answers**: `list`. A list of attribute nodes in the scene graph that share the same subgraph structure, except the target attribute node and itself.
- **Partitions**.
  - **Partition 1**.
    * **Template**
      · **Q**: What is the <attribute type> of the <target attribute's corresponding object and object's other attributes in the subgraph> that <obj reference(other reference objects, attributes, and relations in the subgraph)>?
      · **A**: <attribute>
    * **Example**
      · **Q**: What is the material of the smooth object that is to the right of the yellow container?
      · **A**: plastic
- **Recommendations**: This task generator is ideal for evaluating a model's ability to recognize and identify attributes within complex scenes, using real images and corresponding scene graphs from GQA versions of Visual Genome to provide detailed contextual information for attribute identification.

# WhatRelationSceneGraphTaskGenerator

- **Basic Information**.
    - **Task Type**. ImageQA
    - **Question Type**. what object
    - **Answer Type**. object
    - **Data Type**. real image
- **Source Data**.
    - Real images from GQA versions of Visual Genome with its corresponding scene graphs
    - Scene graph contains the objects, relations, and attributes in the image.
- **Task Plan Schema**.
    - **question type**: `string`. The question type of these tasks will be "what relation".
    - **relation**: `string`. The target relation edge between source object node and target object node
    - **source object**: `string`. The source object node of the question.
    - **target object** : `string`. The target object node of the question.
    - **source subgraph** : `string`. The subgraph with the source object node as its root.
    - **target subgraph** : `string`. The subgraph with the target object node as its root.
    - **scene graph id** : `string`. The identifier of the scene graph.
    - **answers**: `list`. A list of relation edges in the scene graph that connect the same source subgraph and target subgraph.
- **Partitions**.
    - **Partition 1**.
        * **Template**
            · **Q**: What is the relation from the <source object's attributes in the source subgraph> object, which <source obj reference(other reference objects, attributes, and relations in the source subgraph)>, to the <target object's attributes in the source subgraph> object, which <target obj reference(other reference objects, attributes, and relations in the target subgraph)>?
            · **A**: <relation>
        * **Example**
            · **Q**: What is the relation from the standing object, which the colorful and long snowboard is to the right of, to the blue and long object, which is to the left of the patterned skis?
            · **A**: holding
- **Recommendations**: This task generator is well-suited for evaluating a model's ability to recognize and identify relationships between objects within complex scenes, using real images and corresponding scene graphs from GQA versions of Visual Genome to provide detailed contextual information for understanding object relations.

# WhatObjectVideoSceneGraphTaskGenerator

- **Basic Information**.
  - **Task Type**. VideoQA
  - **Question Type**. what object
  - **Answer Type**. object
  - **Data Type**. real video
- **Source Data**.
  - Real videos from AGQA version of Action Genome with its corresponding scene graph
  - Scene graph contains each key frame's objects, relations, and actions.
- **Task Plan Schema**.
  - **question type**: `string`. The question type of these tasks will be "what object video".
  - **object** : `string`. The target object the person in the video interacts with.
  - **relation** : `string`. The relation between the person and the target object it interacts with.
  - **reference action** : `string`. The reference action to locate the moment when a person is interacting with the target object.
  - **reference type** : `string`. The target object of the relation between the person and the target object it interacts with, can be "spatial" or "contact".
  - **temporal reference type** : `string`. Type of the temporal reference between the reference action and the moment when a person is interacting with the target object. Can be "before", "while", or "after".
  - **video scene graph id** : `string`. The identifier of the video scene graph.
- **Partitions**.
  - **Partition 1**.
    * **Template**
      · **Q**: What is the object that the person is <reference> <temporal reference type> the person <reference action>?
      · **A**: <object>
    * **Example**
      · **Q**: What is the object that the person is behind after the person watching something in a mirror?
      · **A**: floor
- **Recommendations**: This task generator is excellent for evaluating a model's capability to recognize and identify objects and their interactions within video scenes, using real videos and corresponding scene graphs from AGQA version of Action Genome to provide detailed contextual and temporal information for understanding object interactions.

# WhatRelationVideoSceneGraphTaskGenerator

- **Basic Information**.
    - **Task Type**. VideoQA
    - **Question Type**. what relation
    - **Answer Type**. relation
    - **Data Type**. real video
- **Source Data**.
    - Real videos from AGQA version of Action Genome with its corresponding scene graph
    - Scene graph contains each key frame's objects, relations, and actions.
- **Task Plan Schema**.
    - **question type**: `string`. The question type of these tasks will be "what relation video".
    - **object** : `string`. The object the person in the video interacts by the target relation.
    - **relation** : `string`. The target relation between the person and the target object it interacts with.
    - **reference action** : `string`. The reference action to locate the moment when a person is interacting with the object.
    - **reference type** : `string`. The type of the target relation between the person and the object it interacts with, can be "spatial" or "contact".
    - **temporal reference type** : `string`. Type of the temporal reference between the reference action and the moment when a person is interacting with the object. Can be "before", "while", or "after".
    - **video scene graph id** : `string`. The identifier of the video scene graph.
- **Partitions**.
    - **Partition 1**.
        * **Template**
            · **Q**: What is the spatial relation of the person to the <object> while the person <reference action>.
            · **A**: <relation>
        * **Example**
            · **Q**: What is the spatial relation of the person to the closet while the person closing a closet?
            · **A**: behind
    - **Partition 2**.
        * **Template**
            · **Q**: What is the person doing to the <object> before the person <reference action>?
            · **A**: <relation>
        * **Example**
            · **Q**: What is the person doing to the blanket before the person putting a phone somewhere?
            · **A**: touching
- **Recommendations**: This task generator is suitable for evaluating a model's ability to recognize and identify relationships between objects and actions within video scenes, using real videos and corresponding scene graphs from AGQA version of Action Genome to provide detailed contextual and temporal information for understanding object relations and interactions.

# WhatActionVideoSceneGraphTaskGenerator

- **Basic Information**.
    - **Task Type**. VideoQA
    - **Question Type**. what action
    - **Answer Type**. action
    - **Data Type**. real image
- **Source Data**.
    - Real videos from AGQA version of Action Genome with its corresponding scene graph
    - Scene graph contains each key frame's objects, relations, and actions.
- **Task Plan Schema**.
    - **question type**: `string`. The question type of these tasks will be "what action video".
    - **action** : `string`. The target action that the person in the video performs.
    - **reference action** : `string`. The reference action to locate the moment when a person is performing the target action.
    - **temporal reference type** : `string`. Type of the temporal reference between the reference action and the moment when a person is performing the target action. Can be "before", "while", or "after".
    - **video scene graph id** : `string`. The identifier of the video scene graph.
- **Partitions**.
    - **Partition 1**.
        * **Template**
            · **Q**: What action is the person doing while <reference action>?
            · **A**: <action>
        * **Example**
            · **Q**: What action is the person doing while laughing at something?
            · **A**: sitting at a table
- **Recommendations**: This task generator is excellent for evaluating a model's ability to recognize and identify actions performed by individuals within video scenes, using real videos and corresponding scene graphs from AGQA version of Action Genome to provide detailed temporal and contextual information for understanding actions.