# OpenReview forum: "Task Me Anything"
_NeurIPS.cc/2024/Datasets_and_Benchmarks_Track — NeurIPS 2024 Track Datasets and Benchmarks Poster_

### Official Review · Reviewer_d4Ru · 2024-07-19
**Very interesting and useful resource for VLMs**

**Rating:** 8
**Confidence:** 4

**Review:**

I judge the work to be significant and clear for the most part w.r.t. to its goals and aspirations, with very little detail lacking. Their repo provides further documentation and demonstrations, which makes their work very likely to be used and useful. The dataset generation and data generation scheme that they propose allows for targeted evaluation of specific aspects, which can help disentangle the benefits and the struggles of different models, and can act as benchmarks that can guide the advancement of VMLs. A few details in the paper need some polish.

**Strengths:**

As aforementioned, the scale, the ease of use, and the ability to evaluate targeted capabilities of VLMs makes the generation scheme extremely useful for the targeted evaluation of different capabilities in VLMs.

**Additional Feedback:**

Nice work!

**Clarity:**

As mentioned above, the fact that natural language is not used for the queries should be made clearer, and perhaps the usage of some terms can be confusing.

**Correctness:**

The authors control for the generation of the data, so the assertions made and the results presented in the paper seem very likely to be correct and accurate.

**Documentation:**

The authors have provided enough details in the paper, as well as a well designed repo to go with the manuscript.

**Limitations:**

The authors don’t make clear the extend to which the data is generated vs. real. As I mention above and the authors acknowledge in the supplementary, generated data might not necessarily reflect IRL performance. Of course, the generation process ensures that the models are indeed tested on the requested capabilities, yet how transferable this is to real data might still remain a question. Furthermore, since automated methods are used to query data even for real-world data, the human evaluation should have looked at whether the generated datasets are actually doing what they’re supposed to. If the authors implicitly mean that the humans looked for and did not find any mistakes by not mentioning this issue, it should become explicit in subsequent versions of the manuscript. For example, have there been any studies looking at the error rate in human annotations in the used datasets?

**Opportunities For Improvement:**

Throughout the paper, a couple of things really confused me. Majorly, I found no mentions of how the seemingly natural language queries were translated into actionable queries within the dataset generation engine, what’s the possibility of error, etc. It only became evident from the supplementary material and the repo that a specific UI with knobs and switches is utilized. That should be made clear from the main text. For example, line 39 seems to suggest that the system is indeed using some sort of semantic parser for natural language. Finally, I found the usage of “task” and “instance” surprisingly confusing to keep up with, since my intuition about which should be used where were conflicting with the actual usage.

Moreover, I believe that whatever discussion about the limitations of the framework due to generated data should be brought from the supplementary material to the main text, and the discussion should be expanded (perhaps the additional details can be placed in the supplementary).

**Relation To Prior Work:**

The authors describe previous work adequately, although some mentions of the validity of the datasets, if available, should be discussed to reassure users of the validity of the generated.

**Summary And Contributions:**

The authors introduce a benchmark generation engine for Image- and Video-Language models that can accommodate testing for various spatial reasoning capabilities, etc. The generated benchmark comprises instances synthetically generated or real that can allow practitioners and other researchers to test particular features of their models. The authors provide analysis of existing models, and perform human evaluations to bolster their claims.

---

> ### Author Rebuttal · Authors · 2024-08-17
>
> **how transferable generated data is to real data might still remain a question**
>
> Thanks for your questions and comments! To answer your question on “the extent to which data is generated vs. real”, we note that TaskMeAnything can enable evaluations with up to 113K+ realistic images and 9K+ videos and infinite synthetic images and videos. As for our evaluation on the random set, we’ve included 1,500 2D images, 3,300 3D images and 900 real-world images as well as 1800 3D synthetic and 900 realistic videos.
>
> Please refer to our first point in the **General Response (paragraphs 3-7)** for detailed discussions and analyses on the transferability between evaluations on synthetic and real-world images.
>
> **have there been any studies looking at the error rate in human annotations in the used datasets?**
>
> Yes. The human evaluation has checked the generated benchmark is correct and the system is doing what they are supposed to do. We will make it explicit in the next version.
>
> In fact, we have gone through multiple rounds of human evaluation before we finalized the system. Because our system, by design, facilitates easy benchmark debugging and regeneration, every time some errors of the generated benchmarks are reported, we could easily attribute the error to either the source human annotation or the code we used to generate benchmarks, and then fixed them right away and re-ran the code to re-generate benchmarks.
>
>
> **Clarification on natural language queries vs actionable queries, and definition of task and instance**
>
> Thanks for the suggestion! We will make both points clearer in future versions.
>
> **Discussion about the limitations of the framework due to generated data should be brought from the supplementary material to the main text**
>
> That’s a great suggestion! We will move them upfront and add more details.

---

> > ### Comment · Reviewer_d4Ru · 2024-08-26
> > **Reviewer Response**
> >
> > Thanks for the reply, good luck with your paper.

---

### Official Review · Reviewer_jor3 · 2024-07-22

**Rating:** 6
**Confidence:** 3
**Correctness:** Yes.
**Clarity:** Yes.

**Review:**

### Strengths
1. The paper is generally well-written and easy to follow.
2. The task of automatically creating benchmarks for MLMs given a specific application is practical in the real world.
3. To the best of my knowledge, this is the first work that tackles the above task for MLMs, although some core components are similar to existing techniques.
4. The conclusions drawn from the benchmark are useful in facilitating the development of MLMs.

### Weaknesses
1. As mentioned in Sec. 2.2, the authors introduce the process of synthesizing VQA instances. I'm not sure if the images/videos produced by such a process have any domain gap with real-world applications, especially in fields like finance or medicine. Does MLM that performs well on synthetic images/videos also perform well on real-domain images/videos?
2. Although the authors argue that they propose an automatic engine rather than an off-the-shelf benchmark, which is also the main difference from existing works, I have concerns about the generalization ability of such an engine. As the taxonomy, skills of the tasks, and task templates are predefined, it is actually a close-set benchmark with fine-grained subtasks. Let me know if I have any misunderstandings.
3. Although the authors have discussed existing benchmarks in the Appendix, is it possible to add more experimental comparisons on different benchmarks as previous ones (e.g., MMBench, SEED-Bench) also have fine-grained subtasks? For example, is it possible to verify that on the same subtask, the benchmark created with the proposed engine is more accurate than the existing benchmark in evaluating MLMs?
4. As mentioned in L74-85, we appreciate the conclusions drawn from the benchmarks. However, most of the existing conclusions are common sense, such as larger models outperforming smaller models. Are there other interesting insights?

**Strengths:**

Please see **Strengths** in #Review.

**Additional Feedback:**

n/a

**Documentation:**

Yes, the code is open-source.

**Ethics:**

No.

**Limitations:**

The authors discuss the limitation of validating only the task of visual perception. Another limitation should be the domain gap between synthetic images/videos and the real world.

**Opportunities For Improvement:**

Please see **Weaknesses** in #Review.

**Relation To Prior Work:**

Yes. It is recommended to add more experimental comparisons.

**Summary And Contributions:**

The paper introduces a benchmark generation engine, namely, Task-Me-Anything, for properly evaluating multimodal language models (MLMs) under various scenarios. Specifically, the engine comprises a taxonomy, programmatic task generators, and approximate evaluation algorithms to save computational overhead. The current version focuses on visual perception but the authors claim that the engine is extendable. Based on the proposed benchmark engine, the authors draw several conclusions about existing MLMs, e.g., existing MLMs are not good at counting, spatial and temporal understanding.

---

> ### Author Rebuttal · Authors · 2024-08-17
>
> **Does MLM that performs well on synthetic images/videos also perform well on real-domain images/videos?**
>
> Thanks for your question! Our short answer to your question is: Yes, we observed strong positive correlations between models’ rankings on synthetic images/videos and on real-world images/videos. Please refer to our first point in the **General Response (paragraphs 3-7)** for detailed discussions and analyses on the transferability between evaluations on synthetic and real-world images.
>
> **Any domain gap with real-world applications, especially in fields like finance or medicine**
>
> The produced  images/videos may have domain gaps for fields like finance and medicine, yet in TaskMeAnything we show how to leverage real images/videos to generate benchmarks. In particular, we can generate benchmarks based on real images from Visual Genome and real videos from Charades dataset with additional scene graph annotations. To build TaskMeAnything for domains like finance and medicine, we would also need some additional annotations and specialized design, but the core paradigm of user query -> generation of a corresponding benchmark -> efficient evaluation and fine-grained localization of model weaknesses will still be applicable as well as our algorithms for evaluation query results approximations.
>
> **it is actually a close-set benchmark with fine-grained subtasks**
>
> TaskMeAnything is a versatile and extendable system, where the taxonomy and task generator is not static. We maintain the system itself (eg, codebase and assets used to generate benchmarks) rather than a static benchmark and by design, the taxonomy and task generators are extendable, so one can easily expand the taxonomy (eg, adding more 3D objects or other assets) or add new templates/task generators to the system (with minimal modification of the codebase) to expand the task space. In fact, we plan to gradually expand the task space by expanding the taxonomy and adding new task generators, and we also welcome contributions from the community to collectively grow the system.
>
> **More comparison with existing benchmark**
>
> Thanks for your suggestions! While evaluating the accuracy of benchmark might be difficult, we show the correlation between our generated benchmark and existing ones as in the **General Response**. We also want to point out that although existing benchmarks have different categories of tasks, the TaskMeAnything system supports generating benchmarks as fine-grained as, for example, objects- one can generate benchmarks particularly focusing on objects like apples or fish. To the best of our knowledge, existing benchmarks for MLMs do not have such a fine-grained granularity and support user query regarding fine-grained visual concepts.

---

> > ### Author Rebuttal · Authors · 2024-08-17
> >
> > **The conclusions are common sense, are there other interesting insights?**
> >
> > Due to the length limitations, we didn’t elaborate on our analysis in detail in the main body of the paper. Here we highlight some interesting insights and new analysis we have since submission.
> >
> > ***Detailed analysis on prompt sensitivity.***
> > We tested more recent models (GLM-4V, CogVLM2, Idefics2, Phi-3-vision, PaliGemma) on our random subsets of tasks. First, we observed that MLMs are highly sensitive to prompts, with different prompts resulting in performance differences ranging from 0% to 40%. However, the newer models are becoming more stable with different prompts (e.g., CogVLM-2 and InternVL-v1.5 perform nearly the same with different prompts, please find an visualization of performance difference wrt the release time in the attached PDF).
> >
> > Another interesting finding is that, for most models, detailed prompts lead to slightly better results, while GPT-4v and GPT-4o perform notably better with succinct versions of prompts; this difference between GPT models and other models indicates that GPT models may have a significantly different way of training or fine-tuning their models.
> >
> > ImageQA
> > | **Model**                 | **Detailed Prompt** | **Succinct Prompt** | **Difference** |
> > |---------------------------|------------|------------|--------------|
> > | InstructBLIP-7B            | 30.36     | 0.64      | -29.72       |
> > | InstructBLIP-13B           | 29.70     | 0.68      | -29.02       |
> > | Qwen-VL                    | 29.98     | 11.89     | -18.10       |
> > | Qwen-VL-Chat               | 34.26     | 34.22     | -0.04        |
> > | LLaVA-7B                   | 30.71     | 31.95     | +1.24        |
> > | LLaVA-13B                  | 34.76     | 33.24     | -1.52        |
> > | GLM-4v                     | 39.22     | 38.45     | -0.77        |
> > | Idefics2-8B                | 36.74     | 37.87     | +1.13        |
> > | Phi-3-vision-3B            | 34.96     | 37.31     | +2.35        |
> > | PaliGemma-3B               | 39.85     | 40.63     | +0.79        |
> > | CogVLM2-19B                | 38.91     | 39.67     | +0.76        |
> > | InternVL-Chat-1.5-24B      | 45.80     | 43.88     | -1.92        |
> > | GPT4o                      | 38.45     | 47.40     | +8.95        |
> >
> >
> > VideoQA
> > | **Model**                 | **Detailed Prompt** | **Succinct Prompt** | **Difference** |
> > |---------------------------|--------------------|---------------------|----------------|
> > | Video-ChatGPT-7B           | 14.21              | 12.37               | -1.84          |
> > | Video-LLaVA-7B             | 20.64              | 28.34               | 7.71           |
> > | VideoChat2-7B              | 26.00              | 18.22               | -7.79          |
> > | Video-LLaMA-2-7B           | 23.06              | 15.48               | -7.58          |
> > | Video-LLaMA-2-13B          | 20.24              | 13.72               | -6.52          |
> > | Chat-UniVi-7B              | 26.46              | 20.57               | -5.90          |
> > | Chat-UniVi-13B             | 22.57              | 17.98               | -4.59          |
> > | InternVL-Chat-1.5-24B      | 38.86              | 30.80               | -8.06          |
> > | GPT4o                      | 42.42              | 46.38               | 3.96           |

---

> > ### Author Rebuttal · Authors · 2024-08-17
> >
> > ***Analysis on a challenging benchmark generated by TaskMeAnything.***
> > In addition to the random subsets of tasks we discussed in the submission, we created a challenging benchmark using the TaskMeAnything system. In particular, we use the Top-K queries and active approximation algorithms to  automatically find challenging ImageQA and VideoQA tasks for the same batch MLMs as above.
> >
> > Evaluation of open-source models and GPT-4o on this new benchmark reveals a 10-30% performance drop compared to the random subset of TaskMeAnything. In particular, we present performance comparison of models on both benchmarks as below. One interesting findings is that on random subset Phi-3-vision-3B’s performance is close to GPT4o (only 0.2 point of difference), but it becomes larger on the challenging subset (GPT4o is better by 6 point), this indicates a static set of tasks may underestimate the performance gap between models, while our TaskMeAnything system allows user to conduct rigorous model comparison given the large task space it provides. We are actively working on more detailed analysis and releasing of this new challenging benchmark. More details of this new benchmark can be found in our github repository.
> >
> > ImageQA
> > | **Model**                   | **Random** | **Challenging** | **Performance Drop** |
> > |-----------------------------|---------------------------------|------------------------------|----------------------|
> > | InstructBlip-7B             | 22.04                           | 15.50                        | -6.54                |
> > | InstructBlip-13B            | 40.23                           | 15.19                        | -25.04               |
> > | Qwen-VL                     | 29.58                           | 20.93                        | -8.65                |
> > | Qwen-VL-Chat                | 52.05                           | 34.24                        | -17.81               |
> > | LLaVA-7B                    | 48.97                           | 31.33                        | -17.64               |
> > | LLaVA-13B                   | 56.19                           | 34.00                        | -22.19               |
> > | GLM-4v                      | 59.57                           | 38.83                        | -20.74               |
> > | Idefics2-8B                 | 57.95                           | 37.31                        | -20.65               |
> > | Phi-3-vision-3B             | 59.59                           | 36.14                        | -23.46               |
> > | PaliGemma-3B                | 58.70                           | 40.24                        | -18.46               |
> > | CogVLM2-19B                 | 56.68                           | 39.29                        | -17.38               |
> > | InternVL-Chat-1.5-24B       | 67.28                           | 44.84                        | -22.45               |
> > | GPT4o                       | 59.74                           | 42.93                        | -16.81               |
> >
> > VideoQA
> > | **Model**                  | **Random** | **Challenging** | **Performance Drop** |
> > |----------------------------|---------------------------------|------------------------------|----------------------|
> > | Video-ChatGPT-7B           | 24.74                           | 13.29                        | -11.45               |
> > | Video-LLaVA-7B             | 38.39                           | 24.49                        | -13.90               |
> > | Video-ChatGPT-2-7B         | 32.49                           | 22.11                        | -10.38               |
> > | Video-LLaMA-2-7B           | 27.03                           | 19.27                        | -7.77                |
> > | Video-LLaMA-2-13B          | 25.53                           | 16.98                        | -8.55                |
> > | ChatUni-Vi-S               | 37.70                           | 23.51                        | -14.18               |
> > | ChatUni-Vi-L               | 34.74                           | 20.28                        | -14.46               |
> > | InternVL-Chat-1.5-24B              | 48.61                           | 34.83                        | -13.78               |
> > | GPT-4o                     | 53.29                           | 44.40                        | -8.89                |

---

> > ### Author Response · Authors · 2024-08-27
> >
> > Dear Reviewer:
> >
> > Thanks again for your efforts in reviewing. The discussion deadline is approaching. We hope that you can check our response and let us know whether your concern is addressed. Thanks!

---

### Official Review · Reviewer_PNK4 · 2024-07-25
**Review for Task-Me-Anything**

**Rating:** 6
**Confidence:** 4
**Clarity:** The paper is well written.

**Review:**

The proposed Task-Me-Anything is a novel benchmark generation process. Both the idea and experiments are interesting and insightful.

**Strengths:**

- The proposed Task-Me-Anything (TMA) is interesting and well-designed in the context of MLM evaluation.
- The taxonomy and the task generation process are detailed and informative.
- The experiments are insightful. The paper presents a solid analysis of MLMs by using the proposed TMA. The discussed aspects are diverse and interesting, benefiting from the flexibility of benchmark generation process.

**Additional Feedback:**

N/A

**Correctness:**

The data construction is valid but limited to perception tasks. It is not "any" multimodal tasks

**Documentation:**

There are sufficient details.

**Limitations:**

see session "Opportunities For Improvement"

**Opportunities For Improvement:**

Overall this is a good paper. However, I find some limitations that need to be further improved.

- One critical limitation is that the current benchmark generation only considers the perception tasks (or computer vision tasks). As the paper claims asking for "anything", one would expect a variety of possible multimodal benchmark tasks. Since the current method highly relies on 3D render engine, it is unclear how the current method can be extended to other modalities.

- In general, people are more interested in the evaluation with real data and real scenarios, instead of synthetic ones. The observations on synthetic benchmarks may not be always transferable to real datasets. In other words, the value of synthetic benchmarks remains unclear. The synthetic benchmark generally cannot replace the real ones.

**Relation To Prior Work:**

The discussion is clear.

**Summary And Contributions:**

The key contribution is a new concept of user-driven benchmark generation engine. It is very flexible and scalable to generate a large number of tasks with desired input-output pairs.

This paper also provides insights regarding the perception performance of existing MLMs.

---

> ### Author Rebuttal · Authors · 2024-08-17
>
> **The benchmark only focuses on “perception” tasks, and the reliance on 3D render engine might limit it to extend to other modalities.**
>
> The core idea behind “Task Me Anything” is to establish a paradigm and system that can easily provide a huge amount of benchmarking tasks for varying evaluation needs. The core paradigm of TaskMeAnything: user query -> generation of a corresponding benchmark -> efficient evaluation and fine-grained localization of model weaknesses—is designed to be versatile. This approach is not inherently tied to 3D render engines or synthetic scenarios. As we also included 100k real images (from Visual Genome), and 10k real videos (from image) in TaskMeAnything. Although we are focusing on perception tasks in this first version (because perception is already worth a huge task space), we plan to gradually extend the system with more diverse multimodal tasks in the future.
>
> It’s worth noting that this paradigm is not limited to even large multimodal models, we have ongoing projects that apply this paradigm to other modalities and models, including LLMs, diffusion models, AI for healthcare, etc.
>
> **The observations on synthetic benchmarks may not be always transferable to real datasets.**
>
> Thanks for your comments! Please refer to our first point in the **General Response (paragraphs 3-7)** for detailed discussions and analyses on the transferability between evaluations on synthetic and real-world images.

---

> > ### Comment · Reviewer_PNK4 · 2024-08-31
> > **comment**
> >
> > Thank the authors for their feedback. It would be great to add these discussions to the final paper. I am positive to this paper.

---

> > > ### Author Response · Authors · 2024-08-31
> > > **Thanks!**
> > >
> > > Thank you for being positive to our work, we will incorporate these in our final version!
> > >
> > > If our rebuttal addressed your concern, would you mind reconsidering the overall score for our work? Thanks!

---

> ### Author Response · Authors · 2024-08-27
>
> Dear Reviewer:
>
> Thanks again for your efforts in reviewing. The discussion deadline is approaching. We hope that you can check our response and let us know whether your concern is addressed. Thanks!

---

### Author Rebuttal · Authors · 2024-08-17

We appreciate the feedback from all reviewers. We thank Reviewer PNK4 for recognizing the Task-Me-Anything as “interesting and well-designed,” finding our taxonomy and task generation process “detailed and informative.” and our experiments were deemed “insightful”. We thank Reviewer jor3 for appreciating the practicality of automatically creating benchmarks for MLMs, for the recognition of our work as the first to address this task for MLMs, and for finding our conclusions useful for the development of MLMs. Finally, we thank Reviewer d4Ru for noting that our dataset generation and evaluation scheme are valuable for targeted assessment of MLMs.

In this response, we discuss additional studies to address common inquiries raised by reviewers. We will update the paper with these additional analyses. Please let us know if there are any follow-up questions, and we will make sure to answer all of them during the paper discussion period and in our final revision.

**All the reviewers raised concerns about the potential gap between synthetic data and real data and asked for more analysis**

We thank the reviewers for pointing out this potential domain gap. We highlight a few points and analyses below to address these concerns.

First, we note that the current TaskMeAnything benchmark includes not just 2D and 3D synthetic images but also a large number of real-world images and videos – 113K+ images and 9K+ videos –  that were taken by people in the real world.

Second, to quantitatively justify the transferability between evaluations on synthetic and real-world images, we compared 18 models’ rankings on synthetic vs. realistic images and obsered strong positive correlations between them. Across the three different image sources in Task-Me-Anything, we found that the correlation between models’ rankings on 2D and 3D synthetic images is the strongest (r=0.99). While the correlations between models’ rankings on real and 2D images, and between real and 3D images are slightly smaller (r=0.85 and 0.79 respectively), these numbers still suggest strong positive correlations between models’ performance on synthetic images and on real images. (**Figure 1 of the attached PDF**).

Similarly, we have also conducted a comparative analysis of models’ “counting” performance on Task-Me-Anything’s synthetic images vs. on TallyQA’s realistic images, which are all human-annotated. Again, we found that models’ performance on our synthetic scenes is highly correlated with their performance on TallyQA (r=0.714). See the Appendix H.4 for additional details on this analysis.

Last but not least, in addition to strong correlations with evaluations using real images, we argue that our evaluation with synthetic images offers two unique advantages over real images. First, our human evaluation shows that human annotators achieve near-perfect performance (99.4, 99.73, 97.33%) across all image sources, including the synthetic ones, whereas MLMs perform relatively poorly on 2D and 3D synthetic images compared to real images. This suggests that our evaluation with synthetic images consists of reasonable tasks that are especially challenging to the models. Second, evaluation with synthetic images is free from data contamination issues that might occur with real images. As the community continues to scale up training data without always fully disclosing the data sources, many publicly available real images typically used for evaluation might have already been seen by the models during training, making the evaluations with these images less rigorous/challenging.  Nevertheless, our synthetic images are totally new and have not been seen by any models during training. Therefore, our evaluation with these images is arguably more robust and reliable.




**Reviewers jor3 ask for more comparison with existing benchmarks**

We conduct a comparison between 11 models’ performance rankings (including GPT4o, PaliGemma-3B, LLaVA, Qwen, Phi-3-vision-3B, Idefics2-8B, GLM-4v, CogVLM2-19B, InternVL-Chat etc.) on TaskMeAnything and on other benchmarks, including MMMU, MMBench, etc. Here are the models we used:

We found that there is a positive correlation between models’ rankings on TaskMeAnything and on other benchmarks, with strong correlations (>=0.8) on the most commonly used ones such as MMMU and MMBench. Notably, the average correlation between TaskMeAnything and the other benchmarks is 0.77, which is greater than that of some other benchmarks such as ScienceQA and LLaVABench (0.70 and 0.57). These results suggest that the evaluation results on our benchmark align with those on existing benchmarks (**Figure 2 of the attached PDF**).

**Our response to each reviewer will be posted separately. We look forward to interacting with you further during the discussion period!**

---

### Decision · Program_Chairs · 2024-09-26

**Decision:**

Accept (Poster)

**Comment:**

The paper presents an innovative approach to automatic benchmark generation, focusing on evaluating multimodal language models (MLMs) under various tasks. The core contribution lies in its flexible, scalable benchmark generation engine, capable of creating a large array of tasks based on user-driven inputs. This system supports fine-grained evaluation of MLMs and aims to offer insights into the performance of existing models. The consensus among the reviewers is that the paper provides a meaningful contribution to MLM evaluation with its novel task generation engine. The detailed experiments and insightful analysis justify its relevance, despite some limitations in scope and generalizability. The authors have adequately addressed most concerns in their rebuttal, particularly regarding the system's extendability and correlation between synthetic and real-world performance. The AC agrees with the reviewers to accept the submission.